# Reflective Hamiltonian Monte Carlo:
# Mixing Analysis and Application to Sampling on Stiefel Manifold

Kwangmin Lee [1]   Yeonhee Park [2]   Sewon Park [3]

## Abstract

Sampling from distributions with bounded supports is a fundamental challenge in constrained statistical inference. Reflective Hamiltonian Monte Carlo (ReHMC) provides a useful approach for this setting, but relies on convexity assumptions and lacks non-asymptotic mixing-time bounds. To bridge this gap, we propose a convex-container-plus-thinning framework that extends ReHMC beyond smooth convex supports to a broad class of bounded supports. We establish the first non-asymptotic total-variation mixing-time bounds for ReHMC, achieving a polynomial dimension dependence of $O(d^2)$ for $L$-smooth targets, though with exponential dependence on smoothness parameters. Under an additional $m$-strong convexity assumption, we derive a sharper bound that eliminates this exponential dependence. We further apply this approach to sampling on the Stiefel manifold via a well-conditioned polar reparameterization and demonstrate improved numerical stability and computational efficiency in simulation studies.

## 1. Introduction

Markov chain Monte Carlo (MCMC) methods are a central tool for sampling from probability distributions. Among these methods, Hamiltonian Monte Carlo (HMC) leverages approximate Hamiltonian dynamics to efficiently explore high-probability regions of a target density. Classical HMC, introduced by Neal et al. (2011), is naturally formulated for unconstrained domains, where the dynamics evolve freely in $\mathbb{R}^d$.

[1]Department of Big Data Convergence, Chonnam National University, Gwangju, Republic of Korea [2]Department of Statistics, Sungkyunkwan University, Seoul, Republic of Korea [3]Department of Statistics, Sookmyung Women's University, Seoul, Republic of Korea. Correspondence to: Sewon Park <swpark0413@sookmyung.ac.kr>.

*Proceedings of the 43rd International Conference on Machine Learning*, Seoul, South Korea. PMLR 306, 2026. Copyright 2026 by the author(s).

In many applications, the target density is supported on a bounded or otherwise constrained domain, including truncated supports and manifolds. Extending HMC to such settings requires modifying the underlying dynamics to enforce the constraints, since unconstrained Hamiltonian trajectories may exit the admissible region. For example, Mohasel Afshar & Domke (2015) have proposed reflective or refractive versions of HMC, where trajectories bounce or bend when they hit the boundary. While these methods often perform well in practice, theoretical results on their convergence rates in constrained settings are not well established.

An important theoretical problem for any MCMC algorithm is its mixing-time, which quantifies how many iterations are required for the chain to approach the target distribution within a prescribed accuracy. For unconstrained targets, the mixing behavior of MCMC algorithms has been extensively studied. For example, when Metropolized HMC targets an $m$-strongly log-concave and $L$-smooth density on $\mathbb{R}^d$, the mixing time scales as $\kappa d \log(1/\varepsilon)$, where $\kappa = L/m$ and $\varepsilon > 0$ is the prescribed total variation accuracy (Lee et al., 2020).

In contrast, for target densities supported on bounded domains and sampled using reflective HMC (ReHMC), theoretical mixing-time results are unavailable. While recent work on ReHMC for truncated log-concave distributions (Chalkis et al., 2023) shows promising empirical results, establishing a rigorous mixing-time bound remains an open challenge. Although an earlier preprint of Chalkis et al. (2023) attempted to provide a polynomial mixing-time bound under $L$-smoothness and $m$-strong convexity, the final published version focuses on empirical performance, leaving the theoretical guarantee for this setting yet to be established.

In this paper, we fill this gap by proving mixing-time bounds for ReHMC on bounded domains. We consider target densities supported on compact convex sets with smooth boundary and analyze Metropolized ReHMC under two settings. First, assuming only that the negative log-density is $L$-smooth, we obtain a general mixing-time bound whose dimension dependence is of order $d^2$ (up to logarithmic factors), although the bound depends exponentially on certain smoothness parameters. Second, under an additional $m$-strong convexity assumption, we derive a sharper result

that removes this exponential dependence while preserving the same $d^2$ scaling in dimension. In terms of leading dimension dependence, our bounds improve over existing Langevin-type guarantees, which scale as $d^3$ (Srinivasan et al., 2024), as well as Hit-and-Run–type bounds with even higher polynomial dependence (Narayanan & Srivastava, 2022).

Moreover, Mohasel Afshar & Domke (2015) and Chalkis et al. (2023) require the support set to be a polytope and a convex body, respectively, so that reflections are well defined. Beyond these favorable geometric settings, we develop a convex-container-plus-thinning method that enables ReHMC to be applied to general bounded supports. Specifically, we embed the target support into a smooth convex container, extend the potential smoothly outside the original set, and run ReHMC on the container. Exact samples from the original target density are then obtained by retaining only those samples that fall within the target support, since the extended potential coincides with the original one on the support. This approach enables the application of ReHMC and its mixing-time analysis to a much broader class of bounded supports, including nonconvex domains with a $C^2$ boundary and nonsmooth domains such as polytopes; we provide concrete implementation guidelines for the construction in Section 3 and Appendix D.

As an application, we study sampling from distributions on the Stiefel manifold, the set of $d \times r$ orthonormal matrices, which arises frequently in statistical models with orthogonal parameters, including factor analysis and dimension reduction. Existing MCMC methods include Bingham-based Gibbs samplers (Hoff, 2009), geodesic–flow MCMC methods (Byrne & Girolami, 2013), and polar-expansion–based Euclidean reparameterizations (Jauch et al., 2020; Pourzanjani et al., 2020; Jauch et al., 2021). While polar-expansion–based methods are attractive for their simplicity and compatibility with Euclidean samplers, they can suffer from numerical instability near rank-deficient matrices, leading to ill-conditioned potentials and slow mixing. We propose a modified polar expansion with a uniformly bounded condition number, and when combined with our convex-container-plus-thinning method, obtain improved stability and efficiency in experiments.

The remainder of the paper is organized as follows. In Section 2, we review HMC with the leapfrog integrator on unconstrained Euclidean spaces and introduce ReHMC for sampling from densities supported on bounded sets. In Section 3, we present a convex-container-plus-thinning method that enables ReHMC to handle general bounded supports. In Section 4, we establish non-asymptotic mixing-time guarantees for ReHMC under general smoothness and strong convexity assumptions. In Section 5, we apply the proposed framework to sampling on the Stiefel manifold via a well-

---

**Algorithm 1** Generic Metropolized HMC update

1: **Input:** current position $q$, step size $\eta > 0$, mass matrix $M \succ 0$
2: Draw momentum $p \sim \mathcal{N}(\mathbf{0}, M)$
3: $p' \leftarrow p - \frac{\eta}{2} \nabla U(q)$
4: $(\tilde{q}, p') \leftarrow \text{POSITIONUPDATE}(q, p', \eta, M)$
5: $\tilde{p} \leftarrow p' - \frac{\eta}{2} \nabla U(\tilde{q})$
6: Accept $(\tilde{q}, \tilde{p})$ with Metropolis–Hastings acceptance probability;
7:     otherwise set $(\tilde{q}, \tilde{p}) \leftarrow (q, p)$
8: (Optional) set $\tilde{p} \leftarrow -\tilde{p}$
9: **Output:** next position $q^{\text{new}} = \tilde{q}$

---

conditioned polar reparameterization. Section 6 presents numerical experiments that demonstrate the stability and efficiency of our approach. Finally, Section 7 concludes with a discussion of limitations and future directions.

## 2. Reflective Hamiltonian Monte Carlo

We briefly review ReHMC, originally proposed by Mohasel Afshar & Domke (2015) and further developed by Chalkis et al. (2023), for sampling from target densities supported on bounded domains. To clarify the relationship between unconstrained HMC and its reflective extension, we first present a generic Metropolized HMC update, and then specify how the position update is modified in the presence of boundary constraints.

**Generic Metropolized HMC update.** Let $\pi(q) \propto \exp\{-U(q)\}$ be a target density on $\mathbb{R}^d$. HMC augments the position variable $q$ with a momentum $p \in \mathbb{R}^d$ with mass matrix $M \succ 0$, targeting the joint density

$$\pi(q, p) \propto \exp\{-U(q) - \tfrac{1}{2} p^\top M^{-1} p\}.$$

We refer to Algorithm 1 as a generic Metropolized HMC update. The momentum updates in lines 3 and 5 are common to both unconstrained HMC and ReHMC. In contrast, the position update in line 4, $\text{POSITIONUPDATE}(q, p', \eta, M)$, is the only component that must be adapted to the geometry of the support of the target density.

**Unconstrained HMC.** The position update on line 4 of Algorithm 1 is given by

$$\text{POSITIONUPDATE}(q, p', \eta, M) = (q + \eta M^{-1} p', \ p'),$$

which recovers the standard Metropolized HMC algorithm.

**Reflective HMC.** We consider target densities supported on a bounded set: $\pi(q) \propto e^{-U(q)} \mathbf{1}\{q \in K\}$, where $K \subset \mathbb{R}^d$ has a $C^2$ boundary so that outward unit normals are well defined. ReHMC modifies only the position update

---

**Algorithm 2** REFLECTIVEPOSITION($\boldsymbol{q}, \boldsymbol{p}', \eta, \boldsymbol{M}, K$)

---

1: **Input:** $\boldsymbol{q} \in K$, momentum $\boldsymbol{p}'$, step size $\eta$, mass $\boldsymbol{M}$, convex compact $K$
2: $\mathbf{d} \leftarrow \eta \boldsymbol{M}^{-1} \boldsymbol{p}', t \leftarrow 0$
3: **while** $t < 1$ **do**
4:     Find first hit time $t_1 \in (0, 1-t]$ of $\boldsymbol{q} + s\mathbf{d}$ with $\partial K$
5:     **if** no hit on $(0, 1-t]$ **then**
6:        $\boldsymbol{q} \leftarrow \boldsymbol{q} + (1-t)\mathbf{d}; t \leftarrow 1$
7:     **else**
8:        $\boldsymbol{q} \leftarrow \boldsymbol{q} + t_1 \mathbf{d}$
9:        $\boldsymbol{n}_* \leftarrow$ outward unit normal at $\boldsymbol{q}$
10:       $\boldsymbol{p}' \leftarrow \boldsymbol{p}' - 2\langle \boldsymbol{p}', \boldsymbol{n}_* \rangle \boldsymbol{n}_*$
11:       $\mathbf{d} \leftarrow \eta \boldsymbol{M}^{-1} \boldsymbol{p}'; t \leftarrow t + t_1$
12:     **end if**
13: **end while**
14: **Output:** $(\boldsymbol{q}, \boldsymbol{p}')$

---

on line 4 in Algorithm 1. Whenever a straight-line position move would exit $K$, the trajectory is reflected specularly at the boundary hit point. Formally, the position update is defined as

$$\text{POSITIONUPDATE}(\boldsymbol{q}, \boldsymbol{p}', \eta, \boldsymbol{M})$$
$$= \text{REFLECTIVEPOSITION}(\boldsymbol{q}, \boldsymbol{p}', \eta, \boldsymbol{M}, K),$$

where REFLECTIVEPOSITION performs straight-line motion with specular reflections at $\partial K$ and is given in Algorithm 2. Further details of the reflection mechanism are provided in Mohasel Afshar & Domke (2015) and Chalkis et al. (2023).

## 3. A Convex-Container and Thinning Construction for General Bounded Supports

ReHMC in Section 2 requires the support set $K$ to have a $C^2$ boundary so that outward unit normals are well defined. We extend the method to a general bounded set $K \subset \mathbb{R}^d$, which may be nonconvex or nonsmooth, by embedding $K$ into a smooth convex container and extending the potential outside $K$.

**Convex container and smooth extension.** We embed $K$ into a compact convex set $\widetilde{K} \supseteq K$ with a $C^2$ boundary and perform reflective HMC on $\widetilde{K}$. On $\widetilde{K}$, we construct an extended potential $\widetilde{U}$ that coincides with the original potential $U$ on $K$ and is smoothly extended outside $K$. The resulting target density is defined as

$$\widetilde{\pi}(\boldsymbol{q}) \propto \exp\{-\widetilde{U}(\boldsymbol{q})\} \mathbf{1}\{\boldsymbol{q} \in \widetilde{K}\}.$$

By construction, the support of $\widetilde{\pi}$, namely $\tilde{K}$, satisfies the geometric regularity conditions required to apply ReHMC.

Moreover, to ensure that ReHMC is well defined on $\widetilde{K}$, the extended potential $\widetilde{U}$ must admit computable gradients and have bounded smoothness. These properties are essential for the mixing–time guarantees established in Section 4.

General extension results, such as Whitney–type extension theorems (Whitney, 1934), guarantee the existence of smooth extensions. Here we use an explicit penalty-based construction that satisfies the required regularity conditions while retaining a simple and interpretable form.

Specifically, we propose the extended potential on the compact convex container $\widetilde{K} \supseteq K$ as

$$\widetilde{U}(\boldsymbol{q}) = U(\boldsymbol{q}) + \lambda \, \psi\left(\frac{\varphi(\boldsymbol{q})}{b}\right). \tag{1}$$

Here, $\varphi$ is a $C^2$ boundary-violation surrogate on $\widetilde{K}$, satisfying $\varphi(\boldsymbol{q}) = 0$ for $\boldsymbol{q} \in K$ and $\varphi(\boldsymbol{q}) > 0$ for $\boldsymbol{q} \in \widetilde{K} \setminus K$. The penalty term $\lambda\psi(\varphi(\boldsymbol{q})/b)$ then induces a repulsive force that discourages trajectories from drifting deep into $\widetilde{K} \setminus K$, as illustrated in Figure 1. The penalty function $\psi$ is chosen to satisfy the required regularity conditions, namely $\psi(s) = \psi'(s) = \psi''(s) = 0$ for all $s \leq 0$.

Under these constructions, $\widetilde{U}$ coincides with $U$ on the target support $K$, and Proposition 3.1 guarantees that $\widetilde{U}$ admits uniformly bounded derivatives on $\widetilde{K}$, thereby ensuring the stability of Hamiltonian dynamics.

**Proposition 3.1** (Global Hessian bound for the extended potential)**.** *Under the smoothness and boundedness conditions on the boundary surrogate $\varphi$ and the penalty function $\psi$ specified above, the penalty-based extended potential $\widetilde{U}$ admits a globally bounded Hessian on $\widetilde{K}$: there exists a finite constant $C_\star > 0$ such that*

$$\|\nabla^2 \widetilde{U}(\boldsymbol{q})\|_2 \leq C_\star, \qquad \forall \boldsymbol{q} \in \widetilde{K}.$$

*Equivalently, the gradient $\nabla\widetilde{U}$ is globally Lipschitz on $\widetilde{K}$.*

The proof of Proposition 3.1 is provided in Appendix A.

In practical implementation, we use a smooth boundary-violation surrogate of the softplus form

$$\varphi_a(x) = \frac{1}{a} \log\{1 + \exp(a\phi(x))\},$$

where $\phi$ is a differentiable signed-distance-like function associated with the boundary $\partial K$. Specifically, $\phi$ encodes feasibility by its sign: $\phi(x) \leq 0$ for $x \in K$ and $\phi(x) > 0$ for $x \notin K$, with $|\phi(x)|$ increasing as $x$ moves farther away from $\partial K$. Hence, $\varphi_a$ is a differentiable approximation to the boundary violation $\max\{\phi, 0\}$, balancing smoothness and computational efficiency. Since the softplus surrogate does not satisfy $\varphi_\sigma(x) = 0$ exactly on $K$, the penalty term introduces a small residual on the original support. For the quartic penalty, this residual is uniformly bounded by $O(a^{-4})$

and can be made arbitrarily small by increasing $a$; see Appendix A.3 for details.

**Sampling by thinning.**    While ReHMC is performed on the extended space $\widetilde{K}$ with target distribution $\widetilde{\pi}$, exact inference for the original target distribution $\pi$ is recovered by exploiting the fact that $\widetilde{U}$ coincides with $U$ on $K$. In particular, the conditional distribution of $\widetilde{\pi}$ given $\boldsymbol{q} \in K$ equals $\pi$.

Accordingly, expectations with respect to the target distribution $\pi$ can be expressed in terms of expectations under the extended distribution $\widetilde{\pi}$ as

$$\mathbb{E}_\pi[f] \;=\; \frac{\mathbb{E}_{\widetilde{\pi}}[f(\boldsymbol{q})\,\mathbf{1}\{\boldsymbol{q} \in K\}]}{\mathbb{E}_{\widetilde{\pi}}[\mathbf{1}\{\boldsymbol{q} \in K\}]}, \qquad (2)$$

for any integrable function $f$. In practice, samples $\boldsymbol{q}_1, \ldots, \boldsymbol{q}_N$ are generated by running ReHMC targeting $\widetilde{\pi}$. Substituting the expectations in (2) with their corresponding Monte Carlo averages yields the self-normalized estimator

$$\widehat{\mathbb{E}}_\pi[f] \;=\; \frac{\sum_{t=1}^{N} f(\boldsymbol{q}_t)\,\mathbf{1}\{\boldsymbol{q}_t \in K\}}{\sum_{t=1}^{N} \mathbf{1}\{\boldsymbol{q}_t \in K\}},$$

which corresponds to averaging $f(\boldsymbol{q}_t)$ over the retained samples $\{\boldsymbol{q}_t \in K\}$. Under standard ergodicity assumptions for the Markov chain targeting $\widetilde{\pi}$, this estimator is consistent for $\mathbb{E}_\pi[f]$.

It therefore suffices to analyze the convergence properties of the Markov chain targeting the extended distribution $\widetilde{\pi}$. In the next section, we focus on establishing mixing–time guarantees for ReHMC under $\widetilde{\pi}$, from which consistency and efficiency of inference under $\pi$ follow directly via the thinning construction.

# 4. Mixing Analysis

This section establishes non-asymptotic total-variation mixing-time bounds for the ReHMC method. All proofs are deferred to Appendix B. We first derive explicit bounds for compact convex supports with smooth boundaries, and then extend the analysis to the convex container plus thinning construction introduced in Section 3.

**Proof strategy and relation to prior work.**    The high-level proof architecture is based on combining local TV contraction with a blocking-conductance bound, following the framework of Lee et al. (2020). Our main technical novelty lies in establishing local TV contraction for the reflective proposal (Lemma B.5), which requires new reflection-geometry estimates (Lemmas B.7-B.19) specific to the boundary-reflection setting; these arguments have no counterpart in unconstrained HMC. The warm-start construction (Lemma 4.4) adapts the truncated-Gaussian initialization of Chalkis et al. (2023, Theorem 2); however,

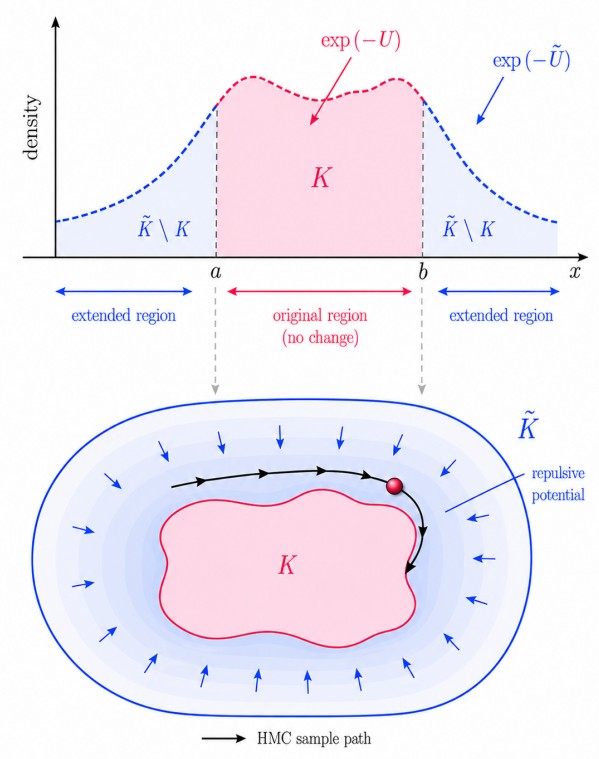

*Figure 1.* (**Top**) One-dimensional illustration of the original unnormalized density $\exp\{-U\}$ and its extension $\exp\{-\widetilde{U}\}$, which coincide inside $K$ (pink). (**Bottom**) Geometric view of the smooth container construction, where $K$ is embedded into a convex set $\widetilde{K}$ and a repulsive penalty acts only outside $K$.

that work does not establish a non-asymptotic mixing-time bound for ReHMC, which is the central contribution of this paper.

**Mixing-time.**    Let $\mathcal{T}$ denote the one-step Markov transition kernel induced by Algorithm 1, with stationary distribution $\pi$. For each $i \geq 0$, let $\mathcal{T}^i$ denote the $i$-step transition kernel. For $k \geq 1$, we define the Cesàro average of the iterated kernels by $\bar{\mathcal{T}}_k := \frac{1}{k}\sum_{i=0}^{k-1} \mathcal{T}^i$. Given $\varepsilon \in (0,1)$, the (Cesàro-averaged) mixing time $k_*$ is defined as the smallest integer such that

$$\left\|\pi_0 \bar{\mathcal{T}}_k - \pi\right\|_{\mathrm{TV}} \;\leq\; \varepsilon \qquad \text{for all } k \geq k_*,$$

where $\pi_0$ denotes the initial distribution.

**Assumptions.**    We consider target densities of the form

$$\pi(x) \;\propto\; \exp\{-U(x)\}\,\mathbf{1}\{x \in K\},$$

where $K \subseteq \mathbb{R}^d$ is nonempty, compact, and convex. We impose the following assumptions on the geometry of $K$ and the regularity of the potential $U$.

(A1) **(Tubular neighborhood and curvature control).** The boundary $\partial K$ is $C^2$. Its principal curvatures are uniformly bounded: there exists $\bar\kappa \in (0, \infty)$ such that

$$0 \leq \kappa_i(p) \leq \bar\kappa \qquad (\forall\, p \in \partial K,\ i = 1, \ldots, d-1).$$

Moreover, there exists $\rho \in (0, 1/\bar\kappa)$ for which the normal map

$$\Phi:\ \partial K \times (-\rho, \rho) \to \mathcal{N}_\rho, \qquad \Phi(p, s) = p + s\, n(p),$$

is a $C^1$ diffeomorphism onto the tubular neighborhood $\mathcal{N}_\rho := \{x \in \mathbb{R}^d :\ \mathrm{dist}(x, \partial K) < \rho\}$.

(A2) **(Smooth potential).** The gradient $\nabla U$ is $L$-Lipschitz on $K$, equivalently, $U$ is $L$-smooth on $K$. In addition, the gradient scale is bounded:

$$G := \sup_{x \in K} \|\nabla U(x)\| < \infty.$$

Assumption (A1) encodes a smoothness condition on the support set $K$, quantified through curvature bounds and the existence of a tubular neighborhood around the boundary. This assumption controls the reflection geometry near $\partial K$ and is required by both Theorems 4.3 and 4.5, regardless of whether $U$ is convex. It is satisfied whenever $\partial K$ is a compact $C^2$ hypersurface (Foote, 1984). We state it in explicit quantitative form because the geometric constants $\rho$ and $\bar\kappa$ enter directly into the step-size conditions through $L_n(\rho) = \bar\kappa/(1 - \bar\kappa\rho)$. Assumption (A2) imposes a smoothness condition on the target potential $U$, with the parameters $L$ and $G$ quantifying this regularity.

**Warmness of the initialization.** Following standard practice in non-asymptotic mixing-time analysis, we quantify the quality of the initial distribution via a warmness parameter.

**Definition 4.1** (Warmness). An initial distribution $\pi_0$ is said to be $\beta$-*warm* with respect to the target distribution $\pi$ if

$$\sup_x \frac{\pi_0(x)}{\pi(x)} \leq \beta$$

The warmness parameter $\beta$ measures the discrepancy between the initial distribution and the target distribution and appears explicitly in the mixing-time bounds derived below.

**General $L$-smooth case.** We begin with the general setting in which the potential $U$ is $L$-smooth on $K$, without assuming convexity of $U$. Theorem 4.3 provides a non-asymptotic total-variation mixing-time bound for ReHMC in this setting. The bound depends on geometric parameters of the support set $K$, regularity parameters of $U$, the dimension $d$, the warmness parameter $\beta$ of the initial distribution, and the step size $\eta$.

By combining Theorem 4.3 with the warm-start bound in Lemma 4.4, and choosing the step size $\eta$ to optimize the resulting bound, we obtain the following mixing-time estimate:

$$k \ \gtrsim\ R\, d^2\, \exp\!\big(C_1\, RG + C_2\, LR^2\big)\, \log\!\Big(\tfrac{1}{\varepsilon}\Big)^2, \quad (3)$$

where $C_1, C_2 > 0$ are constants independent of the dimension $d$.

*Remark* 4.2 (Dimension scaling). The $O(d^2)$ dependence, compared to the $O(d)$ bound for unconstrained HMC (Lee et al., 2020), arises from a more restrictive step-size requirement in the reflective setting. Even when two starting points $x, y$ are close, their leapfrog trajectories may undergo reflections at different boundary points or at different angles, causing post-reflection proposals to diverge. Ensuring that the transition kernels $\mathcal{T}_x$ and $\mathcal{T}_y$ remain close in total variation requires $\eta \lesssim d^{-1}$ (Lemma B.5), yielding $1/\eta^2 = O(d^2)$. In contrast, the unconstrained case requires only $\eta \lesssim d^{-1/2}$, giving $O(d)$. We also verify this scaling empirically in Appendix E.

In terms of the dimension $d$, the leading-order dependence is quadratic. However, the bound exhibits exponential dependence on the regularity parameters $G$, $L$, and the diameter scale $R$. Similar exponential factors in smoothness and geometry parameters also appear in nonconvex mixing-time bounds for Langevin-type methods; see, for example, Ma et al. (2019). When $G$, $L$, and $R$ are small, the bound yields a quantitative mixing-time guarantee even in high dimensions, whereas it can become very loose as these parameters increase. This motivates a sharper analysis under additional structural assumptions, which we consider next.

**Theorem 4.3** (General smooth case). *For notational simplicity, the proofs are carried out under $M = I$; the extension to general $M \succ 0$ is discussed in Appendix B (Remark B.1). Consider the target density $\pi(x) \propto \exp\{-U(x)\}\, \mathbf{1}\{x \in K\}$ with assumptions (A1) and (A2). Suppose that $\mathrm{diam}(K) \leq 2R$ for some $R > 0$. Let $\mathcal{T}$ denote the transition kernel of the reflective Metropolized Hamiltonian Monte Carlo algorithm in Algorithm 1, with stationary density $\pi$. If the step size $\eta$ satisfies*

$$\eta \ \lesssim\ \frac{\rho}{\sqrt{d} + G} \ \wedge\ \frac{1}{G\sqrt{d}} \ \wedge\ \frac{1}{\sqrt{L}\, d}$$
$$\wedge\ \frac{1}{\sqrt{L}\, G} \ \wedge\ \frac{1}{d\, L_n(\rho)} \ \wedge\ \frac{1}{G^2 L_n(\rho)},$$

*where $L_n(\rho) := \dfrac{\bar\kappa}{1 - \bar\kappa\rho}$, then, for any $\varepsilon \in (0, 1)$, $\big\|\bar{\mathcal{T}}_k \pi_0 - \pi\big\|_{\mathrm{TV}} \leq \varepsilon$ is satisfied for all*

$$k \ \gtrsim\ \left[\frac{R}{\eta^2}\, \exp\!\big(8RG + (8L+2)R^2\big) + 1\right] \log\!\Big(\tfrac{\beta}{16\varepsilon}\Big) \log\!\Big(\tfrac{1}{\varepsilon}\Big).$$

**Lemma 4.4** (Warm-start bound). *Let $K \subset \mathbb{R}^d$ be convex and bounded, and let $U : K \to \mathbb{R}$ be $L$-smooth on $K$. Suppose that $\mathrm{diam}(K) \leq 2R$. Then, for a truncated Gaussian initialization centered at $x_\star \in K$ with covariance $\sigma^2 \boldsymbol{I}$, the warmness parameter satisfies*

$$\beta \ \leq \ \exp\Big( 4R\|\nabla U(x_\star)\| + 4LR^2 + 2\sigma^{-2}R^2 \Big).$$

**Strongly log-concave case.** We turn to the regime where the $U$ satisfies an additional strong convexity condition:

(A3) There exists $m > 0$ such that, for all $x, y \in K$,

$$U(y) - U(x) - \nabla U(x)^\mathsf{T}(y - x) \ \geq \ \frac{m}{2}\|y - x\|_2^2.$$

Under this structural assumption, the exponential dependence on smoothness-related parameters that appears in the general $L$-smooth bound can be removed as shown in Theorem 4.5. Theorem 4.5 provides a refined total-variation mixing-time bound whose dependence on the problem parameters is polynomial, including the step size $\eta$.

**Theorem 4.5** (Strongly log-concave case). *Consider the target density $\pi(x) \ \propto \ \exp\{-U(x)\}\,\mathbf{1}\{x \in K\}$ with assumptions (A1)–(A3). Let $\mathcal{T}$ denote the transition kernel of Algorithm 1 with stationary density $\pi$, and suppose that the initial distribution $\pi_0$ is $\beta$-warm with respect to $\pi$. There exists a universal constant $c_0 > 0$ such that if the step size $\eta$ satisfies*

$$\eta \ \leq \ c_0 \Big( (G\sqrt{d})^{-1} \ \wedge \ (L^{1/2}d)^{-1} \ \wedge \ (L^{1/2}G)^{-1}$$
$$\wedge \ (d\, L_n(\rho))^{-1} \ \wedge \ (G^2 L_n(\rho))^{-1} \ \wedge \ m^{-1/2} \Big), \quad (4)$$

*then for any $\varepsilon \in (0, 1/2)$, $\big\|\bar{\mathcal{T}}_k \pi_0 - \pi\big\|_{\mathrm{TV}} \ \leq \ \varepsilon$ for all*

$$k \ \gtrsim \ \frac{1}{m\,\eta^2} \ \log\Big(\log(\beta/\varepsilon) + 1\Big) \log\Big(\tfrac{1}{\varepsilon}\Big). \quad (5)$$

The mixing-time bound in Theorem 4.5 depends on the warmness parameter $\beta$, which we instantiate using the warm-start construction of Chalkis et al. (2023). For a truncated Gaussian initialization centered at the minimizer $x^* \ = \ \arg\min_{x \in K} U(x)$ with covariance matrix $L^{-1} I_d$, their Theorem 2 in Chalkis et al. (2023) yields

$$\beta \ = \ O\big((\kappa\gamma)^d\big),$$

where $\kappa = L/m$ and $\gamma$ denotes the sandwiching ratio of $K$ around $x^*$, defined by

$$\gamma \ := \ \inf_{R > r > 0} \Big\{ R/r \ : \ B(x^*, r) \subseteq K \subseteq B(x^*, R) \Big\}.$$

Substituting this warm-start bound into Theorem 4.5 and choosing the step size $\eta \asymp (L^{1/2}d)^{-1}$, which satisfies the

step-size conditions under the assumption $L_n(\rho)^2 \lesssim L$, we obtain

$$k \ \gtrsim \ \kappa\, d^2\, \log\Big(d\log(\kappa\gamma) + \log(1/\varepsilon)\Big) \, \log\Big(\tfrac{1}{\varepsilon}\Big). \quad (6)$$

In particular, the leading dependence on the dimension is of order $d^2$ up to logarithmic factors. The remaining parameter dependence is polynomial in the condition number $\kappa$ and the geometric quantity $\gamma$, while the exponential dependence on smoothness parameters that appears in the general $L$-smooth case is eliminated.

**Mixing time for the convex-container-plus-thinning method.** We analyze the mixing-time of the convex-container-plus-thinning construction. The original target density is

$$\pi(x) \ \propto \ \exp\{-U(x)\}\,\mathbf{1}\{x \in K\},$$

where $K \subset \mathbb{R}^d$ is an arbitrary bounded set. We do not assume that $K$ satisfies Assumption (A1); we only assume that $U$ satisfies the smoothness condition (A2) on $K$.

Let $\widetilde{K} \supseteq K$ be a compact convex set with $C^2$ boundary that satisfies Assumption (A1), and let $\widetilde{U}$ be the smoothly extended potential constructed in Section 3, satisfying $\widetilde{U} \equiv U$ on $K$. By Proposition 3.1, the extended potential $\widetilde{U}$ satisfies an analogue of Assumption (A2) on $\widetilde{K}$, possibly with modified constants $(\widetilde{L}, \widetilde{G})$ that depend only on $(L, G)$ and the extension parameters. We define the extended target

$$\widetilde{\pi}(x) \ \propto \ \exp\{-\widetilde{U}(x)\}\,\mathbf{1}\{x \in \widetilde{K}\}.$$

Since $\widetilde{U} = U$ on $K$, we have $\widetilde{\pi}(\cdot \mid K) = \pi$.

Let $\mu_k$ denote the law of the $k$-th iterate (or its Cesàro average) of ReHMC targeting $\widetilde{\pi}$, and define the thinned law $\mu_k^{\mathrm{thin}} := \mathrm{Law}(X_k \mid X_k \in K)$. Then the mixing-time of the thinned chain is bounded by

$$k_{\mathrm{thin}}(\varepsilon) \ \leq \ k_{\widetilde{\pi}}\Big(\frac{\alpha\varepsilon}{2}\Big), \quad \alpha := \widetilde{\pi}(K),$$

where $k_{\widetilde{\pi}}(\cdot)$ denotes the corresponding mixing-time bound for ReHMC targeting $\widetilde{\pi}$; concretely, one may take $k_{\widetilde{\pi}}(\varepsilon)$ to be the bound in (3) or (6), depending on which assumptions hold.

This bound follows from the fact that to ensure $\|\mu_k^{\mathrm{thin}} - \pi\|_{\mathrm{TV}} \leq \varepsilon$, it suffices that $\|\mu_k - \widetilde{\pi}\|_{\mathrm{TV}} \ \leq \ \frac{\alpha\varepsilon}{2}$, which is a direct consequence of a standard total-variation inequality for conditional distributions.

Substituting the bounds of Theorems 4.3 and 4.5 yields, respectively,

$$k_{\mathrm{thin}}(\varepsilon) \ \lesssim \ \Big(\frac{R}{\eta}\Big)^2 \exp\big(8RG + (8L+2)R^2 + 1\big)$$
$$\times \log\frac{\beta}{8\alpha\varepsilon} \, \log\frac{2}{\alpha\varepsilon}$$

in the general smooth case, and

$$k_{\text{thin}}(\varepsilon) \;\lesssim\; \frac{1}{m\eta^2}\, \log\!\Big(\log\frac{2\beta}{\alpha\varepsilon} + 1\Big)\, \log\frac{2}{\alpha\varepsilon}$$

in the strongly log-concave case.

Since $\alpha = \widetilde{\pi}(K)$ appears only inside logarithmic terms in these bounds, the convex–container plus thinning method preserves the same mixing order whenever $\widetilde{\pi}(K)$ is bounded away from zero.

## 5. HMC for Densities Supported on the Stiefel Manifold

We consider a target density $\pi(\mathbf{\Gamma})$ supported on the Stiefel manifold

$$V_{r,u} \;:=\; \{\mathbf{\Gamma} \in \mathbb{R}^{r \times u} :\, \mathbf{\Gamma}^\top \mathbf{\Gamma} = I_u\}, \qquad r \geq u.$$

To facilitate sampling from $\pi(\mathbf{\Gamma})$, we employ the polar expansion framework of Jauch et al. (2021). Specifically, (i) we introduce an auxiliary parameter to obtain an extended representation of $\mathbf{\Gamma}$, (ii) we embed this representation into a constrained Euclidean space, and (iii) we apply the ReHMC algorithm to perform sampling in the constrained space.

**(i) Auxiliary-variable representation.** We introduce an auxiliary positive–definite matrix $\mathbf{S} \in \mathcal{S}^u_{++}$ and consider the joint target $\pi(\mathbf{\Gamma}, \mathbf{S}) = \pi(\mathbf{\Gamma})\,\pi_S(\mathbf{S})$, where $\pi_S$ is a user–specified auxiliary density. While Jauch et al. (2021) consider a (non–truncated) Wishart distribution for $\pi_S$, we instead adopt the following truncated Wishart–type density in order to obtain mixing–time guarantees:

$$\pi_S(\mathbf{S}; c, M) \;\propto\; |\mathbf{S}|^{\frac{r-u-1}{2}}\, \mathbf{1}\{\lambda_{\min}(\mathbf{S}) \geq c\}\, \mathbf{1}\{\text{tr}(\mathbf{S}) \leq M\}, \tag{7}$$

where $c > 0$ and $M < \infty$.

This truncation is imposed to ensure that the smoothness parameters of the reparameterized target are uniformly bounded, as established after the constrained Euclidean embedding. By construction, $\mathbf{\Gamma}$ and $\mathbf{S}$ are independent under the joint target $\pi(\mathbf{\Gamma}, \mathbf{S})$, and the marginal distribution of $\mathbf{\Gamma}$ is exactly $\pi(\mathbf{\Gamma})$, regardless of the specific choice of the auxiliary density $\pi_S$. Consequently, if we sample $(\mathbf{\Gamma}, \mathbf{S})$ from $\pi(\mathbf{\Gamma}, \mathbf{S})$ and discard $\mathbf{S}$, the retained samples of $\mathbf{\Gamma}$ follow the original target density $\pi(\mathbf{\Gamma})$.

**(ii) Constrained Euclidean embedding.** We embed the auxiliary-variable representation $(\mathbf{\Gamma}, \mathbf{S})$ into a Euclidean parameter $\mathbf{B}$. For any $\mathbf{B} \in \mathbb{R}^{r \times u}$ with full column rank, define

$$\mathbf{\Gamma}(\mathbf{B}) := \mathbf{B}(\mathbf{B}^\top \mathbf{B})^{-1/2}, \qquad \mathbf{S}(\mathbf{B}) := \mathbf{B}^\top \mathbf{B}.$$

These mappings establish a bijection between

$$\{\mathbf{B} \in \mathbb{R}^{r \times u} : \text{rank}(\mathbf{B}) = u\} \quad \text{and} \quad V_{r,u} \times \mathcal{S}^u_{++},$$

with inverse given by $\mathbf{B} = \mathbf{\Gamma}\,\mathbf{S}^{1/2}$. The corresponding Jacobian determinant factorizes the volume element as

$$d\mathbf{B} \;=\; c_{r,u}\, |\mathbf{S}|^{\frac{r-u-1}{2}}\, d\mu_V(\mathbf{\Gamma})\, d\mathbf{S}, \tag{8}$$

for a constant $c_{r,u} > 0$, where $d\mu_V$ denotes the Haar measure on the Stiefel manifold $V_{r,u}$.

Combining (7) and (8), the induced target density for $\mathbf{B}$ is given by the compactly supported density

$$\pi(\mathbf{B}) \;\propto\; \pi(\mathbf{\Gamma}(\mathbf{B}))\, \mathbf{1}\{\lambda_{\min}(\mathbf{B}^\top \mathbf{B}) \geq c\}\, \mathbf{1}\{\|\mathbf{B}\|_F^2 \leq M\}, \tag{9}$$

since $\text{tr}(\mathbf{S}) = \|\mathbf{B}\|_F^2$. Thus, sampling $\mathbf{B}$ from (9) and mapping back via $\mathbf{\Gamma}(\mathbf{B})$ yields valid draws from the original target density $\pi(\mathbf{\Gamma})$.

**(iii) Sampling from (9).** We sample from (9) using ReHMC combined with the convex–container–plus–thinning method. The target density $\pi(\mathbf{B})$ is defined on $\mathbb{R}^{r \times u}$ (equivalently, $\mathbb{R}^{ru}$) with support

$$K \;:=\; \Big\{\mathbf{B} \in \mathbb{R}^{r \times u} : \lambda_{\min}(\mathbf{B}^\top \mathbf{B}) \geq c,\; \|\mathbf{B}\|_F^2 \leq M\Big\}.$$

The eigenvalue constraint $\lambda_{\min}(\mathbf{B}^\top \mathbf{B}) \geq c$ renders $K$ non-convex. We therefore embed $K$ into the convex container

$$\widetilde{K} \;:=\; \Big\{\mathbf{B} \in \mathbb{R}^{r \times u} : \|\mathbf{B}\|_F^2 \leq M\Big\},$$

namely the Frobenius ball of radius $\sqrt{M}$, which satisfies Assumption (A1). We run ReHMC on the convex container $\widetilde{K}$, together with the thinning step described in Section 3. Algorithm 3 summarizes the resulting Polar Reflective HMC (PR-HMC) sampler, which combines reflective dynamics on $\widetilde{K}$ with thinning. The choices of algorithmic tuning parameters, including the step size $\eta$, the integration length $b$, and the penalty parameter $\lambda$, are specified in the numerical experiments.

For the mixing-time analysis of this algorithm, our goal is to verify that the reparameterized target satisfies Assumption (A2). It suffices to assume that the original target density $\pi(\mathbf{\Gamma})$ is locally smooth on an open neighborhood of the Stiefel manifold $V_{r,u}$. Under this assumption, we show that the reparameterized target admits uniformly bounded smoothness parameters over its effective domain.

This conclusion follows from the compactness of the support $K$ of the induced density $\pi(\mathbf{B})$ and the smoothness of the mapping $\mathbf{B} \mapsto \mathbf{\Gamma}(\mathbf{B})$. In particular, the support $K$ is compact by construction, and the mapping $\mathbf{B} \mapsto \mathbf{\Gamma}(\mathbf{B})$ admits uniformly bounded derivatives on $K$, due to the eigenvalue lower bound $c > 0$ and the Frobenius norm upper bound $M < \infty$. As a result, local smoothness of $\pi(\mathbf{\Gamma})$ implies global smoothness of the induced density $\pi(\mathbf{B})$ on $K$. By Proposition 3.1, these smoothness bounds extend uniformly to the convex container $\widetilde{K}$. Consequently, Assumption (A2) holds for the ReHMC dynamics on $\widetilde{K}$.

**Algorithm 3** Polar Reflective HMC for Stiefel-Constrained Targets

1: **Input:** current state $B \in \widetilde{K}$, step size $\eta$, leapfrog steps $L_f$, mass matrix $\boldsymbol{M}$, boundary width $b$, penalty strength $\lambda$, spectral threshold $c$, and Frobenius–norm bound $M$
2: **Define:** $\widetilde{U}(B) = U(B) + \lambda \, \psi(\varphi(B)/b)$ on $\widetilde{K}$
3: Draw momentum $P \sim \mathcal{N}(0, \boldsymbol{M})$
4: Set $(B_0, P_0) \leftarrow (B, P)$
5: **for** $\ell = 0, 1, \ldots, L_f - 1$ **do**
6:     $P_{\ell+\frac{1}{2}} \leftarrow P_\ell - \frac{\eta}{2} \nabla \widetilde{U}(B_\ell)$
7:     $(B_{\ell+1}, P_{\ell+\frac{1}{2}}) \leftarrow$
8:         REFLECTIVEPOSITION$(B_\ell, P_{\ell+\frac{1}{2}}, \eta, \boldsymbol{M}, \widetilde{K})$
9:     $P_{\ell+1} \leftarrow P_{\ell+\frac{1}{2}} - \frac{\eta}{2} \nabla \widetilde{U}(B_{\ell+1})$
10: **end for**
11: Set $(B', P') \leftarrow (B_L, P_L)$
12: Accept $B^{\text{new}} \leftarrow B'$ with prob. $e^{-\widetilde{U}(B')} \mathbf{1}\{B' \in \widetilde{K}\})$
13:     otherwise set $B^{\text{new}} \leftarrow B$
14: **if** $B^{\text{new}} \in K$ **then**
15:     Compute $\Gamma \leftarrow \Gamma(B^{\text{new}})$
16: **end if**
17: (Optional) flip momentum: $P' \leftarrow -P'$

## 6. Experiments

We evaluate the computational efficiency and convergence behavior of the proposed Polar Reflective HMC (PR-HMC) by comparing it with the unconstrained polar-expansion HMC method of Jauch et al. (2021), which serves as the baseline algorithm throughout our experiments.

In all experiments, we fix $b = 0.2$, $\lambda = 2$, and $L_f = 50$. For each experimental setting, we perform a grid search over $c \in \{0.01, 0.1\}$ and $M \in \{20, 40\}$, and report results using the best-performing configuration for that setting. The step size and mass matrix are adaptively tuned during the warm-up period following the procedure of (Hoffman et al., 2014). We discard the first 2,000 iterations as burn-in and retain 2,000 iterations for sampling across four parallel chains. For PR-HMC, ESS is computed only from the retained samples after thinning; discarded samples incur additional cost but are excluded from the ESS count, making the comparison conservative. We report the resulting ESS further averaged across chains. Detailed experimental settings and full results are provided in Appendix C. The code is available at https://github.com/swpark0413/prhmc.

As evaluation criteria, we report the effective sample size (ESS) and the Potential Scale Reduction Factor (PSRF, also known as split-$\widehat{R}$) (Gelman & Rubin, 1992; Vehtari et al., 2021). The PSRF diagnoses convergence by comparing the between-chain and within-chain variabilities; values near 1 indicate that the chains have reached stationarity, while values exceeding 1.05 suggest incomplete mixing. We compute

*Table 1.* Computational efficiency comparison on the Stiefel manifold with $r = 40$. Larger ESS is better; for PSRF (split-$\hat{R}$), values closer to 1.0 indicate convergence, with values above 1.05 suggesting insufficient mixing (Vehtari et al., 2021). The best result for each $(u, \omega)$ configuration is shown in **bold**. NA denotes configurations where numerical instability made MCMC infeasible.

| | | BASELINE | | PR-HMC | |
|---|---|---|---|---|---|
| $u$ | $\omega$ | ESS | PSRF | ESS | PSRF |
| 20 | 10 | 626.68 | 1.006 | 640.42 | 1.006 |
| | 100 | 1432.54 | 1.004 | 1343.78 | 1.003 |
| | 200 | 1351.09 | 1.005 | 1379.87 | 1.005 |
| | 500 | 1046.10 | 1.005 | 1054.42 | 1.006 |
| 30 | 10 | 1214.27 | 1.002 | 1292.33 | 1.003 |
| | 100 | 994.89 | 1.003 | 1056.54 | 1.003 |
| | 200 | NA | NA | **1246.13** | 1.003 |
| | 500 | NA | NA | **1219.83** | 1.009 |
| 39 | 10 | 1352.64 | 1.002 | 1475.80 | 1.002 |
| | 100 | 459.30 | 1.013 | **1162.82** | 1.002 |
| | 200 | 579.38 | 1.011 | **1331.58** | 1.002 |
| | 500 | 900.54 | 1.004 | **1522.68** | 1.002 |

the ESS of the entries of the projection matrices $\boldsymbol{\Gamma}\boldsymbol{\Gamma}^\top$, treating each upper-triangular element $P_{ij}$ as a scalar summary of the sampled subspaces, and first average the ESS values across all such elements.

### 6.1. Directional distributions on the Stiefel manifold

We illustrate the proposed method to the sampling of Matrix von Mises–Fisher (vMF) distribution (Hoff, 2009; Pal et al., 2020):

$$\pi(\boldsymbol{\Gamma}) \propto \exp\bigl(\omega \operatorname{tr}(\boldsymbol{A}^\top \boldsymbol{\Gamma})\bigr), \qquad \boldsymbol{\Gamma} \in V_{r,u},$$

where $\boldsymbol{A} \in \mathbb{R}^{r \times u}$ is a fixed parameter matrix and $\omega > 0$ determines the concentration.

We consider $r = \{20, 40\}$, $u = \{2r/4, 3r/4, r-1\}$, and $\omega \in \{10, 100, 200, 500\}$. For brevity, Table 1 reports the results for $r = 40$, while the corresponding results for $r = 20$ are provided in Appendix C.

Table 1 presents that the proposed PR-HMC consistently achieves comparable or superior ESS values across most configurations, with the performance gap becoming more pronounced as the subspace dimension $u$ approaches the ambient dimension $r$. Notably, at $u = 39$, PR-HMC demonstrates substantial improvements with an average ESS gain of 90.3% over the baseline. Furthermore, while the baseline method fails to converge for $u = 30$ with $\omega \in \{200, 500\}$, PR-HMC maintains robust performance, demonstrating superior numerical stability in challenging parameter settings where the baseline method struggles.

## 6.2. PCA with an orthogonal loading subspace

Next, we evaluate performance on a Bayesian PCA model, where inference on the orthogonal loading structure is central. We generate $n = 30$ observations from

$$\boldsymbol{Y} = \boldsymbol{Z}\boldsymbol{R}\boldsymbol{\Gamma}^\top + \boldsymbol{E},$$

where $\boldsymbol{Y} \in \mathbb{R}^{n \times r}$ is the observed data matrix, $\boldsymbol{\Gamma} \in V_{r,u}$ is an orthogonal loading matrix, $\boldsymbol{Z} \in \mathbb{R}^{n \times u}$ consists of latent scores drawn from a standard normal distribution, and $\boldsymbol{E} \in \mathbb{R}^{n \times r}$ is isotropic Gaussian noise. The diagonal matrix $R = \mathrm{diag}(r_1, \ldots, r_u)$ specifies the signal strength of each latent factor, with distinct ordered entries to resolve the rotational non-identifiability. We assign a Matrix Angular Central Gaussian (Kent et al., 2013) prior to $\boldsymbol{\Gamma}$, which is a natural distribution on the Stiefel manifold.

We fix the subspace dimension at $u = 2$ and vary $r \in \{10, 20, 30, 40\}$. The data generation utilizes the scaling matrix $\boldsymbol{R} = \mathrm{diag}(5, 0.5)$ to induce distinct signal strengths, while the MACG prior on $\Gamma$ has density $\propto |\Gamma^\top \Phi^{-1} \Gamma|^{-r/2}$ with $\Phi = \mathrm{diag}(1, \varepsilon)$; $\Phi$ is a prior hyperparameter, and smaller $\varepsilon$ yields a more anisotropic posterior. To account for data-generation variability, we repeat the analysis over 30 independently generated datasets.

Table 2 demonstrates that under moderate anisotropy ($\varepsilon \in \{0.1, 0.05\}$), both methods achieve comparable ESS values across all ambient dimensions $r$. However, under high anisotropy ($\varepsilon = 0.01$), the baseline method fails to converge for all configurations, whereas PR-HMC consistently maintains stable performance with ESS values exceeding 921. These results highlight the robustness of PR-HMC in highly anisotropic settings, where the baseline method encounters numerical difficulties.

## 7. Conclusions

We established non-asymptotic total-variation mixing-time bounds for reflective Hamiltonian Monte Carlo (ReHMC) on bounded domains, with a leading $d^2$ dependence under $L$-smoothness and sharper guarantees under strong convexity. To handle general bounded supports, we introduced a convex-container-plus-thinning framework and demonstrated its effectiveness for sampling on the Stiefel manifold. Beyond the Stiefel manifold, our framework may be applied to sampling over $\mathrm{SU}(N)$ in lattice gauge theory (Duane et al., 1987; Christ et al., 2025) and to high-dimensional polytope sampling in systems biology (Haraldsdóttir et al., 2017); see Appendix D for a polytope demonstration.

The main limitations are twofold. First, the general smooth mixing bound in Theorem 4.3 has exponential dependence on the problem-dependent constants $G, L$, and $R$, which can make the bound conservative. Second, the thinning step requires the retention probability $\alpha = \widetilde{\pi}(K)$ to be bounded

Table 2. Computational efficiency in Bayesian PCA with orthogonal loadings across ambient dimensions $r$ and anisotropy levels $\varepsilon$ (smaller $\varepsilon$ indicates higher anisotropy). Larger ESS is better; for PSRF, values closer to 1.0 indicate convergence. Best per configuration in **bold**. NA denotes configurations where numerical instability made MCMC infeasible.

| $r$ | $\varepsilon$ | BASELINE | | PR-HMC | |
| --- | --- | --- | --- | --- | --- |
| | | ESS | PSRF | ESS | PSRF |
| 10 | 0.1 | 1066.73 | 1.0014 | 1075.97 | 1.014 |
| | 0.05 | 1248.59 | 1.003 | 1258.29 | 1.003 |
| | 0.01 | NA | NA | **1061.04** | 1.007 |
| 20 | 0.1 | 964.89 | 1.046 | 968.74 | 1.043 |
| | 0.05 | 1109.68 | 1.011 | 1128.09 | 1.011 |
| | 0.01 | NA | NA | **989.37** | 1.012 |
| 30 | 0.1 | 995.71 | 1.031 | 1011.71 | 1.044 |
| | 0.05 | 1073.04 | 1.026 | 1067.64 | 1.040 |
| | 0.01 | NA | NA | **977.32** | 1.042 |
| 40 | 0.1 | 1010.57 | 1.081 | 1027.61 | 1.073 |
| | 0.05 | 993.4 | 1.092 | 1022.38 | 1.099 |
| | 0.01 | NA | NA | **921.12** | 1.089 |

away from zero; if $\alpha$ is small, most samples are discarded and the effective sample size deteriorates.

An important direction for future work is to extend this framework to target densities supported on a product of bounded and unbounded parameter spaces, which commonly arise in Bayesian models with both orthogonal and additional unconstrained parameters, such as factor models.

## Acknowledgements

**Kwangmin Lee** was supported by Basic Science Research Program through the National Research Foundation of Korea (NRF) funded by the Korea government (MSIT) (RS-2025-25433229) and Global-Learning & Academic research institution for Master's · PhD students, and Postdocs (LAMP) Program of the National Research Foundation of Korea (NRF) grant funded by the Ministry of Education (RS-2024-00442775). **Yeonhee Park** was supported by the National Research Foundation of Korea (NRF) grant funded by the Korea government (MSIT) (RS-2025-02216235, RS-2026-25473157)

## Impact Statement

This paper develops theory and algorithms for reflective Hamiltonian Monte Carlo on constrained domains, with applications to Bayesian inference and Monte Carlo computation. The work is methodological in nature, and we do not identify specific societal or ethical impacts requiring further discussion beyond those of general-purpose statistical computation.

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

## Organization of the Appendix

Appendix A provides details of the smooth-extension construction for general bounded supports, including the penalty-based extension and the softplus approximation used in implementation. Appendix B contains the proofs of the mixing-time results. Appendix C reports additional experimental details and full numerical results. Appendix D illustrates the proposed framework on polytope-constrained targets, and Appendix E provides an empirical verification of the predicted $O(d^2)$ dimension scaling. Appendix F compares the bounded reflecting-container extension with an unbounded penalty-only extension, thereby illustrating why the extension should retain a reflecting boundary rather than relying solely on the penalty term.

## A. Details of the Smooth Extension for General Bounded Supports

This appendix provides a concrete construction underlying Section 3. The purpose is to extend reflective HMC to a general bounded support by embedding it into a smooth convex container and applying thinning. Throughout, we separate the theoretical construction, which uses an exactly vanishing smooth boundary surrogate, from the softplus approximation used in numerical implementations.

### A.1. Container

Let $\widetilde{K} \subset \mathbb{R}^d$ be a compact, convex set with $C^2$ boundary. The $C^2$ boundary regularity ensures a well-defined inward normal at every boundary point and smooth dependence of reflections on the state. We assume throughout that

$$K \subset \widetilde{K}.$$

The specific choice of $\widetilde{K}$ is not unique and may be adapted to the geometry of $K$. In practice, $\widetilde{K}$ is chosen to be a simple geometric set that strictly contains $K$, such as a Euclidean ball or a hyperrectangle, with sufficient margin to accommodate a smooth transition near $\partial K$.

**Examples.**

- **Bounding ball.** If $K$ is bounded, one may take

$$\widetilde{K} = \{q \in \mathbb{R}^d : \|q - q_0\| \leq R\},$$

  where $q_0$ is a fixed center and $R$ is chosen so that $K \subset \widetilde{K}$. This choice yields a $C^\infty$ boundary and simplifies reflective dynamics.

- **Ellipsoidal container.** When $K$ exhibits anisotropic scaling, one may take

$$\widetilde{K} = \{q : (q - q_0)^\top \Sigma^{-1} (q - q_0) \leq R^2\},$$

  for some positive definite matrix $\Sigma$. This choice aligns the container with the principal axes of $K$ and can improve numerical stability.

### A.2. Construction of the extended potential

We now construct a smooth extension of the potential that enforces the support constraint through a smooth penalty, without introducing any explicit projection or retraction toward $K$. The basic idea is to preserve the original potential on $K$ while adding a smooth penalty outside $K$ within the container $\widetilde{K}$. Under the exact-vanishing surrogate condition introduced below, the resulting extended potential is globally $C^2$ on $\widetilde{K}$ and agrees with the original potential on $K$ up to second order.

#### A.2.1. BOUNDARY DESCRIPTION

Let $K \subset \mathbb{R}^d$ be bounded. Assume that there exists an open neighborhood $N$ of $\partial K$ and a boundary defining function $\phi : N \to \mathbb{R}$ such that

$$K \cap N = \{q \in N : \phi(q) \leq 0\}, \qquad \partial K = \{q \in N : \phi(q) = 0\}, \qquad \nabla\phi(q) \neq 0 \quad (q \in \partial K),$$

with $\phi \in C^2(N)$. The sign of $\phi$ encodes feasibility near the boundary, so $\phi$ can be viewed as a signed-distance-like constraint function.

For nonsmooth supports, such a regular defining function may not be available globally. The container construction avoids direct reflection on $\partial K$; reflection is instead performed on $\partial \widetilde{K}$. The original constraint $K$ is handled through a smooth penalty term, introduced below via a boundary-violation surrogate $\varphi$.

### A.2.2. EXACT-VANISHING BOUNDARY SURROGATE

Assume that there exists a function

$$\varphi : \widetilde{K} \to [0, \infty)$$

such that

$$\varphi \in C^2(\widetilde{K}), \qquad \varphi(q) = 0 \quad \text{for all } q \in K, \qquad \varphi(q) > 0 \quad \text{for } q \in \widetilde{K} \setminus K.$$

We call $\varphi$ an exact-vanishing boundary-violation surrogate. The role of $\varphi$ is to measure the amount of violation outside $K$ while remaining exactly zero on the original support.

When a $C^2$ defining function $\phi$ is available, a typical construction is

$$\varphi(q) = h(\phi(q)),$$

where $h : \mathbb{R} \to [0, \infty)$ is a $C^2$ smooth hinge satisfying

$$h(t) = 0 \quad (t \leq 0), \qquad h(t) > 0 \quad (t > 0).$$

For example, one may take $h(t) = 0$ for $t \leq 0$, $h(t) = t$ for $t \geq \varepsilon$, and connect the two pieces on $(0, \varepsilon)$ by a $C^2$ smoothstep interpolation.

### A.2.3. PENALTY FUNCTION

Let $\psi : \mathbb{R} \to \mathbb{R}$ be a penalty function satisfying

$$\psi \in C^2(\mathbb{R}), \qquad \psi(s) = 0 \quad \text{for } s \leq 0, \qquad \psi(0) = \psi'(0) = \psi''(0) = 0.$$

In applications, $\psi$ is chosen to be nonnegative and increasing on $[0, \infty)$. A typical choice is the one-sided polynomial penalty

$$\psi(s) = (s_+)^{2k}, \qquad k \geq 2,$$

where $s_+ = \max\{s, 0\}$. Since $\varphi(q) = 0$ on $K$, these conditions imply that the added penalty term has zero value, zero gradient, and zero Hessian on $K$.

### A.2.4. PENALTY-BASED EXTENDED POTENTIAL

Let $U : \widetilde{K} \to \mathbb{R}$ denote a $C^2$ extension of the original potential to the container $\widetilde{K}$. Fix parameters $\lambda > 0$ and $b > 0$, and define

$$\widetilde{U}(q) := U(q) + \lambda \, \psi\left(\frac{\varphi(q)}{b}\right), \qquad q \in \widetilde{K}. \tag{10}$$

Here $b$ controls the thickness of the boundary layer, and $\lambda$ controls the strength of the repulsive penalty outside $K$.

In this construction,

- for $q \in K$, the surrogate satisfies $\varphi(q) = 0$, so the penalty term vanishes exactly;

- for $q \in \widetilde{K} \setminus K$, the penalty increases smoothly with the magnitude of the constraint violation.

Thus the extended potential agrees with the original potential on $K$ and adds a smooth penalty outside $K$ within the container.

**Lemma A.1** (Regularity and interior matching). *Suppose that $U \in C^2(\widetilde{K})$ and that $\varphi \in C^2(\widetilde{K})$ satisfies*

$$\varphi(q) = 0, \qquad q \in K.$$

*Let $\psi \in C^2(\mathbb{R})$ satisfy*

$$\psi(0) = \psi'(0) = \psi''(0) = 0.$$

*Then the penalty-based extended potential*

$$\widetilde{U}(q) = U(q) + \lambda \, \psi\left(\frac{\varphi(q)}{b}\right)$$

*belongs to $C^2(\widetilde{K})$ and satisfies*

$$\widetilde{U}(q) = U(q), \qquad \nabla\widetilde{U}(q) = \nabla U(q), \qquad \nabla^2\widetilde{U}(q) = \nabla^2 U(q), \quad \forall q \in K.$$

*Proof.* Define

$$s(q) := \frac{\varphi(q)}{b}, \qquad S(q) := \psi(s(q)).$$

Since $\varphi \in C^2(\widetilde{K})$ and $\psi \in C^2(\mathbb{R})$, the composition $S = \psi \circ s$ belongs to $C^2(\widetilde{K})$. Therefore,

$$\widetilde{U}(q) = U(q) + \lambda S(q)$$

belongs to $C^2(\widetilde{K})$.

Fix $q \in K$. By the exact-vanishing property of $\varphi$, we have $\varphi(q) = 0$ and hence $s(q) = 0$. Therefore,

$$\widetilde{U}(q) = U(q) + \lambda\psi(0) = U(q).$$

Next, by the chain rule,

$$\nabla\widetilde{U}(q) = \nabla U(q) + \lambda \, \psi'(s(q)) \, \nabla s(q) = \nabla U(q) + \frac{\lambda}{b}\psi'(s(q)) \, \nabla\varphi(q).$$

Since $s(q) = 0$ and $\psi'(0) = 0$ for $q \in K$, the additional term vanishes. Thus

$$\nabla\widetilde{U}(q) = \nabla U(q), \qquad q \in K.$$

Differentiating once more gives

$$\nabla^2\widetilde{U}(q) = \nabla^2 U(q) + \lambda \left\{ \psi''(s(q)) \, \nabla s(q)\nabla s(q)^\top + \psi'(s(q)) \, \nabla^2 s(q) \right\}.$$

Since

$$\nabla s(q) = \frac{1}{b}\nabla\varphi(q), \qquad \nabla^2 s(q) = \frac{1}{b}\nabla^2\varphi(q),$$

we obtain

$$\nabla^2\widetilde{U}(q) = \nabla^2 U(q) + \frac{\lambda}{b^2}\psi''(s(q))\nabla\varphi(q)\nabla\varphi(q)^\top + \frac{\lambda}{b}\psi'(s(q))\nabla^2\varphi(q).$$

For $q \in K$, $s(q) = 0$. The flat endpoint conditions $\psi'(0) = \psi''(0) = 0$ imply that both additional terms vanish. Therefore,

$$\nabla^2\widetilde{U}(q) = \nabla^2 U(q), \qquad q \in K.$$

$\square$

*Remark* A.2 (Existence of the exact-vanishing surrogate). Lemma A.1 assumes the existence of a $C^2$ exact-vanishing surrogate $\varphi$. This assumption should be viewed as a regularity condition on the support description used for the smooth extension. When $K$ admits a $C^2$ defining function $\phi$ in a neighborhood of its boundary, such a surrogate can be built by composing $\phi$ with a $C^2$ one-sided smoothing $h$ satisfying $h(t) = 0$ for $t \leq 0$. For supports described by nonsmooth constraints, one should first construct or choose a $C^2$ surrogate $\varphi$ that vanishes exactly on $K$ and is positive outside $K$. The subsequent results depend only on these properties of $\varphi$, not on the particular representation of the original constraints.

## A.3. Approximation error for the softplus surrogate

The exact interior matching in Lemma A.1 relies on the condition $\varphi(q) = 0$ for all $q \in K$. A smooth alternative is the softplus surrogate

$$\varphi_\sigma(q) = \frac{1}{\sigma} \log\{1 + \exp(\sigma \phi(q))\}.$$

Although $\varphi_\sigma$ is $C^\infty$, it does not vanish exactly on $\{\phi \leq 0\}$. Thus, unlike the exact-vanishing surrogate used above, the softplus surrogate does not give exact interior matching on $K$. In particular, the identities

$$\widetilde{U} = U, \qquad \nabla \widetilde{U} = \nabla U, \qquad \nabla^2 \widetilde{U} = \nabla^2 U \quad \text{on } K$$

need not hold exactly for the softplus-based extension. Nevertheless, the resulting discrepancy can be controlled uniformly on $K$.

Assume the sign convention

$$K = \{q : \phi(q) \leq 0\}$$

and consider the quartic penalty $\psi(s) = s^4$. The corresponding softplus-based extended potential is

$$\widetilde{U}_\sigma(q) = U(q) + \lambda \left( \frac{\varphi_\sigma(q)}{b} \right)^4.$$

For any $q \in K$, we have $\phi(q) \leq 0$. Since $\sigma > 0$, this implies $\exp(\sigma \phi(q)) \leq 1$, and therefore

$$1 + \exp(\sigma \phi(q)) \leq 2.$$

Hence

$$0 \leq \varphi_\sigma(q) = \frac{1}{\sigma} \log\{1 + \exp(\sigma \phi(q))\} \leq \frac{\log 2}{\sigma}.$$

It follows that, for all $q \in K$,

$$0 \leq \widetilde{U}_\sigma(q) - U(q) = \lambda \left( \frac{\varphi_\sigma(q)}{b} \right)^4 \leq \lambda \left( \frac{\log 2}{\sigma b} \right)^4.$$

Consequently,

$$\sup_{q \in K} |\widetilde{U}_\sigma(q) - U(q)| \leq \lambda \left( \frac{\log 2}{\sigma b} \right)^4 = O(\sigma^{-4}).$$

Therefore, the softplus construction does not give exact interior matching, but its error on $K$ can be made uniformly small by increasing $\sigma$.

*Remark* A.3. The bound above is stated for the quartic penalty $\psi(s) = s^4$. More generally, if $\psi(s) = s^{2k}$ for $k \geq 2$, then the same argument gives

$$\sup_{q \in K} |\widetilde{U}_\sigma(q) - U(q)| \leq \lambda \left( \frac{\log 2}{\sigma b} \right)^{2k} = O(\sigma^{-2k}).$$

## A.4. Additional smoothness and global boundedness

The results of the previous subsection establish that the penalty-based extension is well defined under $C^2$ regularity and, under the exact-vanishing condition on the surrogate $\varphi$ together with the flat-endpoint conditions on the penalty function $\psi$, satisfies exact interior matching on $K$. In this subsection, we impose additional global smoothness assumptions and derive uniform boundedness properties for the extended potential.

These bounds are not required for the validity of the construction itself, but play a central role in quantitative mixing analyses such as Theorem 4.3. In particular, global bounds on the Hessian of $\widetilde{U}$ provide step-size restrictions and stability terms in mixing-time estimates.

**Assumption A.4** (Global boundedness framework for the smooth penalty)**.** Assume that $\widetilde{K} \subset \mathbb{R}^d$ is compact and that $U \in C^2(\widetilde{K})$ with $\nabla U$ being $L$-Lipschitz on $\widetilde{K}$. Assume further that $\varphi \in C^2(\widetilde{K})$ is a fixed boundary-violation surrogate. Let $b > 0$ and define the penalty coordinate

$$s(q) := \frac{\varphi(q)}{b}, \qquad q \in \widetilde{K}.$$

Assume $\psi \in C^2(\mathbb{R})$ satisfies the one-sided condition

$$\psi(s) = 0, \qquad s \leq 0.$$

Define the attainable range of $s$ on the compact container by

$$s_{\min} := \inf_{q \in \widetilde{K}} s(q), \qquad s_{\max} := \sup_{q \in \widetilde{K}} s(q),$$

and define the penalty constants on this attainable range by

$$m_j := \sup_{t \in [s_{\min}, s_{\max}]} |\psi^{(j)}(t)|, \qquad j = 0, 1, 2. \tag{11}$$

*Remark* A.5. The restriction in (11) to the attainable range is essential. For typical choices such as the one-sided polynomial penalty

$$\psi(t) = (t_+)^4, \qquad t_+ = \max\{t, 0\},$$

the global suprema $\sup_{t \in \mathbb{R}} |\psi^{(j)}(t)|$ may be infinite, whereas the attainable-range quantities in (11) are finite because $[s_{\min}, s_{\max}]$ is bounded whenever $s$ is bounded on the compact set $\widetilde{K}$.

**Lemma A.6** (Finiteness of penalty constants). *Under Assumption A.4, the constants used in the global curvature analysis are finite. In particular,*

$$s_{\min}, \ s_{\max} \in \mathbb{R}, \qquad B_1 := \sup_{q \in \widetilde{K}} \|\nabla s(q)\| < \infty, \qquad B_2 := \sup_{q \in \widetilde{K}} \|\nabla^2 s(q)\|_2 < \infty,$$

*and the penalty constants defined on the attainable range satisfy*

$$m_j := \sup_{t \in [s_{\min}, s_{\max}]} |\psi^{(j)}(t)| < \infty, \qquad j = 0, 1, 2.$$

*Proof.* Since $\widetilde{K}$ is compact and $U \in C^2(\widetilde{K})$, all derivatives of $U$ up to order two are continuous on $\widetilde{K}$ and hence bounded. Next, since $\varphi \in C^2(\widetilde{K})$ and $\widetilde{K}$ is compact, $\varphi$ and its first two derivatives are continuous and bounded on $\widetilde{K}$. Hence

$$M_0 := \sup_{q \in \widetilde{K}} |\varphi(q)| < \infty, \qquad M_1 := \sup_{q \in \widetilde{K}} \|\nabla \varphi(q)\| < \infty, \qquad M_2 := \sup_{q \in \widetilde{K}} \|\nabla^2 \varphi(q)\|_2 < \infty.$$

By definition, $s(q) = b^{-1} \varphi(q)$. Therefore $s$ is continuous on $\widetilde{K}$ and hence attains its infimum and supremum on $\widetilde{K}$. This proves

$$s_{\min}, s_{\max} \in \mathbb{R}.$$

Moreover,

$$\nabla s(q) = \frac{1}{b} \nabla \varphi(q), \qquad \nabla^2 s(q) = \frac{1}{b} \nabla^2 \varphi(q).$$

Thus

$$B_1 = \sup_{q \in \widetilde{K}} \|\nabla s(q)\| \leq \frac{M_1}{b} < \infty,$$

and

$$B_2 = \sup_{q \in \widetilde{K}} \|\nabla^2 s(q)\|_2 \leq \frac{M_2}{b} < \infty.$$

Finally, since $s(\widetilde{K}) \subset [s_{\min}, s_{\max}]$ with finite endpoints and $\psi \in C^2(\mathbb{R})$, each $\psi^{(j)}$ is continuous and therefore bounded on $[s_{\min}, s_{\max}]$. Hence

$$m_j < \infty, \qquad j = 0, 1, 2.$$

$\square$

*Proof of Proposition 3.1.* Differentiating (10) yields

$$\nabla \widetilde{U}(q) = \nabla U(q) + \lambda \, \psi'(s(q)) \, \nabla s(q),$$

and differentiating once more gives

$$\nabla^2 \widetilde{U}(q) = \nabla^2 U(q) + \lambda \Big( \psi''(s(q)) \, \nabla s(q) \nabla s(q)^\top + \psi'(s(q)) \, \nabla^2 s(q) \Big). \tag{12}$$

Since $\nabla U$ is $L$-Lipschitz on $\widetilde{K}$, we have $\|\nabla^2 U(q)\|_2 \leq L$ for all $q \in \widetilde{K}$. Taking operator norms in (12) and using the definitions of $B_1, B_2$ and $m_1, m_2$ yields, for all $q \in \widetilde{K}$,

$$\|\nabla^2 \widetilde{U}(q)\|_2 \leq \|\nabla^2 U(q)\|_2 + \lambda \Big( |\psi''(s(q))| \, \|\nabla s(q)\|^2 + |\psi'(s(q))| \, \|\nabla^2 s(q)\|_2 \Big) \leq L + \lambda \big( m_2 B_1^2 + m_1 B_2 \big).$$

The right-hand side is finite under Assumption A.4 and Lemma A.6. Defining

$$C_\star := L + \lambda(m_2 B_1^2 + m_1 B_2)$$

completes the proof. $\qquad \square$

*Remark A.7.* When $\lambda = 0$, the curvature bound reduces to $\|\nabla^2 \widetilde{U}\|_2 \leq L$, so the bound is inherited entirely from the base potential $U$. Moreover, if $U$ is $L$-smooth on $\widetilde{K}$, then for any reference point $x_\star \in \widetilde{K}$,

$$G := \sup_{q \in \widetilde{K}} \|\nabla U(q)\| \leq \|\nabla U(x_\star)\| + L \operatorname{diam}(\widetilde{K}).$$

This yields an alternative parameterization of constants in mixing bounds in terms of $(L, \|\nabla U(x_\star)\|)$ and the container diameter.

# B. Proof of Theorems

In this section, we prove Theorems 4.3 and 4.5.

*Remark B.1.* (General mass matrix). The proofs below assume $M = I$ for notational convenience. The extension to general $M \succ 0$ follows by replacing the Euclidean inner product $\langle \cdot, \cdot \rangle$ with the $M$-inner product $\langle x, y \rangle_M = x^\top M^{-1} y$ and the Euclidean norm with $\| \cdot \|_M$. Provided that $0 < \lambda_{\min}(M) \leq \lambda_{\max}(M) < \infty$, this substitution introduces only constant factors depending on the condition number $\lambda_{\max}(M)/\lambda_{\min}(M)$ into the bounds of Theorems 4.3 and 4.5, without altering the proof structure.

For this purpose, we invoke Lemma 12 of Lee et al. (2020), restated as Lemma B.2. This reduces the proofs of Theorems 4.3 and 4.5 to identifying $k_{\min}$ such that (13) holds. We then establish conditions under which ReHMC satisfies (13) and obtain an explicit $k_{\min}$.

**Lemma B.2** (Lemma 12 of Lee et al. (2020)). *Let $\mathcal{T}$ be a Markov kernel with stationary distribution $\pi^*$. For $k \geq 1$, define the Cesàro average $\bar{\mathcal{T}}_k := k^{-1} \sum_{i=0}^{k-1} \mathcal{T}^i$. Suppose there exists an integer $k_{\min} \geq 0$ such that for every distribution $\pi$ that is $\beta/\varepsilon$-warm with respect to $\pi^*$,*

$$\big\| \bar{\mathcal{T}}_{k_{\min}} \pi - \pi^* \big\|_{\mathrm{TV}} \leq \frac{1}{2e}. \tag{13}$$

*Then, if $\pi_0$ is $\beta$-warm with respect to $\pi^*$, for any $k \geq k_{\min} \log(\varepsilon^{-1})$,*

$$\big\| \bar{\mathcal{T}}_k \pi_0 - \pi^* \big\|_{\mathrm{TV}} \leq \varepsilon.$$

To identify $k_{\min}$ satisfying (13), we employ the blocking–conductance inequality Kannan et al. (2006); Lee et al. (2020), restated in Lemma B.3.

**Lemma B.3** (Blocking–conductance mixing bound). *Suppose the same setting of Lemma B.2. Let*

$$Q(S) := \int_S \mathcal{T}(x, S^c) \, d\pi^*(x)$$

*denote the usual conductance profile of $\mathcal{T}$. Suppose $\pi_0$ is $\beta$-warm, and there exist $c > 0$ and a decreasing function $\phi : [c, 1/4] \to \mathbb{R}_+$ such that*

$$\frac{\pi^*(S)}{Q(S)^2} \leq \phi(t) \qquad \text{for all } S \text{ with } \pi^*(S) = t \in [c, 1/2],$$

*and $\phi(t) \leq M$ for $t \in [1/4, 1/2]$. Then, for all $k \geq 1$,*

$$\|\bar{\mathcal{T}}_k \pi_0 - \pi^*\|_{\mathrm{TV}} \leq \beta c + \frac{32}{k} \left( \int_c^{1/4} \phi(x)\, dx + \frac{M}{4} \right). \tag{14}$$

We next characterize the blocking profile $\phi(t)$ needed to control the right-hand side of (14) under two regimes given in Theorems 4.3 and 4.5.

### B.1. Proof of Theorem 4.3

**Lemma B.4.** *Suppose $\pi^*(x) \propto e^{-U(x)}$ defined on $K \subseteq \mathbb{R}^d$, and let $\rho(\pi^*)$ denote a log–Sobolev constant such that*

$$\mathrm{Ent}_{\pi^*}(f^2) \leq \rho(\pi^*) \int_{\mathbb{R}^d} \|\nabla f(x)\|^2\, d\pi^*(x),$$

*for any smooth function $f$, where*

$$\mathrm{Ent}_{\pi^*}(f^2) := \int_{\mathbb{R}^d} f(x)^2 \log\left( \frac{f(x)^2}{\int_{\mathbb{R}^d} f(y)^2\, d\pi^*(y)} \right) d\pi^*(x).$$

*Let $\Omega \subseteq K$ satisfy $\pi^*(\Omega) = 1 - s$, and suppose that for some $\delta > 0$ and $\alpha \in (0, 1]$,*

$$\|\mathcal{T}_x - \mathcal{T}_y\|_{\mathrm{TV}} \leq 1 - \alpha \quad \text{for all } x, y \in \Omega \text{ with } \|x - y\| \leq \delta, \tag{15}$$

*where $\mathcal{T}_x(\cdot) := \mathcal{T}(x, \cdot)$ is the transition kernel. Then, for $t \in [s, 1/2]$ and any $S$ with $\pi^*(S) = t$,*

$$Q(S) \geq \frac{\alpha}{4} \left[ \frac{\rho(\pi^*)\delta}{4} (t - s)(1 - t - s) - s \right].$$

*In particular, if*

$$s \leq \min\left\{ \frac{\rho(\pi^*)\delta}{32} t(1 - t),\ \frac{t}{2},\ \frac{1 - t}{2} \right\},$$

*then*

$$Q(S) \geq \frac{\alpha}{4} C t(1 - t), \qquad C = \min\left\{ \frac{\rho(\pi^*)\delta}{32},\ \frac{1}{2} \right\}.$$

**Lemma B.5.** *Suppose the same setting of Lemma B.4 and (A1)-(A2) hold. Let $K \subset \mathbb{R}^d$ be a bounded convex set with $\mathrm{diam}(K) \leq 2R$, and let $U \in C^1(K)$ have an $L$-Lipschitz gradient on $K$. Define*

$$\pi^*(x) \propto e^{-U(x)} \mathbf{1}_K(x).$$

*There exists a universal constant $c_{\mathrm{LV}} > 0$ such that, for every anchor $x_\star \in K$,*

$$\rho(\pi^*) \geq \frac{c_{\mathrm{LV}}}{2R} \exp\left( -8R\|\nabla U(x_\star)\| - (8L + 2)R^2 \right) \geq \frac{c_{\mathrm{LV}}}{2R} \exp\left( -8RG - (8L + 2)R^2 \right).$$

**Lemma B.6** (Local TV contraction)**.** *Suppose Assumptions (A1)-(A2) hold. If*

$$\eta \leq C_0 \left( \frac{\rho}{\sqrt{d} + G} \ \wedge\ \frac{1}{G\sqrt{d}} \ \wedge\ \frac{1}{\sqrt{L d}} \ \wedge\ \frac{1}{\sqrt{L} G} \ \wedge\ \frac{1}{d L_n(\rho)} \ \wedge\ \frac{1}{G^2 L_n(\rho)} \right)$$

*for some constant $C_0$, then*

$$\sup_{\substack{x,y\in\Omega \\ \|x-y\|\le C_1\eta}} \left\|\mathcal{T}_x - \mathcal{T}_y\right\|_{\mathrm{TV}} \le 1-\alpha, \qquad \alpha = \tfrac{1}{8},$$

*for some positive constant $C_1$.*

The proofs of Lemmas B.4-B.5 and Lemma B.6 are provided in Sections B.3 and B.4, respectively.

*Proof of Theorem 4.3.* Applying Lemma B.4 to the blocking–conductance bound (Lemma B.3), when $t \le 1/2$ and $\Omega = K$ ($s = 0$), we can set

$$\phi(t) \;=\; \frac{64}{\alpha^2 C^2 t}, \qquad C = \min\left\{\frac{\rho(\pi^*)\delta}{32}, \frac{1}{2}\right\}, \quad c = \frac{\varepsilon}{4e\beta}, \quad M = \phi(1/4) = \frac{256}{\alpha^2 C^2}, \quad \delta = \rho_U \eta,$$

Note that the condition of $\eta$ comes from Lemma B.6, which is needed to satisfy (15) in Lemma B.4. For any $\beta/\varepsilon$-warm initial distribution $\pi$, Lemma B.3 gives

$$\|\bar{\mathcal{T}}_k\pi - \pi^*\|_{\mathrm{TV}} \le \frac{1}{4e} + \frac{32}{k}\left[\frac{64}{\alpha^2 C^2}\big(2 + \log\beta - \log\varepsilon\big)\right].$$

Hence, when

$$k \;\gtrsim\; \frac{1}{\alpha^2}\,\max\left\{\frac{256}{\rho(\pi^*)^2\,\delta^2}, 1\right\}\log\left(\frac{\beta}{\varepsilon}\right),$$

we obtain $\|\rho_k - \pi^*\|_{\mathrm{TV}} \le \frac{1}{2e}$. By applying Lemmas B.2 and B.5 with $\delta \asymp \eta$, we obtain $\left\|\bar{\mathcal{T}}_k\pi_0 - \pi^*\right\|_{\mathrm{TV}} \le \varepsilon$, when

$$k \gtrsim \left[\frac{R}{\eta^2}\exp(8RG + (8L+2)R^2) + 1\right]\log\left(\frac{\beta}{\varepsilon}\right)\log\left(\frac{1}{\varepsilon}\right),$$

$$\eta \lesssim \frac{\rho}{\sqrt{d}+G} \;\wedge\; \frac{1}{G\sqrt{d}} \;\wedge\; \frac{1}{\sqrt{L\,d}} \;\wedge\; \frac{1}{\sqrt{L}\,G} \;\wedge\; \frac{1}{d\,L_n(\rho)} \;\wedge\; \frac{1}{G^2 L_n(\rho)}.$$

$\square$

## B.2. Proof of Theorem 4.5

**Lemma B.7** (K-constrained version of Lemma 28 of Lee et al. (2020))**.** *Let $K \subset \mathbb{R}^d$ be convex and let $\pi_K^*$ be an $m$-strongly log-concave distribution supported on $K$, i.e.*

$$\pi_K^*(x) \;\propto\; \exp\big(-U(x)\big)\mathbf{1}_K(x),$$

*with $m$-strongly convex $U(x)$. Suppose there exists $\Omega \subset K$ with $\pi_K^*(\Omega) = 1 - s$ and that for all $x, y \in \Omega$ with $\|x - y\| \le \delta$,*

$$\|T_x - T_y\|_{\mathrm{TV}} \;\le\; 1-\alpha,$$

*and $\delta\sqrt{m} < 1$ is satisfied. Then, for all $s \le t \le 1/2$ and all measurable $S \subset K$ with $\pi_K^*(S) = t$,*

$$\frac{\pi_K^*(S)}{Q_K(S)^2} \;\le\; \frac{16t}{\alpha^2\left(\frac{\delta\sqrt{m}}{4}(t-s)\log^{1/2}\big[(1+1/t)\big] - s\right)^2},$$

*where $Q_K(S)$ is the conductance of $S$ for the Markov kernel $T$ with stationary distribution $\pi_K^*$, which is defined in Lemma B.3. In particular, if*

$$s \;\le\; \min\left(\tfrac{t}{2}, \tfrac{\delta\sqrt{m}\,t}{16}\sqrt{\log 3}\right),$$

*then*

$$\frac{\pi_K^*(S)}{Q_K(S)^2} \;\le\; \frac{2^{16}}{\alpha^2\,\delta^2\,m\,t\log(1/t)}.$$

*Proof.* First, we provide the inequality that is similar to Lemma 27 in Lee et al. (2020). While we consider the $K$-restricted distribution, Lemma 27 in Lee et al. (2020) does not.

Let $x^\star$ be the unique minimizer of $U$. Since $U$ is $m$-strongly convex, we can write

$$U(x) = g(x) + \frac{m}{2}\|x - x^\star\|^2,$$

where $g$ is convex. Because $K$ is convex, the indicator $\mathbf{1}_K$ is log-concave. Hence

$$\tilde{q}(x) := \exp(-g(x))\,\mathbf{1}_K(x)$$

is log-concave, and after normalization $q(x) := \tilde{q}(x)/Z_q$ is a log-concave density. Let $\gamma$ denote the density of the Gaussian distribution $\mathcal{N}(x^\star, m^{-1}I_d)$. Then $\pi_K^*$ can be written, up to a constant, as

$$\pi_K^*(x) = q(x)\,\gamma(x).$$

We now apply Lemma 16 in Chen et al. (2020) to the density $\pi_K^* = q \cdot \gamma$. Identifying $\sigma^2 = 1/m$, Lemma 16 yields the logarithmic isoperimetric inequality for any partition $(S_1, S_2, S_3)$ of $K$:

$$\pi_K^*(S_3) \geq \frac{\sqrt{m}}{2}\,d(S_1, S_2)\,\min\{\pi_K^*(S_1), \pi_K^*(S_2)\}\log\left[\frac{1}{2}\left(1 + \frac{1}{\min\{\pi_K^*(S_1), \pi_K^*(S_2)\}}\right)\right].$$

This is precisely the analogue of Lemma 27 in Lee et al. (2020) for the $K$-restricted distribution. Applying the above $K$-restricted log-isoperimetric inequality to the partition $(A_1, A_2, A_3)$ and following the same steps in the proof of Lemma 28 in Lee et al. (2020), we obtain the lower bound on $Q_K(S)$. This yields the stated bound on $\pi_K^*(S)/Q_K(S)^2$. $\qquad\square$

*Proof of Theorem 4.5.* *Step 1: Local TV contraction.* Under Assumptions (A1), (A2), and (A3), Lemma B.6 shows that, provided $\eta$ satisfies

$$\eta \lesssim \frac{\rho}{\sqrt{d} + G} \wedge \frac{1}{G\sqrt{d}} \wedge \frac{1}{\sqrt{L\,d}} \wedge \frac{1}{\sqrt{L}\,G} \wedge \frac{1}{d\,L_n(\rho)} \wedge \frac{1}{G^2 L_n(\rho)},$$

there exist constants $\alpha \in (0, 1]$ and $\delta > 0$ such that

$$\|\mathcal{T}_x - \mathcal{T}_y\|_{\mathrm{TV}} \leq 1 - \alpha \quad \text{whenever } x, y \in K, \ \|x - y\| \leq \delta,$$

with $\alpha$ universal (e.g. $\alpha = 1/8$) and $\delta \asymp \eta$. The additional constraint $\delta\sqrt{m} < 1$ appearing in Lemma B.7 then yields the extra requirement $\eta \lesssim m^{-1/2}$. Altogether this gives the step–size condition (4) (after rescaling the universal constant $c_0$).

*Step 2: Conductance profile under strong log–concavity.* Since $U$ is $m$–strongly convex on $K$, the truncated target $\pi^*(x) \propto e^{-U(x)}\mathbf{1}_K(x)$ is $m$–strongly log–concave on the convex set $K$. Applying Lemma B.7 (with $s = 0$ and the same $\alpha, \delta$ as in Step 1), we obtain a conductance lower bound of the form

$$\frac{\pi^*(S)}{Q(S)^2} \leq \phi(t) \quad \text{whenever } \pi^*(S) = t \in (0, 1/2],$$

with

$$\phi(t) \asymp \frac{1}{\alpha^2\,\delta^2\,m\,t\log(1/t)}.$$

This coincides, up to universal constants, with the conductance profile in Lee et al. (2020, Eq. (36)).

*Step 3: Blocking–conductance bound.* We plug this conductance profile into the blocking–conductance inequality stated in Lemma B.3. Take $c = \varepsilon/(4e\beta)$ and the $\phi$ from Step 2. Then, we obtain

$$k_{\min} \asymp \frac{1}{m\,\delta^2}\left[\log\log(\beta/\varepsilon) + 1\right]. \tag{16}$$

Since $\delta \asymp \eta$ by Lemma B.6, we obtain

$$k_{\min} \; \asymp \; \frac{1}{m\,\eta^2}\left[\log\log(\beta/\varepsilon) + 1\right],$$

such that for every $\beta/\varepsilon$–warm initial distribution $\pi$,

$$\left\|\bar{\mathcal{T}}_{k_{\min}}\pi - \pi^*\right\|_{\text{TV}} \; \leq \; \frac{1}{2e}.$$

Finally, by Lemma B.2, we obtain

$$\left\|\bar{\mathcal{T}}_{k}\pi_0 - \pi^*\right\|_{\text{TV}} \; \leq \; \varepsilon.$$

for

$$k \; \gtrsim \; \frac{1}{m\,\eta^2}\,\log\!\Big(\log(\beta/\varepsilon) + 1\Big)\,\log\!\Big(\tfrac{1}{\varepsilon}\Big),$$

which is (5).

$\square$

## B.3. Proofs of Lemmas B.4-B.5

*Proof of Lemma B.4.* Fix $S$ with $\pi^*(S) = t \in [s, 1/2]$ and we have $Q(S) := \int_S \mathcal{T}_x(S^c)\,d\pi^*(x) = \int_{S^c} \mathcal{T}_x(S)\,d\pi^*(x)$ by the stationarity of $\pi^*$, where $S^c = K \setminus S$.

Define

$$A_1 := \{x \in S \cap \Omega : \mathcal{T}_x(S^c) < \alpha/2\}, \quad A_2 := \{x \in S^c \cap \Omega : \mathcal{T}_x(S) < \alpha/2\}, \quad A_3 := (A_1 \cup A_2)^c.$$

We consider three cases: (1) $\pi^*(A_1) < \frac{1}{2}\pi^*(S \cap \Omega)$, (2) $\pi^*(A_2) < \frac{1}{2}\pi^*(S^c \cap \Omega)$, and (3) $\pi^*(A_1) \geq \frac{1}{2}\pi^*(S \cap \Omega)$ & $\pi^*(A_2) \geq \frac{1}{2}\pi^*(S^c \cap \Omega)$.

When the first case holds, $\pi^*(A_1) < \frac{1}{2}\pi^*(S \cap \Omega)$, we have

$$\begin{aligned}
Q(S) &= \int_S \mathcal{T}_x(S^c)d\pi^*(x) \\
&\geq \int_{(S\cap\Omega)\setminus A_1} \mathcal{T}_x(S^c)\,d\pi^*(x) \\
&\geq \frac{\alpha}{2}\pi^*((S \cap \Omega) \setminus A_1) \\
&\geq \frac{\alpha}{4}\pi^*(S \cap \Omega) \\
&\geq \frac{\alpha}{4}(t - s),
\end{aligned}$$

where the last inequality is satisfied since $\pi^*(S \cap \Omega) \geq \pi^*(S) - \pi^*(\Omega^c) = t - s$.

When the second case holds, $\pi^*(A_2) < \frac{1}{2}\pi^*(S^c \cap \Omega)$, we have

$$\begin{aligned}
Q(S) &= \int_{S^c} \mathcal{T}_x(S)d\pi^*(x) \\
&\geq \int_{(S^c\cap\Omega)\setminus A_2} \mathcal{T}_x(S)\,d\pi^*(x) \\
&\geq \frac{\alpha}{2}\pi^*((S^c \cap \Omega) \setminus A_2) \\
&\geq \frac{\alpha}{4}\pi^*(S^c \cap \Omega) \\
&\geq \frac{\alpha}{4}(1 - t - s),
\end{aligned}$$

where the last inequality is satisfied since $\pi^*(S^c \cap \Omega) \geq \pi^*(S^c) - \pi^*(\Omega^c) = 1 - t - s$.

Next, we consider the third case, i.e.,

$$\pi^*(A_1) \geq \tfrac{1}{2}(t - s), \qquad \pi^*(A_2) \geq \tfrac{1}{2}(1 - t - s).$$

By Lemma 10 in Ma et al. (2019), we have

$$\pi^*(A_3) \geq \rho_U \, d(A_1, A_2) \, \pi^*(A_1) \, \pi^*(A_2) \geq \rho_U \, d(A_1, A_2) \cdot \frac{(t - s)(1 - t - s)}{4}, \tag{17}$$

where $d(A_1, A_2) = \inf\{\|x - y\| : x \in A_1, y \in A_2\}$.

If $x \in A_1$ and $y \in A_2$, then $\|\mathcal{T}_x - \mathcal{T}_y\|_{\mathrm{TV}} \geq \mathcal{T}_x(S) - \mathcal{T}_y(S) = 1 - \mathcal{T}_x(S^c) - \mathcal{T}_y(S) > 1 - \alpha$. By (15), we obtain $\|x - y\| > \delta$, which gives $d(A_1, A_2) \geq \delta$.

Restricting to $\Omega$ yields

$$\pi^*(A_3 \cap \Omega) \geq \pi^*(A_3) - \pi^*(\Omega^c) \geq \frac{\rho_U \delta}{4}(t - s)(1 - t - s) - s. \tag{18}$$

We have

$$\begin{aligned} Q(S) &= \tfrac{1}{2}\left(\int_S \mathcal{T}_x(S^c)\, d\pi^*(x) + \int_{S^c} \mathcal{T}_x(S)\, d\pi^*(x)\right) \\ &\geq \tfrac{1}{2}\left(\int_{S \cap \Omega \cap A_3} \mathcal{T}_x(S^c)\, d\pi^*(x) + \int_{S^c \cap \Omega \cap A_3} \mathcal{T}_x(S)\, d\pi^*(x)\right) \\ &\geq \frac{\alpha}{4}\{\pi^*(S \cap A_3 \cap \Omega) + \pi^*(S^c \cap A_3 \cap \Omega)\} \\ &= \frac{\alpha}{4}\pi^*(A_3 \cap \Omega). \end{aligned}$$

Combine with (18) to obtain

$$Q(S) \geq \frac{\alpha}{4}\left[\frac{\rho_U \delta}{4}(t - s)(1 - t - s) - s\right].$$

Collecting all the cases, we obtain

$$Q(S) \geq \frac{\alpha}{4}\min\left\{\frac{\rho_U \delta}{4}(t - s)(1 - t - s) - s, t - s, 1 - t - s\right\}.$$

From the bound obtained above, we may simplify it under mild assumptions on $s$. If $s \leq \min\{\frac{t}{2}, \frac{1-t}{2}\}$, then $(t - s)(1 - t - s) \geq \frac{1}{4}t(1 - t)$. Moreover, if $s \leq \min\left\{\frac{\rho_U \delta}{32}t(1 - t), \frac{t}{2}, \frac{1-t}{2}\right\}$, we obtain

$$\frac{\rho_U \delta}{4}(t - s)(1 - t - s) - s \geq \frac{\rho_U \delta}{16}t(1 - t) - s \geq \frac{\rho_U \delta}{32}t(1 - t).$$

Hence, in this regime, we obtain $Q(S) \geq \frac{\alpha}{4}C\,t(1 - t)$, where $C = \min\{\frac{\rho_U \delta}{32}, \frac{1}{2}\}$. $\qquad\square$

*Proof of Lemma B.5.* Let $\nu$ denote the density function of the truncated Gaussian distribution of $N(\bar{x}, \boldsymbol{I})I_K$ with $\bar{x} \in K$ and $\sup_{x \in K}\|x - \bar{x}\| \leq R$. By Theorem 1.8 of Lee & Vempala (2018), the log-Sobolev constant of $\nu$ satisfies

$$\rho(\nu) \geq \frac{c_{\mathrm{LV}}}{\mathrm{diam}(K)} \geq \frac{c_{\mathrm{LV}}}{2R},$$

for a universal constant $c_{\mathrm{LV}} > 0$.

On $K$, we have

$$\frac{\pi^*(x)}{\nu(x)} \propto \exp\left(-U(x) + \tfrac{1}{2}\|x - \bar{x}\|^2\right) =: e^{\psi(x)}.$$

By the Holley–Stroock theorem Holley & Stroock (1987),

$$\rho(\pi^*) \geq e^{-\mathrm{osc}_K(\|x-\bar{x}\|^2/2-U(x))} \rho(\nu)$$
$$\geq e^{-\mathrm{osc}_K(\|x-\bar{x}\|^2/2)-\mathrm{osc}_K(U(x))} \rho(\nu),$$

where $\mathrm{osc}_K(f(x)) := \sup_{x \in K} f(x) - \inf_{x \in K} f(x)$ for a function $f$. Since

$$\mathrm{osc}_K\left(\tfrac{1}{2}\|x-\bar{x}\|^2\right) := \sup_{x \in K} \tfrac{1}{2}\|x-\bar{x}\|^2 - \inf_{x \in K} \tfrac{1}{2}\|x-\bar{x}\|^2 \leq R^2/2.$$

we obtain $\rho(\Pi) \geq \frac{c_{LV}}{2R} e^{-\mathrm{osc}_K(U(x))-R^2/2}$.

Next, we provide the upper bound of $\mathrm{osc}_K(U(x))$. By the assumption that $\nabla U$ is $L$-Lipschitz on $K$, we have

$$U(x) = U(x_\star) + \langle \nabla U(x_\star), x - x_\star \rangle + r(x), \qquad |r(x)| \leq \tfrac{L}{2}\|x - x_\star\|^2,$$

for any anchor $x_\star \in K$.

Taking $\sup$ and $\inf$ over $x \in K$ and subtracting,

$$\mathrm{osc}_K(U) \leq w_K(\nabla U(x_\star)) + \frac{L}{2}\,\mathrm{diam}(K)^2,$$

where $w_K(g) := \sup_{x \in K}\langle g, x \rangle - \inf_{x \in K}\langle g, x \rangle \leq \|g\|\,\mathrm{diam}(K)$. Then, we obtain

$$\mathrm{osc}_K(U) \leq \|\nabla U(x_\star)\|\mathrm{diam}(K) + \frac{L}{2}\,\mathrm{diam}(K)^2 \tag{19}$$

Therefore,

$$\rho(\Pi) \geq \frac{c_{\mathrm{LV}}}{2R} \exp\left(-2R\|\nabla U(x_\star)\| - \left(2L + \frac{1}{2}\right)R^2\right).$$

This yields the claimed anchored lower bound. $\qquad\square$

## B.4. Proof of Lemma B.6

### B.4.1. LEMMAS CONCERNING REFLECTION GEOMETRY

**Lemma B.8.** *Assume (A1). Then:*

 (i) *The nearest-point projection* $\mathrm{proj} : \mathcal{N}_\rho \to \partial K$ *is well-defined (single-valued) and of class $C^1$.*

 (ii) *Let $d_K(x)$ denote the signed distance between $x$ and $\partial K$. We set the sign as positive (negative) when $x \in K^c$ ($x \in K$). The $d_K$ belongs to $C^2(\mathcal{N}_\rho)$ and*

$$\nabla d_K(x) = n(\mathrm{proj}(x)) \qquad \text{for all } x \in \mathcal{N}_\rho.$$

 (iii) *Writing $x = \Phi(p, s)$ with $p = \mathrm{proj}(x)$ and $s = d_K(x)$, the Hessian is*

$$D^2 d_K(x) = D(n \circ \mathrm{proj})(x) = -S_{p(x)}(I - s(x) S_{p(x)})^{-1} \Pi_{T_{p(x)}} = -(I - s(x) S_{p(x)})^{-1} S_{p(x)} \Pi_{T_{p(x)}},$$

*where $S_p$ is the shape operator at $p \in \partial K$, i.e. $S_p = -Dn(p)\big|_{T_p(\partial K)}$, and $\Pi_{T_p}$ is the orthogonal projection onto $T_p(\partial K)$. In particular,*

$$\|D^2 d_K(x)\| \leq \frac{\bar{\kappa}}{1 - \bar{\kappa}\,|s(x)|} \leq \frac{\bar{\kappa}}{1 - \bar{\kappa}\rho}.$$

*Proof. Step 1 (Tubular coordinates and $C^1$ regularity).* By Assumption (A1), the normal map $\Phi$ is a $C^1$ diffeomorphism. Hence every $x \in \mathcal{N}_\rho$ has a unique representation

$$x = \Phi(p(x), s(x)) = p(x) + s(x)\, n(p(x)), \qquad (p(x), s(x)) \in \partial K \times (-\rho, \rho),$$

where $n\big(p(x)\big)$ denotes the outward unit normal to $\partial K$ at $p(x)$.

*Step 2.* We show that $p(x)$ becomes the nearest point among $\partial K$, i.e., $\mathrm{proj}(x) := p(x)$. Fix $x \in \mathcal{N}_\rho$ and write $x = p(x) + s(x)\, n(p(x))$. If there is no confusion, we let $p$ denote $p(x)$. For any $y \in \partial K$,

$$\|x - y\|^2 = \|p - y\|^2 + s^2 + 2s\,\langle n(p),\, p - y\rangle.$$

Convexity implies $\langle n(p),\, y - p\rangle \le 0$ for all $y \in \partial K$; in particular $\langle n(p),\, p - y\rangle \ge 0$ for $y \in \partial K$. Thus $\|x - y\|^2 \ge s^2$, with equality iff $y = p$. Therefore $p$ is the unique nearest boundary point and $\mathrm{dist}(x, \partial K) = |s|$. Thus, $\mathrm{proj}(x) = p(x)$ and $d_K(x) = s(x)$, and both $\mathrm{proj}$ and $d_K$ are $C^1$ because $\Phi^{-1}$ is $C^1$.

*Step 3 (First derivatives: explicit formulas for $Dp$ and $ds$).* Differentiate the identity $x = \Phi(p(x), s(x)) = p(x) + s(x)\, n(p(x))$ in the direction $h \in \mathbb{R}^d$. Then, since $Dn(p)[v] = -S_p v$ for $v \in T_p(\partial K)$ (Weingarten equation (Weingarten, 1861)), we obtain

$$h = (I - s\, S_p)\, v + \tau\, n(p),$$

where $(v, \tau) := (Dp[h],\, ds[h]) \in T_p(\partial K) \times \mathbb{R}$.

Taking the normal component gives $\langle n(p), h\rangle = \tau$, i.e. $ds[h] = \langle n(p), h\rangle$. Projecting tangentially yields $\Pi_{T_p} h = (I - s\, S_p)\, v$, hence

$$Dp[h] = (I - s\, S_p)^{-1}\, \Pi_{T_p} h, \qquad ds[h] = \langle n(p), h\rangle.$$

*Step 4 (Gradient identity and $C^2$ regularity).* By definition of the gradient, $\langle \nabla d_K(x), h\rangle = ds[h] = \langle n(p), h\rangle$ for all $h$, hence $\nabla d_K(x) = n(\mathrm{proj}(x))$. Because $n$ is $C^1$ on $\partial K$ and $\mathrm{proj} \in C^1$ on $\mathcal{N}_\rho$, the composition $\nabla d_K = n \circ \mathrm{proj}$ is $C^1$, so $d_K \in C^2$.

*Step 5 (Hessian formula and bound).* Differentiate $\nabla d_K(x) = n(\mathrm{proj}(x))$:

$$D^2 d_K(x)[h] = D(n \circ \mathrm{proj})(x)[h] = Dn(p)\big[Dp[h]\big] = -S_p\, (I - s\, S_p)^{-1}\, \Pi_T h.$$

Taking operator norms and using $\|S_p\| \le \bar\kappa$,

$$\|D^2 d_K(x)\| \le \|S_p\|\, \|(I - sS_p)^{-1}\| \le \frac{\bar\kappa}{1 - \bar\kappa|s|} \le \frac{\bar\kappa}{1 - \bar\kappa\rho}.$$

$\square$

**Corollary B.9.** *Suppose the same setting of Lemma B.8. Define $L_n(\rho) := \dfrac{\bar\kappa}{1 - \bar\kappa\rho}$. Then for any segment $t \mapsto x + t\delta$ contained in $\mathcal{N}_\rho$ and any $s, t$,*

$$\big\| n(\mathrm{proj}(x + t\delta)) - n(\mathrm{proj}(x + s\delta)) \big\| \ \le\ L_n(\rho)\, \|\delta\|\, |t - s|.$$

*In particular, if the straight line segment $[x, y] \subset \mathcal{N}_\rho$, then $\|n(\mathrm{proj}(x)) - n(\mathrm{proj}(y))\| \le L_n(\rho)\, \|x - y\|$.*

*Proof.* Let $F := n \circ \mathrm{proj}$. By Lemma B.8(iii), $\|DF(z)\| = \|D^2 d_K(z)\| \le L_n(\rho)$ for all $z \in \mathcal{N}_\rho$. Along any segment $t \mapsto x + t\delta \subset \mathcal{N}_\rho$,

$$F(x + t\delta) - F(x + s\delta) = \int_s^t DF(x + \tau\delta)[\delta]\, d\tau,$$

hence $\|F(x + t\delta) - F(x + s\delta)\| \le \int_s^t \|DF\|\, \|\delta\|\, d\tau \le L_n(\rho)\|\delta\|\, |t - s|$.

$\square$

**Lemma B.10.** *Suppose the same setting of Lemma B.8. Let $R(n) := I - 2nn^\top$. Then, $\|R(n_1) - R(n_2)\| \le 4\|n_1 - n_2\|$ for any unit vectors $n_1, n_2$. In particular, if the straight line segment $[p, q]$ is contained in $\mathcal{N}_\rho$, then*

$$\|R(n(p)) - R(n(q))\| \ \le\ 4\, L_n(\rho)\, \|p - q\|.$$

*Proof.* Expand

$$R(n_1) - R(n_2) = -2(n_1 n_1^\top - n_2 n_2^\top) = -2\big[n_1(n_1 - n_2)^\top + (n_1 - n_2)n_2^\top\big].$$

Taking operator norms and using $\|n_i\| = 1$ yields $\|R(n_1) - R(n_2)\| \le 2(\|n_1\| + \|n_2\|)\|n_1 - n_2\| = 4\|n_1 - n_2\|$. The second bound follows from Corollary B.9. $\square$

B.4.2. DECOMPOSITION OF TOTAL VARIATION DISTANCE

First, we give the detail of the transition kernel $\mathcal{T}_x$ of the reflective Hamiltonian Monte Carlo algorithm.

Let $\eta > 0$ be the step size and assume identity mass. Ignoring reflections for the moment, the leapfrog position proposal distribution is defined as

$$\tilde{\mathcal{P}}_x = \mathcal{N}\big(m(x),\, \eta^2 I_d\big), \qquad m(x) := x - \tfrac{\eta^2}{2}\nabla U(x).$$

Equivalently, for $V \sim \mathcal{N}(0, I_d)$, we have

$$\tilde{Y}_x(V) := x + \eta V - \tfrac{\eta^2}{2}\nabla U(x) \sim \tilde{\mathcal{P}}_x.$$

Next, we define the proposal distribution considering the reflection. Define $\mathsf{Ref}_K(x, z)$ with $x \in K$ and $z \in \mathbb{R}^d$ as below: we follow the straight segment from $x$ to $z$ and applying specular reflections at each transversal hit with $\partial K$ (as in Algorithm 2). Then, we let $\mathsf{Ref}_K(x, z)$ denote the endpoint after reflections. Then, we define the random variable and its law:

$$Y_x(V) := \mathsf{Ref}_K\big(x,\, \tilde{Y}_x(V)\big), \qquad \mathcal{P}_x := \mathrm{Law}\big(Y_x(V)\big) \quad \text{(pre–MH reflective proposal)}.$$

The one-step reflective Metropolized kernel $\mathcal{T}$ on positions is then

$$\mathcal{T}_x(A) \;=\; \int a(x, z)\,\mathbf{1}_A(z)\,\mathcal{P}_x(dz) \;+\; \Big(1 - \int a(x, z)\,\mathcal{P}_x(dz)\Big)\mathbf{1}_A(x),$$

where $a(x, z) \in [0, 1]$ is the usual MH acceptance probability induced by the leapfrog discretization.

**Events.** Let $z \in K$, $H_z := \{N_z \geq 1\}$, $M_z := \{N_z \geq 2\}$, $S_z := \{N_z = 1\}$, where $N_z$ is the number of hits. Let $r(z) := \mathrm{dist}(z, \partial K)$. For $\theta > 0$ and first-hit point $p_z$ (when $H_z$ occurs),

$$\mathsf{NG}_z(\theta) := \big\{\, |\langle \delta(z, V), n(p_z)\rangle| \geq \theta\eta \,\big\},$$

where $\delta(z, V) = \eta V - \tfrac{\eta^2}{2}\nabla U(z)$.

Using the above inequalities, we prove Lemma B.6.

*Proof of Lemma B.6.* We decompose

$$\big\|\mathcal{T}_x - \mathcal{T}_y\big\|_{\mathrm{TV}} \;\leq\; \big\|\mathcal{P}_x - \mathcal{P}_y\big\|_{\mathrm{TV}} \;+\; \big\|\mathcal{P}_x - \mathcal{T}_x\big\|_{\mathrm{TV}} + \big\|\mathcal{T}_y - \mathcal{P}_y\big\|_{\mathrm{TV}}.$$

Through Sections B.4.3-B.4.6, we provide conditions to satisfy

$$\sup_{\{x,y\in\Omega:\, \|x-y\|\leq\delta\}} \big\|\mathcal{P}_x - \mathcal{P}_y\big\|_{\mathrm{TV}} \;\leq\; \tfrac{5}{8}, \tag{20}$$

when

$$\eta \;\leq\; c_0\left(\frac{\rho}{\sqrt{d}+G} \;\wedge\; \frac{1}{d\,L_n(\rho)} \;\wedge\; \frac{1}{G\sqrt{d}\,L_n(\rho)} \;\wedge\; \frac{1}{\sqrt{L}}\right), \qquad L_n(\rho) = \frac{\bar{\kappa}}{1-\bar{\kappa}\rho}, \tag{21}$$

for a universal constant $c_0 > 0$, where $L_n(\rho) = \frac{\bar{\kappa}}{1-\bar{\kappa}\rho}$.

In Section B.4.7, we provide conditions to satisfy

$$\sup_{\{x,y\in\Omega:\, \|x-y\|\leq\delta\}} \big\|\mathcal{P}_x - \mathcal{T}_x\big\|_{\mathrm{TV}} \;\leq\; \tfrac{1}{8}, \tag{22}$$

when $\eta \;\lesssim\; \frac{1}{G\sqrt{d}} \;\wedge\; \frac{1}{\sqrt{L}\,d} \;\wedge\; \frac{1}{\sqrt{L}\,G} \;\wedge\; \frac{1}{d\,L_n(\rho)} \;\wedge\; \frac{1}{G^2 L_n(\rho)}.$

Therefore, if we choose $\eta$ so that it satisfies *both* sets of conditions above, that is,

$$\eta \;\leq\; C_0\left(\frac{\rho}{\sqrt{d}+G} \;\wedge\; \frac{1}{G\sqrt{d}} \;\wedge\; \frac{1}{\sqrt{L}\,d} \;\wedge\; \frac{1}{\sqrt{L}\,G} \;\wedge\; \frac{1}{d\,L_n(\rho)} \;\wedge\; \frac{1}{G^2 L_n(\rho)}\right)$$

for a sufficiently small universal constant $C_0 > 0$, then for all $x, y \in \Omega$ with $\|x - y\| \le \delta$,

$$\left\|\mathcal{P}_x - \mathcal{P}_y\right\|_{\mathrm{TV}} \le \tfrac{5}{8}, \qquad \left\|\mathcal{P}_x - \mathcal{T}_x\right\|_{\mathrm{TV}} \le \tfrac{1}{8}, \qquad \left\|\mathcal{T}_y - \mathcal{P}_y\right\|_{\mathrm{TV}} \le \tfrac{1}{8}.$$

Plugging these bounds into the decomposition yields

$$\left\|\mathcal{T}_x - \mathcal{T}_y\right\|_{\mathrm{TV}} \le \tfrac{5}{8} + \tfrac{1}{8} + \tfrac{1}{8} = \tfrac{7}{8} = 1 - \tfrac{1}{8}.$$

Taking the supremum over all $x, y \in \Omega$ with $\|x - y\| \le \delta$ gives

$$\sup_{\substack{x,y \in \Omega \\ \|x-y\| \le \delta}} \left\|\mathcal{T}_x - \mathcal{T}_y\right\|_{\mathrm{TV}} \le 1 - \alpha, \qquad \alpha = \tfrac{1}{8},$$

which proves Lemma B.6. $\qquad\qquad\qquad\qquad\qquad\qquad\qquad\qquad\qquad\qquad\qquad\qquad\qquad\qquad\qquad$ $\square$

### B.4.3. UPPER BOUND OF $\left\|\mathcal{P}_x - \mathcal{P}_y\right\|_{\mathrm{TV}}$

We provide Lemma B.11 to give the upper bound of $\left\|\mathcal{P}_x - \mathcal{P}_y\right\|_{\mathrm{TV}}$, which is the same as $\left\|\mathrm{Law}(Y_x) - \mathrm{Law}(Y_y)\right\|_{\mathrm{TV}}$.

**Lemma B.11.** *Let* $\tilde{Y}_x = m(x) + \eta V$, $\tilde{Y}_y = m(y) + \eta V$ *with a shared* $V \sim \mathcal{N}(0, I_d)$, *and let* $Y_x, Y_y$ *be the reflective endpoints constructed from* $\tilde{Y}_x, \tilde{Y}_y$. *Define* $A_x := \{N_x = 0\}$, $A_y := \{N_y = 0\}$ *and* $A_0 := A_x \cap A_y$, *where* $N_x$ ($N_y$) *is the number of reflections of* $Y_x$ ($Y_y$). *Then,*

$$\left\|\mathrm{Law}(Y_x) - \mathrm{Law}(Y_y)\right\|_{\mathrm{TV}} \le \left\|\mathrm{Law}(\tilde{Y}_x) - \mathrm{Law}(\tilde{Y}_y)\right\|_{\mathrm{TV}} + \mathbb{P}(A_0^{\mathrm{c}})$$

$$= \left\|\mathrm{Law}(\tilde{Y}_x) - \mathrm{Law}(\tilde{Y}_y)\right\|_{\mathrm{TV}} + \frac{\mathbb{P}(H_x) + \mathbb{P}(H_y) + \mathbb{P}(H_x \triangle H_y)}{2}. \tag{23}$$

*Proof.* Fix a Borel $B \subset K$. Decompose

$$\mathbb{P}(Y_x \in B) = \mathbb{P}(Y_x \in B, A_0) + \mathbb{P}(Y_x \in B, A_0^{\mathrm{c}}).$$

On $A_0$ there is no reflection for either path, hence $Y_x = \tilde{Y}_x$ and

$$\mathbb{P}(Y_x \in B, A_0) = \mathbb{P}(\tilde{Y}_x \in B, A_0) = \mathbb{P}(\tilde{Y}_x \in B) - \mathbb{P}(\tilde{Y}_x \in B, A_0^{\mathrm{c}})$$

Therefore,

$$\mathbb{P}(Y_x \in B) = \mathbb{P}(\tilde{Y}_x \in B) + \Delta_x, \qquad \Delta_x := \mathbb{P}(Y_x \in B, A_0^{\mathrm{c}}) - \mathbb{P}(\tilde{Y}_x \in B, A_0^{\mathrm{c}}).$$

We now simplify $\Delta_x$. Using $A_0^{\mathrm{c}} = A_x^{\mathrm{c}} \sqcup (A_y^{\mathrm{c}} \cap A_x)$,

$$\mathbb{P}(\tilde{Y}_x \in B, A_0^{\mathrm{c}}) = \mathbb{P}(\tilde{Y}_x \in B, A_x^{\mathrm{c}}) + \mathbb{P}(\tilde{Y}_x \in B, A_y^{\mathrm{c}}, A_x)$$
$$\mathbb{P}(Y_x \in B, A_0^{\mathrm{c}}) = \mathbb{P}(Y_x \in B, A_x^{\mathrm{c}}) + \mathbb{P}(Y_x \in B, A_y^{\mathrm{c}}, A_x)$$

Since $B \subset K$ and a reflection of the straight segment from $m(x)$ to $\tilde{Y}_x$ implies the raw endpoint $\tilde{Y}_x \notin K$, we have $\mathbb{P}(\tilde{Y}_x \in B, A_x^{\mathrm{c}}) = 0$. Moreover on $A_x$ we have $Y_x = \tilde{Y}_x$, hence

$$\mathbb{P}(Y_x \in B, A_y^{\mathrm{c}}, A_x) - \mathbb{P}(\tilde{Y}_x \in B, A_y^{\mathrm{c}}, A_x) = 0.$$

It follows that $\Delta_x = \mathbb{P}(Y_x \in B, A_x^{\mathrm{c}})$. A symmetric argument yields $\mathbb{P}(Y_y \in B) = \mathbb{P}(\tilde{Y}_y \in B) + \Delta_y$ with $\Delta_y = \mathbb{P}(Y_y \in B, A_y^{\mathrm{c}})$.

Thus

$$\left|\mathbb{P}(Y_x \in B) - \mathbb{P}(Y_y \in B)\right| \le \left|\mathbb{P}(\tilde{Y}_x \in B) - \mathbb{P}(\tilde{Y}_y \in B)\right| + \left|\Delta_x - \Delta_y\right|.$$

Since $0 \le \Delta_x \le \mathbb{P}(A_x^{\mathrm{c}}) \le \mathbb{P}(A_0^{\mathrm{c}})$ and $0 \le \Delta_y \le \mathbb{P}(A_y^{\mathrm{c}})$, we have

$$\left|\Delta_x - \Delta_y\right| \le \max\{\mathbb{P}(A_x^{\mathrm{c}}), \mathbb{P}(A_y^{\mathrm{c}})\} \le \mathbb{P}(A_x^{\mathrm{c}} \cup A_y^{\mathrm{c}}) = \mathbb{P}(A_0^{\mathrm{c}}).$$

Combining, we obtain

$$\left|\mathbb{P}(Y_x \in B) - \mathbb{P}(Y_y \in B)\right| \leq \left|\mathbb{P}(\tilde{Y}_x \in B) - \mathbb{P}(\tilde{Y}_y \in B)\right| + \mathbb{P}(A_0^c),$$

for a Borel $B \subseteq K$. Taking the supremum over $B \subset K$ then implies the total variation bound (the supremum over a sub-class of sets is no larger than the supremum over all Borel sets).

Since

$$\mathbb{1}_{H_x \cup H_y} = \frac{\mathbb{1}_{H_x} + \mathbb{1}_{H_y} + |\mathbb{1}_{H_x} - \mathbb{1}_{H_y}|}{2}.$$

Taking expectations yields

$$\mathbb{P}(A_0^c) = \mathbb{P}(H_x \cup H_y) = \frac{\mathbb{P}(H_x) + \mathbb{P}(H_y) + \mathbb{P}(H_x \triangle H_y)}{2},$$

where $\mathbb{P}(H_x \triangle H_y) = \mathbb{E}|\mathbb{1}_{H_x} - \mathbb{1}_{H_y}|$. □

We provide Lemma B.12 to give the upper bound of the first term in (23).

**Lemma B.12.** *Let* $\tilde{\mathcal{P}}(x, \cdot) = \mathcal{N}(m(x), \eta^2 I_d)$, *where* $m(x) = x - \frac{\eta^2}{2}\nabla U(x)$, *and suppose that* $\nabla U$ *is L-Lipschitz. Then*

$$\|\tilde{\mathcal{P}}(x, \cdot) - \tilde{\mathcal{P}}(y, \cdot)\|_{\mathrm{TV}} \leq \frac{1}{2\eta}\|m(x) - m(y)\| \leq \frac{1}{2\eta}\left(1 + \frac{\eta^2 L}{2}\right)\|x - y\|.$$

*Proof.* Pinsker's inequality gives $\|\mu - \nu\|_{\mathrm{TV}} \leq \sqrt{\frac{1}{2}\mathrm{KL}(\mu\|\nu)}$. For Gaussians with common covariance $\Sigma = \eta^2 I_d$,

$$\mathrm{KL}\big(\mathcal{N}(\mu, \Sigma)\big\|\mathcal{N}(\nu, \Sigma)\big) = \tfrac{1}{2}(\mu - \nu)^\top \Sigma^{-1}(\mu - \nu) = \frac{\|\mu - \nu\|^2}{2\eta^2}.$$

Thus $\|\cdot\|_{\mathrm{TV}} \leq \frac{1}{2\eta}\|\mu - \nu\|$ and

$$\|m(x) - m(y)\| \leq \|x - y\| + \frac{\eta^2}{2}\|\nabla U(x) - \nabla U(y)\| \leq \left(1 + \frac{\eta^2 L}{2}\right)\|x - y\|.$$

□

The upper bounds of $\mathbb{P}(H_x) + \mathbb{P}(H_y)$ and $\mathbb{P}(H_x \triangle H_y)$ that appear in (23) are provided in Sections B.4.4 and B.4.5.

B.4.4. UPPER BOUND OF $\mathbb{P}(H_x)$

We give the upper bound $\mathbb{P}(H_x)$ in Lemma B.15, the proof of which requires the following lemmas.

**Lemma B.13** (Momentum cutoff). *If* $V \sim \mathcal{N}(0, I_d)$ *and* $R_d := \sqrt{d} + \sqrt{2u}$, *then* $\mathbb{P}(\|V\| > R_d) \leq e^{-u}$.

*Proof.* We use the standard concentration for the Euclidean norm of a standard Gaussian: $\|V\|$ is 1-Lipschitz in $V$ and $\mathbb{E}\|V\| \leq \sqrt{d}$. Gaussian concentration yields $\mathbb{P}(\|V\| - \mathbb{E}\|V\| > t) \leq e^{-t^2/2}$. Taking $t = \sqrt{2u}$ and using $\mathbb{E}\|V\| \leq \sqrt{d}$ gives the claim. □

**Lemma B.14.** *Suppose (A1)-(A2) are satisfied and fix* $x \in K$. *Let* $V \sim \mathcal{N}(0, I_d)$ *and* $\delta_x := \eta V - \frac{\eta^2}{2}\nabla U(x)$. *Let* $\mathcal{E} := \{\|V\| \leq R_d\}$, *where* $R_d = \sqrt{d} + \sqrt{2u}$, *and let* $H_x$ *denote the event that the first hit occurs. Suppose* $\eta R_d + \frac{\eta^2}{2}G \leq \rho$. *On* $H_x \cap \mathcal{E}$, *the segment* $x + t\delta_x$ *stays in* $\mathcal{N}_\rho$ *for all* $t \in [0, 1]$, *and for* $g(t) := d_K(x + t\delta_x)$ *we have*

$$g(t) \leq d_K(x) + t\langle n(\mathrm{proj}(x)), \delta_x \rangle + \frac{1}{2}L_n(\rho)\|\delta_x\|^2 t^2 \qquad (0 \leq t \leq 1),$$

*where* $L_n(\rho) := \dfrac{\bar{\kappa}}{1 - \bar{\kappa}\rho}$.

*Proof. (Tube containment on $H_x \cap \mathcal{E}$).* On $H_x$ the first hit occurs at some $t_x \in (0, 1]$ with $p_x = x + t_x \delta_x$. For $t \in [0, 1]$, we have

$$\mathrm{dist}(x + t\delta_x, \partial K) \; \leq \; \|(t - t_x)\delta_x\| = |t_x - t|\|\delta_x\| \; \leq \; \|\delta_x\|.$$

On $\mathcal{E}$ we have $\|\delta_x\| \leq \eta\|V\| + \frac{\eta^2}{2}G \leq \eta R_d + \frac{\eta^2}{2}G \leq \rho$ by the condition of this lemma. Thus, the whole segment $\{x + t\delta_x : 0 \leq t \leq 1\}$ lies in the tube $\mathcal{N}_\rho$.

By Corollary B.9, $n(\mathrm{proj}(\cdot))$ is well-defined, and by Lemma B.8,

$$g'(t) = \langle n(\mathrm{proj}(x + t\delta_x)), \, \delta_x \rangle.$$

By Corollary B.9, we have

$$\left| g'(s) - g'(0) \right| = \left| \langle n(\mathrm{proj}(x + s\delta_x)) - n(\mathrm{proj}(x)), \, \delta_x \rangle \right| \leq L_n(\rho) \, \|\delta_x\|^2 \, s.$$

Integrating from $0$ to $t$,

$$g(t) \; \leq \; g(0) + t \, g'(0) + \frac{1}{2} \, L_n(\rho) \, \|\delta_x\|^2 \, t^2,$$

which is the desired bound. $\qquad\square$

**Lemma B.15.** *[Upper bound $\mathbb{P}(H_x)$] Suppose the same settings of Lemma B.14. Draw $V \sim \mathcal{N}(0, I_d)$ and set*

$$\delta_x := \eta V - \frac{\eta^2}{2}\nabla U(x), \qquad R_d := \sqrt{d} + \sqrt{2u} \; (u > 0), \qquad B_d := \eta R_d + \frac{\eta^2}{2}G.$$

*Then,*

$$\mathbb{P}(H_x) \; \leq \; \frac{1}{2} \; + \; \frac{(\frac{1}{2}\,L_n(\rho)\,B_d^2 \; + \; \frac{1}{2}\eta^2\,G)_+}{\eta\sqrt{2\pi}} \; + \; e^{-u}.$$

*Proof.* In this proof, we let $n(x)$ denote $n(\mathrm{proj}(x))$. Let $\mathcal{E} := \{\|V\| \leq R_d\}$. On $\mathcal{E}$, Lemma B.14 gives $g(t) \leq q_+(t) := d_K(x) + b\,t + \frac{1}{2}\alpha t^2$ for $t \in [0, 1]$, where $b := \langle n(x), \delta_x \rangle$ and $\alpha := L_n(\rho)\|\delta_x\|^2$. Since $q_+$ is convex, $\max_{t \in [0,1]} q_+(t) = \max\{q_+(0), q_+(1)\}$. Because $q_+(0) = d_K(x) < 0$, the event $\{g(t) \geq 0$ for some $t\}$ is contained in $\{q_+(1) \geq 0\} = \{b \geq -d_K(x) - \frac{1}{2}\alpha\}$. Since $g(t) \geq 0$ for some $t$ when $H_x$ occurs, we obtain

$$\mathbb{P}(H_x) \; \leq \; \mathbb{P}\big(b \geq -d_K(x) - \tfrac{1}{2}\alpha\big) \; + \; \mathbb{P}(\mathcal{E}^c) \; \leq \; \mathbb{P}(Z \geq a) \; + \; e^{-u},$$

where $Z \sim \mathcal{N}(0, 1)$ and

$$a := \frac{-d_K(x) - \frac{1}{2}L_n(\rho)B_d^2 - \frac{\eta^2}{2}\langle \nabla U(x), n(x) \rangle}{\eta} \geq -\frac{\frac{1}{2}\,L_n(\rho)\,B_d^2 \; + \; \frac{1}{2}\eta^2\,G}{\eta}$$

Then, we obtain

$$\mathbb{P}(Z \geq a) \leq \frac{1}{2} + \frac{(-a)_+}{\sqrt{2\pi}}.$$

$\qquad\square$

### B.4.5. UPPER BOUND OF $\mathbb{P}(H_x \triangle H_y)$

**Lemma B.16.** *Suppose the same settings of Lemma B.14. Fix $\theta > 0$ and $u > 0$, and write $R_d := \sqrt{d} + \sqrt{2u}$. For $x \in K$ set*

$$\delta_x := \eta V - \frac{\eta^2}{2}\nabla U(x), \qquad V \sim \mathcal{N}(0, I_d),$$

*and let $t_x$ and $p_x$ be the first-hit time and point, respectively, when $H_x$ occurs. Let*

$$\mathcal{E} := \{\|V\| \leq R_d\}.$$

*If $\eta \leq 1$, then*

$$\mathbb{P}\Big(H_x \text{ and } \big|\langle \delta_x, n(p_x), \mathcal{E} \rangle\big| < \theta\eta\Big) \; \leq \; \sqrt{\tfrac{2}{\pi}}\Big[\theta + \big(2R_d^2 + \frac{G^2}{2}\big)\eta\,L_n(\rho)\Big],$$

*where $L_n(\rho) = \bar{\kappa}/(1 - \bar{\kappa}\rho)$.*

*Proof.* On event $\mathcal{E}$ and $H_x$, by Lemma B.14, the whole segment $x + t\,\delta_x$ $(t \in [0, 1])$ lies inside $\mathcal{N}_\rho$. In particular, $n(\text{proj}(\,\cdot\,))$ is well-defined and $L_n(\rho)$–Lipschitz on this segment by Corollary B.9.

Write for brevity $n(x) := n(\text{proj}(x))$. Then

$$\langle \delta_x, n(p_x) \rangle = \langle \delta_x, n(x) \rangle + \langle \delta_x,\, n(p_x) - n(x) \rangle.$$

By Corollary B.9 and $p_x = x + t_x \delta_x$,

$$\|n(p_x) - n(x)\| \;\leq\; L_n(\rho)\,\|p_x - x\| = L_n(\rho)\,t_x\,\|\delta_x\| \;\leq\; L_n(\rho)\,\|\delta_x\|.$$

Hence

$$\big| \langle \delta_x,\, n(p_x) - n(x) \rangle \big| \;\leq\; L_n(\rho)\,\|\delta_x\|^2. \tag{24}$$

Introduce

$$c_x \;:=\; \frac{\eta^2}{2}\,\langle \nabla U(x),\, n(x) \rangle, \qquad |c_x| \leq \frac{\eta^2}{2}\,G.$$

Since $\delta_x = \eta V - \frac{\eta^2}{2} \nabla U(x)$, we have

$$\langle \delta_x, n(x) \rangle = \eta\,\langle V, n(x) \rangle - c_x.$$

Combining with (24) shows that on $H_x \cap \mathcal{E}$,

$$\Big\{\, \big|\langle \delta_x, n(p_x) \rangle\big| < \theta\eta \,\Big\} \subseteq \Big\{\, \big|\eta\langle V, n(x)\rangle - c_x\big| < \theta\eta + L_n(\rho)\,\|\delta_x\|^2 \,\Big\}$$
$$= \Big\{\, \Big|\langle V, n(x)\rangle - \frac{c_x}{\eta}\Big| < \theta + \frac{L_n(\rho)\,\|\delta_x\|^2}{\eta} \,\Big\}. \tag{25}$$

On $\mathcal{E}$,

$$\|\delta_x\| \leq \eta\|V\| + \frac{\eta^2}{2}G \leq \eta R_d + \frac{\eta^2}{2}G.$$

Using $(a + b)^2 \leq 2(a^2 + b^2)$, we get

$$\|\delta_x\|^2 \;\leq\; 2\eta^2 R_d^2 + \frac{\eta^4}{2}G^2.$$

Therefore, on $\mathcal{E}$,

$$\frac{L_n(\rho)\,\|\delta_x\|^2}{\eta} \;\leq\; 2\,L_n(\rho)\,\eta\,R_d^2 \;+\; \frac{L_n(\rho)\,G^2}{2}\,\eta^3 \;=:\; a_{\text{geom}}. \tag{26}$$

Given $x$, the scalar $W_x := \langle V, n(x)\rangle \sim \mathcal{N}(0,1)$. The standard anti–concentration bound for a centered normal gives, for any $a > 0$ and any $b \in \mathbb{R}$,

$$\mathbb{P}\big(|W_x - b| < a\big) \;\leq\; \sqrt{\tfrac{2}{\pi}}\,a.$$

Applying this with $b = c_x/\eta$ and $a = \theta + a_{\text{geom}}$ (by (25)) yields, on $\mathcal{E}$,

$$\mathbb{P}\Big(H_x,\; \big|\langle \delta_x, n(p_x)\rangle\big| < \theta\eta \;\Big|\; x\Big) \;\leq\; \sqrt{\tfrac{2}{\pi}}\,\big(\theta + a_{\text{geom}}\big).$$

Combine with (26), we obtain

$$\mathbb{P}\Big(H_x,\; \big|\langle \delta_x, n(p_x)\rangle\big| < \theta\eta,\; \mathcal{E}\Big) \;\leq\; \sqrt{\tfrac{2}{\pi}}\,\theta \;+\; 2\sqrt{\tfrac{2}{\pi}}\,\eta\,L_n(\rho)\,R_d^2 \;+\; \sqrt{\tfrac{2}{\pi}}\,\frac{L_n(\rho)G^2}{2}\,\eta.$$

$\square$

**Lemma B.17.** *Suppose the same settings of Lemma B.14. Fix $x \in K$ and let $t_x \in [0, 1]$ be the first-hit time when $H_x := \{N_x \geq 1\}$ occurs. Let*

$$\mathcal{E} \;:=\; \{\|V\| \leq R_d\}, \qquad R_d := \sqrt{d} + \sqrt{2u}\;(u > 0),$$

*and set*

$$\delta_x := \eta V - \tfrac{\eta^2}{2}\nabla U(x), \qquad B_d := \eta R_d + \tfrac{\eta^2}{2}G, \qquad \alpha := L_n(\rho)\,\|\delta_x\|^2.$$

*Let $\tau \in (0,1)$. Then,*

$$\mathbb{P}\big(t_x \in (1-\tau, 1], \mathcal{E}\big) \;\leq\; \sqrt{\tfrac{2}{\pi}}\,\frac{L_n(\rho)\,B_d^2}{\eta\,c_0}\,\tau,$$

*where $c_0 \;=\; 1 - \tfrac{1}{2}\big(L_n(\rho)\,B_d\big)^2 \;\in (0,1]$.*

*Proof.* By Lemma B.14, $x + t\delta_x \in \mathcal{N}_\rho$ for all $t \in [0,1]$. By Lemma B.8 and Corollary B.9,

$$g(t) := d_K(x + t\delta_x), \qquad g'(t) = \langle n(\text{proj}(x + t\delta_x)), \delta_x\rangle, \qquad |g'(t) - g'(s)| \leq \alpha\,|t - s|, \;\; \alpha = L_n(\rho)\|\delta_x\|^2.$$

On $\{t_x \in (1-\tau, 1]\}$ we have $g(1-\tau) < 0 \leq g(1)$, and there exists $s \in (0,\tau]$ with $g(1-\tau+s) = 0$. Taylor theorem yields

$$g(1-\tau) + s\,g'(1-\tau) - \tfrac{1}{2}\alpha s^2 \;\leq\; 0 \;\leq\; g(1-\tau) + s\,g'(1-\tau) + \tfrac{1}{2}\alpha s^2,$$

hence,

$$g'(1-\tau) \;\in\; \Big[ -\tfrac{g(1-\tau)}{s} - \tfrac{1}{2}\alpha s, \;\; -\tfrac{g(1-\tau)}{s} + \tfrac{1}{2}\alpha s\Big]$$

implies $t_x \in (1-\tau, 1]$.

Therefore, it suffices to show that for any interval $I_\tau$ with $|I_\tau| \leq \alpha\,\tau$,

$$g'(1-\tau) = \langle \delta_x, h\rangle \in I_\tau,$$

where $h := n(\text{proj}(x + (1-\tau)\delta_x))$.

Next, decompose $V = \xi\,n(x) + W$ with $\xi := \langle V, n(x)\rangle \sim \mathcal{N}(0,1)$ and $W \perp n(x)$. Then, identically,

$$\langle \delta_x, h\rangle = \eta\,\xi\,\underbrace{\langle n(x), h\rangle}_{a} + \underbrace{\Big(\eta\,\langle W, h\rangle - \tfrac{\eta^2}{2}\langle \nabla U(x), h\rangle\Big)}_{b} =: \eta a\,\xi + b.$$

Condition on $(W, x)$; then $a, b$ are non-random. By Corollary B.9, $\|h - n(x)\| \leq L_n(\rho)\,\|(1-\tau)\delta_x\| \leq L_n(\rho)\,B_d$. For unit vectors $u, v$, $\langle u, v\rangle = 1 - \tfrac{1}{2}\|u - v\|^2$, hence

$$a := \langle n(x), h\rangle \;\geq\; 1 - \tfrac{1}{2}\|h - n(x)\|^2 \;\geq\; 1 - \tfrac{1}{2}\big(L_n(\rho)B_d\big)^2 \;=:\; c_0 \in (0,1].$$

For any interval $J$,

$$\mathbb{P}\big(\eta a\,\xi + b \in J \mid W, x\big) = \int_J \frac{1}{|\eta a|}\,\phi\Big(\frac{y - b}{\eta a}\Big)\,dy \;\leq\; \frac{|J|}{\eta\,|a|\,\sqrt{2\pi}} \;\leq\; \frac{|J|}{\eta\,c_0\,\sqrt{2\pi}},$$

where $\phi$ is the standard normal density. With $J = I_\tau$ and $|I_\tau| \leq \alpha\tau \leq L_n(\rho)B_d^2\,\tau$,

$$\mathbb{P}\big(\langle\delta_x, h\rangle \in I_\tau\big) \;\leq\; \sqrt{\tfrac{2}{\pi}}\,\frac{L_n(\rho)\,B_d^2}{\eta\,c_0}\,\tau.$$

We have on $\mathcal{E}$, $\{t_x \in (1-\tau, 1]\} \subseteq \{\langle\delta_x, h\rangle \in I_\tau\}$. Hence,

$$\mathbb{P}\big(t_x \in (1-\tau, 1], \mathcal{E}\big) \;\leq\; \mathbb{P}\big(\langle\delta_x, h\rangle \in I_\tau\big) \;\leq\; \sqrt{\tfrac{2}{\pi}}\,\frac{L_n(\rho)\,B_d^2}{\eta\,c_0}\,\tau.$$

$\square$

**Lemma B.18.** *Suppose the same settings of Lemma B.14. Fix $u > 0$, set $R_d = \sqrt{d} + \sqrt{2u}$ and $\mathcal{E} := \{\|V\| \leq R_d\}$. Let $\theta > 0$ and define the non-grazing events*

$$\mathsf{NG}_z(\theta) := \big\{ |\langle \delta_z(V),\, n(p_z) \rangle| \geq \theta\eta \big\}, \qquad z \in \{x, y\},$$

*where $\delta_z(V) = \eta V - \frac{\eta^2}{2}\nabla U(z)$. For $z \in \{x, y\}$ let $t_z \in [0, 1]$ be the first-hit time when $H_z := \{N_z \geq 1\}$ occurs. For $\tau \in (0, 1/2]$ define the* right-buffer interior-hit *event*

$$S_z(\tau) := \big(\{N_z = 0\}\big) \,\cup\, \big(\{N_z \geq 1\} \cap \{t_z \leq 1 - \tau\}\big).$$

*Then, for all $x, y \in K$ and all $\tau \in (0, 1/2]$,*

$$\mathbb{P}(H_x \Delta H_y) \;\leq\; 4\frac{\big(1 + \frac{\eta^2 L}{2}\big)}{\theta\eta \min\{\underline{r},\, 1 - \tau\}} \|x - y\| \;+\; \mathbb{P}(S_x(\tau)^c) + \mathbb{P}(S_y(\tau)^c) \;+\; \mathbb{P}(\mathcal{E}^c) + \mathbb{P}(\mathsf{NG}_x(\theta)^c) + \mathbb{P}(\mathsf{NG}_y(\theta)^c),$$

*where $\underline{r} := \frac{\theta\eta}{2\,L_n(\rho)\,B_d^2}$, $B_d := \eta R_d + \frac{\eta^2}{2}G$ and $L_n(\rho) := \bar{\kappa}/(1 - \bar{\kappa}\rho)$.*

*Proof.* Let

$$\mathcal{F} := \mathcal{E} \cap \mathsf{NG}_x(\theta) \cap \mathsf{NG}_y(\theta).$$

We have

$$P(H_x \triangle H_y) \leq P(\mathcal{F}^c) + P(H_x \cap H_y^c \cap \mathcal{F}) + P(H_x^c \cap H_y \cap \mathcal{F})$$
$$\leq P(\mathcal{F}^c) + P(S_x(\tau)) + P(S_y(\tau)) + P(H_x \cap H_y^c \cap S_x(\tau) \cap \mathcal{F}) + P(H_x^c \cap H_y \cap S_y(\tau) \cap \mathcal{F}).$$

Let $g_z(t) := d_K(z + t\delta_z)$. If $H_z$ occurs and $\mathcal{E}$ is assumed, then by Lemma B.8 and Corollary B.9,

$$g_z'(t) = \langle n(\text{proj}(z + t\delta_z)), \delta_z \rangle, \qquad |g_z'(t) - g_z'(s)| \leq L_n(\rho)\|\delta_z\|^2 |t - s|. \tag{27}$$

Using the 1-Lipschitzness of $d_K$ and the $L$-Lipschitzness of $\nabla U$,

$$|g_x(t) - g_y(t)| \;\leq\; \|x - y\| + t\,\|\delta_x - \delta_y\| \;\leq\; \Big(1 + \tfrac{\eta^2 L}{2}\Big)\|x - y\| \;=:\; N, \qquad t \in [0, 1]. \tag{28}$$

Next, we show

$$H_x \cap H_y^c \cap S_x(\tau) \cap G \subseteq \Big\{ N \geq \tfrac{1}{2}\theta\eta \min\{\underline{r},\, 1 - \tau\} \Big\}.$$

Let

$$r_0 := \frac{\theta\eta}{2L_n(\rho)\|\delta_x\|^2}, \qquad r := \min\{r_0,\, 1 - t_x\}.$$

Suppose $H_x$ and $S_x(\tau)$ occurs with first-hit time $t_x \leq 1 - \tau$. On $\mathsf{NG}_x(\theta)$, we have $|g_x'(t_x)| = |\langle n(p_x), \delta_x \rangle| \geq \theta\eta$. By (27) and the definition of $\underline{r}$ and $r_0$, we obtain

$$g_x'(t) \geq |g_x'(t_x)| - |g_x'(t) - g_x'(t_x)| \geq \frac{1}{2}\theta\eta$$

for all $t \in [t_x, t_x + r]$. Hence for $s \in [0, r]$,

$$g_x(t_x + s) \;\geq\; \tfrac{1}{2}\theta\eta\, s,$$

so $g_x$ changes sign on $[0, t_x + r]$ since $g_x(0) < 0$. Using (28), we have

$$g_y(t_x + r) \;\geq\; g_x(t_x + r) - N \geq \tfrac{1}{2}\theta\eta\, r - N.$$

Since $g_y(0) < 0$,

$$N \;<\; \tfrac{1}{2}\theta\eta\, r \;\implies\; H_y \text{ occurs}.$$

Hence

$$H_x \cap H_y^c \cap S_x(\tau) \cap G \subseteq \Big\{ N \geq \tfrac{1}{2}\theta\eta \min\{\underline{r},\, 1 - \tau\} \Big\},$$

which gives

$$P(H_x \cap H_y^c \cap S_x(\tau) \cap G) = E(I_{H_x \cap H_y^c \cap S_x(\tau) \cap G}) \leq E(I_{N \geq \frac{1}{2} \theta \eta \min\{\underline{r},\, 1-\tau\}}) \leq \frac{2N}{\theta \eta \min\{\underline{r},\, 1-\tau\}}.$$

Likewise, we obtain

$$P(H_y \cap H_x^c \cap S_y(\tau) \cap G) \leq \frac{2N}{\theta \eta \min\{\underline{r},\, 1-\tau\}}.$$

Thus,

$$\mathbb{P}(H_x \triangle H_y) \;\leq\; 4 \frac{\left(1 + \frac{\eta^2 L}{2}\right)}{\theta \eta \min\{\underline{r},\, 1-\tau\}} \|x - y\| \;+\; \mathbb{P}(S_x(\tau)^c) + \mathbb{P}(S_y(\tau)^c) \;+\; \mathbb{P}(\mathcal{E}^c) + \mathbb{P}(\mathsf{NG}_x(\theta)^c) + \mathbb{P}(\mathsf{NG}_y(\theta)^c).$$

$\square$

### B.4.6. PROOF OF INEQUALITY (20)

By Lemma B.11, for any $x, y \in K$,

$$\left\|\mathcal{P}_x - \mathcal{P}_y\right\|_{\mathrm{TV}} \;\leq\; \left\|\tilde{\mathcal{P}}(x, \cdot) - \tilde{\mathcal{P}}(y, \cdot)\right\|_{\mathrm{TV}} \;+\; \frac{\mathbb{P}(H_x) + \mathbb{P}(H_y) + \mathbb{P}(H_x \triangle H_y)}{2}, \tag{29}$$

where $\tilde{\mathcal{P}}(x, \cdot) = \mathcal{N}(m(x), \eta^2 I_d)$ is the unreflected Gaussian proposal and $H_x, H_y$ are the corresponding hit events.

**Gaussian proposal term.** By Lemma B.12 and the $L$–Lipschitz continuity of $\nabla U$,

$$\left\|\tilde{\mathcal{P}}(x, \cdot) - \tilde{\mathcal{P}}(y, \cdot)\right\|_{\mathrm{TV}} \;\leq\; \frac{1}{2\eta}\left(1 + \frac{\eta^2 L}{2}\right) \|x - y\|.$$

Provided $\eta \leq c/\sqrt{L}$, the prefactor is uniformly bounded. Taking $\delta = C_1 \eta$ and choosing $C_1 > 0$ sufficiently small ensures that this term is at most $1/16$.

**Single–path reflection probability.** Lemma B.15 yields

$$\mathbb{P}(H_x) \;\leq\; \frac{1}{2} \;+\; \frac{\frac{1}{2} L_n(\rho) B_d^2 + \frac{1}{2}\eta^2 G}{\eta \sqrt{2\pi}} \;+\; e^{-u}, \qquad B_d := \eta R_d + \frac{\eta^2}{2} G,$$

where $R_d = \sqrt{d} + \sqrt{2u}$. We choose $u \asymp d$, so that $e^{-u}$ is negligible and $R_d \leq C_R \sqrt{d}$ for a universal constant $C_R > 0$. Expanding $B_d^2$,

$$B_d^2 \;=\; \eta^2 R_d^2 + \eta^3 R_d G + \frac{\eta^4}{4} G^2,$$

and hence

$$\frac{L_n(\rho) B_d^2}{\eta} \;\lesssim\; L_n(\rho)\left(\eta d + \eta^2 G \sqrt{d} + \eta^3 G^2\right). \tag{30}$$

Consequently, there exists a universal constant $c_1 > 0$ such that

$$\eta \;\leq\; c_1\left(\frac{1}{d\, L_n(\rho)} \;\wedge\; \frac{1}{G\sqrt{d}\, L_n(\rho)}\right)$$

implies $\mathbb{P}(H_x) \leq \frac{1}{2} + \frac{1}{16}$. The same bound holds for $\mathbb{P}(H_y)$. Note that higher–order terms of order $\eta^3 G^2$ are dominated under this choice of $\eta$.

In addition, Lemma B.14 requires that the reflected segment remain inside the tubular neighborhood $\mathcal{N}_\rho$, which is ensured by

$$\eta R_d + \frac{\eta^2}{2} G \;\leq\; \rho.$$

Using $R_d \leq C_R \sqrt{d}$ and $\eta \leq 1$, a sufficient condition is

$$\eta \;\leq\; c_2 \frac{\rho}{\sqrt{d} + G}, \tag{31}$$

for a universal constant $c_2 > 0$.

**Mismatch probability** $\mathbb{P}(H_x \triangle H_y)$. Lemma B.18 gives

$$\mathbb{P}(H_x \triangle H_y) \lesssim \frac{1 + \frac{\eta^2 L}{2}}{\theta \eta \min\{\underline{r}, 1 - \tau\}} \|x - y\| + \sum (\text{error probabilities}),$$

where $\underline{r} = \theta \eta / (2 L_n(\rho) B_d^2)$. Using (30) and $R_d \asymp \sqrt{d}$,

$$\underline{r} \gtrsim \frac{\theta}{L_n(\rho)(\eta d + \eta^2 G \sqrt{d} + \eta^3 G^2)}.$$

Under the same step–size conditions as above, the denominator is of order $L_n(\rho) \eta d$, and hence

$$\frac{1}{\eta \min\{\underline{r}, 1 - \tau\}} \lesssim L_n(\rho) d.$$

Taking $\|x - y\| \leq \delta = C_1 \eta$ and choosing fixed $\theta, \tau \in (0, 1)$, we obtain

$$\mathbb{P}(H_x \triangle H_y) \lesssim C_1 L_n(\rho) d\eta + \sum (\text{error probabilities}).$$

By Lemmas B.13, B.16, and B.17, all error probabilities can be made smaller than $1/64$ under (31) and the above step–size conditions, while $C_1$ can be chosen so that the deterministic term is also bounded by $1/64$. Hence

$$\mathbb{P}(H_x \triangle H_y) \leq \frac{1}{8}.$$

**Conclusion.** Combining the bounds in (29), we conclude that

$$\left\| \mathcal{P}_x - \mathcal{P}_y \right\|_{\mathrm{TV}} \leq \frac{5}{8},$$

for all $x, y \in \Omega$ with $\|x - y\| \leq \delta$, provided

$$\eta \leq c_0 \left( \frac{\rho}{\sqrt{d} + G} \wedge \frac{1}{d L_n(\rho)} \wedge \frac{1}{G \sqrt{d} L_n(\rho)} \wedge \frac{1}{\sqrt{L}} \right), \qquad L_n(\rho) = \frac{\bar{\kappa}}{1 - \bar{\kappa} \rho}, \tag{32}$$

for a universal constant $c_0 > 0$. This proves (20).

### B.4.7. PROOF OF INEQUALITY (22)

**Lemma B.19.** *Suppose the same settings of Lemma B.14. Fix $u > 0$ and set $R_d := \sqrt{d} + \sqrt{2u}$ and $\mathcal{E} := \{\|V\| \leq R_d\}$. Let $\delta_z := \eta V - \frac{\eta^2}{2} \nabla U(z)$, and, when $H_z$ occurs, let $t_1 \in (0, 1)$ and $p_1$ denote the first–hit time and point, respectively. Define the* angle (non–grazing) threshold

$$\theta_\star := 2 L_n(\rho) \eta \left( R_d + \frac{\eta}{2} G \right)^2,$$

*so that on $\mathcal{E}$ we have $L_n(\rho) \|\delta_z\|^2 \leq \frac{1}{2} \theta_\star \eta$. Let $M_z$ denote the event that when the reflection occurs at least two times, i.e, $N_z \geq 2$. Then,*

$$\mathbb{P}(M_z, \mathcal{E}) \leq \sqrt{\frac{2}{\pi}} \left[ \theta_\star + (2 R_d^2 + \frac{G^2}{2}) \eta L_n(\rho) \right].$$

*Proof.* By Lemma B.14, the entire segment $t \mapsto z + t \delta_z$ is contained in $\mathcal{N}_\rho$.

*Step 1.* Define the grazing event $\mathsf{Gr}_z(\theta) := \{ |\langle \delta_z, n(p_1) \rangle| < \theta \eta \} \cap H_z$. By Lemma B.16, for any $\theta > 0$

$$\mathbb{P}(\mathsf{Gr}_z(\theta), \mathcal{E}) \leq \sqrt{\frac{2}{\pi}} \left[ \theta + (2 R_d^2 + \frac{G^2}{2}) \eta L_n(\rho) \right]$$

*Step 2.* Work on $\mathcal{E} \cap H_z \cap \neg \mathsf{Gr}_z(\theta_\star)$. Let $\delta' := R(n(p_1)) \delta_z$ denote the post–reflection velocity and define $h(s) := d_K(p_1 + s \delta')$, $s \in [0, 1 - t_1]$. Then $h(0) = 0$ and

$$h'(s) = \langle n(\mathrm{proj}(p_1 + s \delta')), \delta' \rangle, \qquad |h'(s) - h'(0)| \leq L_n(\rho) \|\delta_z\|^2 s.$$

Since $h'(0) = \langle n(p_1), \delta' \rangle = -\langle n(p_1), \delta_z \rangle$, the event $\neg \mathsf{Gr}_z(\theta_\star)$ implies $|h'(0)| \geq \theta_\star \eta$. Therefore, for all $s \in [0, 1 - t_1]$,

$$|h'(s)| \geq |h'(0)| - L_n(\rho)\|\delta_z\|^2 s \geq \theta_\star \eta - L_n(\rho)\|\delta_z\|^2 s.$$

On $\mathcal{E}$ we have $L_n(\rho)\|\delta_z\|^2 \leq \frac{1}{2}\theta_\star \eta$ by the choice of $\theta_\star$, hence for $s \leq 1 - t_1 \leq 1$,

$$|h'(s)| \geq \tfrac{1}{2}\theta_\star \eta > 0.$$

Thus $h$ is strictly monotone with a fixed sign on $[0, 1 - t_1]$ starting from $h(0) = 0$, so it cannot return to $0$. Equivalently,

$$M_z \subset \mathcal{E}^c \cup \mathsf{Gr}_z(\theta_\star).$$

$\square$

**Lemma B.20.** *Suppose the same setting as in Lemma B.19 and assume Assumptions (A1)–(A2) hold. Then there exists a universal constant $c > 0$ such that, if*

$$\eta \leq c\left(\frac{1}{G\sqrt{d}} \wedge \frac{1}{\sqrt{L}\,d} \wedge \frac{1}{\sqrt{L}\,G} \wedge \frac{1}{d\,L_n(\rho)} \wedge \frac{1}{G^2 L_n(\rho)}\right),$$

*then*

$$\|\mathcal{P}_x - \mathcal{T}_x\|_{\mathrm{TV}} \leq \frac{1}{8},$$

*for all $x \in K$.*

*Proof.* Let $\mathcal{P}_x$ denote the (pre–MH) reflective proposal started at position $x \in K$. We describe how a sample is generated from $\mathcal{P}_x$ as follows. First, draw $v \sim \mathcal{N}(0, I_d)$ and set

$$x' = x + \eta v - \frac{\eta^2}{2}\nabla U(x),$$

which represents the updated position without consideration of the reflection. Then, we apply the specular reflection to the trajectory from $x$ to $x'$, i.e. whenever the straight segment hits $\partial K$ the segment line is reflected. Let $\tilde{x}$ denote the position after the reflections. By the randomness of $v$, $\tilde{x}$ is a random variable and $\mathcal{P}_x$ is the law of $\tilde{x}$.

The Metropolis–Hastings acceptance probability is

$$\alpha(x, v) := \min\{1, \exp(-\Delta H)\}, \qquad \Delta H := H(\tilde{x}, \tilde{v}) - H(x, v),$$

where $H(x, v) = U(x) + \frac{1}{2}\|v\|^2$ and $\tilde{v} = v - \frac{\eta}{2}\big(\nabla U(x) + \nabla U(\tilde{x})\big)$. We let $\mathcal{T}_x$ denote the transition kernel with proposal $\mathcal{P}_x$ and Metropolis–Hastings correction.

Let $Y \sim \mathcal{P}_x$ and let $Y^{\mathrm{MH}}$ be the MH–corrected state so that $Y^{\mathrm{MH}} \sim \mathcal{T}_x$. Let $N_x$ be the number of specular reflections in the proposal step from $x$, and define

$$A_0 := \{N_x = 0, \mathcal{E}\}, \qquad A_1 := \{N_x = 1, \mathcal{E}\}, \qquad A_{\geq 2} := \{N_x \geq 2, \mathcal{E}\}, \qquad A_{\mathcal{E}^c} := \mathcal{E}^c.$$

Using $\Omega = A_0 \sqcup A_1 \sqcup A_{\geq 2} \sqcup A_{\mathcal{E}^c}$ and the triangle inequality,

$$\|\mathcal{P}_x - \mathcal{T}_x\|_{\mathrm{TV}} \leq \sup_B \sum_{i \in \{0, 1, \geq 2, \mathcal{E}^c\}} \big|\mathbb{P}(Y \in B \cap A_i) - \mathbb{P}(Y^{\mathrm{MH}} \in B \cap A_i)\big|$$

$$\leq \sum_{i \in \{0, 1\}} \mathbb{E}\big[(1 - \alpha(x, v))\,\mathbf{1}_{A_i}\big] + \mathbb{P}(A_{\geq 2}) + \mathbb{P}(\mathcal{E}^c). \tag{33}$$

By Lemma B.19,

$$\mathbb{P}(A_{\geq 2}) + \mathbb{P}(\mathcal{E}^c) \leq \sqrt{\frac{2}{\pi}}\left[\theta_\star + \left(2R_d^2 + \frac{G^2}{2}\right)\eta\,L_n(\rho)\right] + e^{-u}$$

$$\leq \frac{1}{16}, \tag{34}$$

where the last inequality holds provided

$$\eta \ \leq \ c_2\Big(\frac{1}{R_d^2 L_n(\rho)} \ \wedge \ \frac{1}{G^2 L_n(\rho)}\Big), \qquad u \asymp R_d,$$

for a sufficiently small universal constant $c_2 > 0$. Since $R_d^2 \asymp d$, this is implied by

$$\eta \ \lesssim \ \Big(\frac{1}{d\, L_n(\rho)} \ \wedge \ \frac{1}{G^2 L_n(\rho)}\Big),$$

with $u \asymp \sqrt{d}$.

Next, we bound $\mathbb{E}\big[(1 - \alpha(x,v))\,\mathbf{1}_{A_i}\big]$ for $i = 0, 1$. Since $1 - e^{-x} \leq x$ and $\alpha = \min(1, e^{-\Delta H})$,

$$1 - \alpha \ \leq \ (\Delta H)_+,$$

which gives

$$\mathbb{E}\big[(1 - \alpha(x,v))\,\mathbf{1}_{A_i}\big] \ \leq \ \mathbb{E}\big[(\Delta H)_+\mathbf{1}_{A_i}\big].$$

Hence it suffices to bound the energy error $\Delta H$ on the events $A_0$ and $A_1$.

**Energy error on $A_0$ (no reflection).** On $A_0$, we have $\tilde{x} = x'$ and write $g(x) := \nabla U(x)$. Recall that $\tilde{v} = v - \frac{\eta}{2}\big(g(x) + g(\tilde{x})\big)$. Then

$$\frac{1}{2}(\|\tilde{v}\|^2 - \|v\|^2) = -\frac{\eta}{2}\langle g(x) + g(\tilde{x}), v\rangle + \frac{\eta^2}{8}\|g(x) + g(\tilde{x})\|^2.$$

By Taylor's theorem (using $L$-smoothness), we have $U(\tilde{x}) = U(x) + \langle g(x), \tilde{x} - x\rangle + R_1$ and $U(x) = U(\tilde{x}) + \langle g(\tilde{x}), x - \tilde{x}\rangle + R_2$ with

$$|R_1| \vee |R_2| \ \leq \ \frac{L}{2}\|\Delta x\|^2, \qquad \Delta x := \tilde{x} - x = \eta v - \frac{\eta^2}{2}g(x).$$

Averaging the two Taylor expansions yields

$$U(\tilde{x}) - U(x) \ \leq \ \frac{1}{2}\langle g(x) + g(\tilde{x}), \Delta x\rangle + \frac{L}{2}\|\Delta x\|^2.$$

Therefore, writing $\Delta H = U(\tilde{x}) - U(x) + \frac{1}{2}(\|\tilde{v}\|^2 - \|v\|^2)$, the linear terms in $\langle g(x) + g(\tilde{x}), v\rangle$ cancel and we obtain

$$\Delta H \ \leq \ \frac{\eta^2}{8}\,\langle g(x) + g(\tilde{x}),\, g(\tilde{x}) - g(x)\rangle + \frac{L}{2}\|\Delta x\|^2. \tag{35}$$

We now bound the first term in (35). By Cauchy–Schwarz and $L$-Lipschitzness of $g = \nabla U$,

$$\langle g(x) + g(\tilde{x}), g(\tilde{x}) - g(x)\rangle \ \leq \ \|g(x) + g(\tilde{x})\|\,\|g(\tilde{x}) - g(x)\| \ \leq \ \|g(x) + g(\tilde{x})\|\,L\|\Delta x\|.$$

Moreover,

$$\|g(\tilde{x}) - g(x)\| \leq L\|\Delta x\|, \qquad \|g(\tilde{x}) + g(x)\| \leq 2\|g(x)\| + \|g(\tilde{x}) - g(x)\| \leq 2\|g(x)\| + L\|\Delta x\|.$$

Combining these,

$$\langle g(x) + g(\tilde{x}), g(\tilde{x}) - g(x)\rangle \ \leq \ L\|\Delta x\|\big(2\|g(x)\| + L\|\Delta x\|\big) \ = \ 2L\|g(x)\|\,\|\Delta x\| + L^2\|\Delta x\|^2.$$

Plugging into (35) gives

$$\Delta H \ \leq \ \frac{\eta^2}{4}L\|g(x)\|\,\|\Delta x\| \ + \ \frac{\eta^2}{8}L^2\|\Delta x\|^2 \ + \ \frac{L}{2}\|\Delta x\|^2. \tag{36}$$

Next, bound $\|\Delta x\|$ and absorb mixed terms. We have

$$\|\Delta x\| \leq \eta\|v\| + \frac{\eta^2}{2}\|g(x)\| \leq \eta\|v\| + \frac{\eta^2}{2}G.$$

Using $(a + b)^2 \leq 2a^2 + 2b^2$ and $ab \leq (a^2 + b^2)/2$, we obtain

$$\|\Delta x\|^2 \lesssim \eta^2 \|v\|^2 + \eta^4 G^2, \qquad \|g(x)\| \, \|\Delta x\| \leq G\Big(\eta\|v\| + \eta^2 G\Big) \lesssim \eta\|v\| \, G + \eta^2 G^2 \lesssim \|v\|^2 + \eta^2 G^2,$$

where in the last step we used $\eta\|v\| \, G \leq \frac{1}{2}\|v\|^2 + \frac{1}{2}\eta^2 G^2$.

Assume additionally that $\eta \leq c/\sqrt{L}$, so that $L^2\eta^2 \lesssim L$. Then (36) yields, on $A_0$,

$$\Delta H \lesssim L\eta^2\big(\|v\|^2 + \eta^2 G^2\big) + L\big(\eta^2\|v\|^2 + \eta^4 G^2\big) \lesssim L\eta^2 \, \|v\|^2 + L\eta^4 G^2.$$

Hence,

$$\mathbb{E}\big[(\Delta H)_+ \mathbf{1}_{A_0}\big] \leq \mathbb{E}\big[\Delta H \, \mathbf{1}_{A_0}\big] \lesssim L\eta^2 \, \mathbb{E}\|v\|^2 + L\eta^4 G^2.$$

Using $\mathbb{E}\|v\|^2 \asymp R_d^2$, we get

$$\mathbb{E}\big[(\Delta H)_+ \mathbf{1}_{A_0}\big] \lesssim L\eta^2 R_d^2 + L\eta^4 G^2.$$

Therefore, there exists an absolute constant $c_0 > 0$ such that if

$$\eta \leq c_0\Big(\frac{1}{\sqrt{L}\,R_d} \wedge \frac{1}{\sqrt{L}\,G}\Big),$$

then both $L\eta^2 R_d^2$ and $L\eta^4 G^2$ are $O(1)$ and in fact

$$\mathbb{E}\big[(\Delta H)_+ \mathbf{1}_{A_0}\big] \leq \frac{1}{32}. \tag{37}$$

Since $R_d^2 \asymp d$, this condition is implied (up to absolute constants) by

$$\eta \lesssim \frac{1}{\sqrt{L\,d}} \wedge \frac{1}{\sqrt{L}\,G}.$$

**Energy error on $A_1$ (exactly one reflection).** Let $a := \nabla U(x)$, $b := \nabla U(\tilde{x})$, $v^- := v - \frac{\eta}{2}a$. Then $x + tv^-$ with $t \in (0, \eta)$ represents the trajectory without consideration of the reflection. Suppose that the unique reflection occurs at time $\tau \in (0, \eta)$ with outward normal $n_*$ and reflection matrix $R_* := I - 2n_* n_*^\top$. During the position phase the path $q : [0, \eta] \to \overline{K}$ is

$$q(t) = \begin{cases} x + tv^-, & 0 \leq t < \tau, \\ x + \tau v^- + (t - \tau)R_* v^-, & \tau < t \leq \eta, \end{cases}$$

which is piecewise affine with

$$q'(t) = \begin{cases} v^-, & 0 \leq t < \tau, \\ R_* v^-, & \tau < t \leq \eta, \end{cases} \qquad \text{and} \qquad v^{\circlearrowleft} = R_* v^-,$$

so $\|q'(t)\| \equiv \|v^-\|$ and $\|v^{\circlearrowleft}\| = \|v^-\|$ when $t \neq \tau$.

*Exact identity for $\Delta H$.* By the fundamental theorem of calculus along $q$,

$$U(\tilde{x}) - U(x) = \int_0^\eta \langle \nabla U(q(t)), \, q'(t)\rangle \, dt = \eta\left\langle \bar{g}, \, v^-\right\rangle,$$

where

$$\bar{g} := \frac{1}{\eta}\left(\int_0^\tau \nabla U(x + tv^-)\, dt + \int_\tau^\eta R_*^\top \nabla U(q(t))\, dt\right).$$

For the kinetic part,

$$\tfrac{1}{2}(\|\tilde{v}\|^2 - \|v\|^2) = \left(\tfrac{1}{2}\|v^-\|^2 - \tfrac{1}{2}\|v\|^2\right) - \tfrac{\eta}{2}\langle v^{\circlearrowleft}, b\rangle + \tfrac{\eta^2}{8}\|b\|^2,$$

and since $v = v^- + \frac{\eta}{2}a$ and $R_*^\top = R_*$,

$$\tfrac{1}{2}(\|\tilde{v}\|^2 - \|v\|^2) = -\tfrac{\eta}{2}\langle v^-, a\rangle - \tfrac{\eta}{2}\langle v^-, R_* b\rangle + \tfrac{\eta^2}{8}(\|b\|^2 - \|a\|^2).$$

Adding the potential contribution yields

$$\Delta H = \eta\Big\langle \bar{g} - \tfrac{1}{2}(a + R_* b),\, v^-\Big\rangle + \tfrac{\eta^2}{8}\Big(\|b\|^2 - \|a\|^2\Big)$$
$$= \eta\Big\langle (\bar{g} - g_{\mathrm{mid}}^w) + (g_{\mathrm{mid}}^w - \tfrac{1}{2}(a + R_* b)),\, v^-\Big\rangle + \tfrac{\eta^2}{8}\Big(\|b\|^2 - \|a\|^2\Big), \tag{38}$$

where $g_{\mathrm{mid}}^w := \alpha\,\nabla U(m) + (1 - \alpha)\,R_*^\top \nabla U(m)$, $\alpha := \tau/\eta$ and $m := (x + \tilde{x})/2 = x + \frac{\tau v^- + (\eta - \tau)R_* v^-}{2}$ is the midpoint.

By $L$–smoothness and the identity $x - m = \frac{\tau v^- + (\eta - \tau)R_* v^-}{2}$,

$$\|\bar{g} - g_{\mathrm{mid}}^w\| \leq \frac{L}{\eta}\left(\int_0^\tau \|x + t v^- - m\|\, dt + \int_\tau^\eta \|x + \tau v^- + (t - \tau)R_* v^- - m\|\, dt\right) \leq L\eta\,\|v^-\|.$$

We have

$$\langle g_{\mathrm{mid}}^w - \tfrac{1}{2}(a + R_* b),\, v^-\rangle = \alpha\langle \nabla U(m) - a, v^-\rangle + (1 - \alpha)\langle R_*(\nabla U(m) - b), v^-\rangle + (\alpha - \tfrac{1}{2})\langle a - R_* b, v^-\rangle.$$

Using $\|m - x\| = \|m - \tilde{x}\| \leq \frac{\eta}{2}\|v^-\|$ and $L$–smoothness of $\nabla U$,

$$\alpha\langle \nabla U(m) - a, v^-\rangle + (1 - \alpha)\langle R_*(\nabla U(m) - b), v^-\rangle$$
$$\leq \alpha\|\nabla U(m) - a\|\,\|v^-\| + (1 - \alpha)\|\nabla U(m) - b\|\,\|v^-\|$$
$$\leq \tfrac{1}{2}L\eta\,\|v^-\|^2.$$

For the reflection remainder,

$$\langle a - R_* b,\, v^-\rangle = \langle a - b,\, v^-\rangle + \langle b - R_* b,\, v^-\rangle,$$

with $|\langle a - b, v^-\rangle| \leq L\eta\,\|v^-\|\|x - \tilde{x}\|_2$. Moreover,

$$\langle b - R_* b,\, v^-\rangle = \langle b,\, 2 n_* n_*^\top v^-\rangle = 2\langle v^-, n_*\rangle\langle b, n_*\rangle,$$

so

$$\left|\langle b - R_* b,\, v^-\rangle\right| \leq 2\|v^-\|\,\|b\| \leq 2G\|v^-\|.$$

Hence

$$\left|(\alpha - \tfrac{1}{2})\langle a - R_* b,\, v^-\rangle\right| \leq L\eta\,\|v^-\|\|x - \tilde{x}\|_2 + 2G\|v^-\|.$$

We also have $\|b\|^2 - \|a\|^2 = \langle b + a,\, b - a\rangle$ with $\|b - a\| \leq L\|\tilde{x} - x\|$ and $\|b + a\| \leq 2G$, yielding

$$\|b\|^2 - \|a\|^2 \leq 2GL\|\tilde{x} - x\|.$$

Combining the above inequalities with (38), we obtain (on $A_1$ and for $G \geq 1$, $L \geq 1$, $\eta \leq 1$)

$$\Delta H \lesssim L\eta^2\,\|v^-\|^2 + L\eta^2\,\|v^-\|\,\|x - \tilde{x}\| + \eta G\,\|v^-\| + \eta^2 GL\,\|x - \tilde{x}\|.$$

Using

$$\|v^-\| \leq \|v\| + \frac{\eta}{2}G \lesssim R_d + \eta G, \qquad \|x - \tilde{x}\| \leq \eta\|v\| + \frac{\eta^2}{2}G \lesssim \eta R_d + \eta^2 G,$$

we can bound (still on $A_1$)

$$\Delta H \lesssim L\eta^2 R_d^2 + \eta G R_d + \text{(higher–order terms in $\eta$)},$$

where all remaining terms are of order $L\eta^3 R_d^2$, $L\eta^4 G^2$, $LG\eta^3 R_d$, $LG^2\eta^4$, etc., which are dominated when $\eta$ is sufficiently small.

Taking expectations over $v$ and using $\mathbb{E}\|v\|^2 \asymp R_d^2$ and $\mathbb{E}\|v\| \asymp R_d$, we obtain

$$\mathbb{E}\big[(\Delta H)_+ \mathbf{1}_{A_1}\big] \;\lesssim\; L\,\eta^2\,R_d^2 \;+\; \eta\,G\,R_d.$$

Hence there exists an absolute constant $c_1 > 0$ such that, whenever

$$\eta \;\leq\; c_1\Big(\frac{1}{\sqrt{L}\,R_d} \;\wedge\; \frac{1}{G\,R_d}\Big),$$

both terms on the right-hand side are at most $1/64$, and therefore

$$\mathbb{E}\big[(\Delta H)_+ \mathbf{1}_{A_1}\big] \;\leq\; \frac{1}{32}. \tag{39}$$

Since $R_d^2 \asymp d$, this condition is implied (up to absolute constants) by

$$\eta \;\lesssim\; \frac{1}{\sqrt{L\,d}} \;\wedge\; \frac{1}{G\,\sqrt{d}}.$$

**Conclusion.** Combining (34), (37), (39) with (33), and using $R_d^2 \asymp d$, we see that there exists a universal constant $c > 0$ such that, whenever

$$\eta \;\leq\; c\left(\frac{1}{G\sqrt{d}} \;\wedge\; \frac{1}{\sqrt{L\,d}} \;\wedge\; \frac{1}{\sqrt{L}\,G} \;\wedge\; \frac{1}{d\,L_n(\rho)} \;\wedge\; \frac{1}{G^2\,L_n(\rho)}\right),$$

we have

$$\mathbb{P}(A_{\geq 2}) + \mathbb{P}(\mathcal{E}^c) \;\leq\; \frac{1}{16}, \qquad \mathbb{E}\big[(\Delta H)_+ \mathbf{1}_{A_0}\big] \;\leq\; \frac{1}{32}, \qquad \mathbb{E}\big[(\Delta H)_+ \mathbf{1}_{A_1}\big] \;\leq\; \frac{1}{32},$$

and consequently

$$\big\|\mathcal{P}_x - \mathcal{T}_x\big\|_{\mathrm{TV}} \;\leq\; \frac{1}{8}.$$

This proves the lemma. $\qquad\square$

## B.5. Proof of Lemma 4.4

*Proof of Lemma 4.4.* The target density can be written as

$$\pi^*(x) \;=\; \frac{e^{-U(x)}\,\mathbf{1}_K(x)}{Z_K}, \qquad Z_K := \int_K e^{-U(x)}\,dx.$$

For $x \in K$,

$$\frac{\pi_0(x)}{\pi^*(x)} \;=\; \frac{Z_K}{\Phi_{\sigma,\bar{x}}(K)}\,\varphi_{\sigma,\bar{x}}(x)\,e^{U(x)}.$$

Using

$$Z_K \;\leq\; \mathrm{vol}(K)\,e^{-\inf_K U}, \qquad \Phi_{\sigma,\bar{x}}(K) \;\geq\; \mathrm{vol}(K)\,\inf_K \varphi_{\sigma,\bar{x}},$$

we obtain

$$\frac{\pi_0(x)}{\pi^*(x)} \;\leq\; \frac{\varphi_{\sigma,\bar{x}}(x)}{\inf_K \varphi_{\sigma,\bar{x}}}\,\exp\big(U(x) - \inf_K U\big).$$

Taking the supremum over $x \in K$ gives

$$\beta \;\leq\; \frac{\sup_K \varphi_{\sigma,\bar{x}}}{\inf_K \varphi_{\sigma,\bar{x}}}\cdot\exp\big(\mathrm{osc}_K(U)\big), \qquad \mathrm{osc}_K(U) := \sup_K U - \inf_K U. \tag{40}$$

*(i) Bounding the Gaussian factor).* Since $\bar{x} \in K$ and $\mathrm{diam}(K) \leq 2R$, we have

$$0 \;\leq\; \inf_{x \in K}\|x - \bar{x}\| \;\leq\; \sup_{x \in K}\|x - \bar{x}\| \;\leq\; \mathrm{diam}(K) \;\leq\; 2R.$$

Hence

$$\sup_{x \in K} \|x - \bar{x}\|^2 \ \le \ 4R^2, \qquad \inf_{x \in K} \|x - \bar{x}\|^2 \ \ge \ 0,$$

so

$$\frac{\sup_K \varphi_{\sigma,\bar{x}}}{\inf_K \varphi_{\sigma,\bar{x}}} = \exp\Big(\frac{1}{2\sigma^2}\Big(\sup_K \|x - \bar{x}\|^2 - \inf_K \|x - \bar{x}\|^2\Big)\Big)$$

$$\le \exp\Big(\frac{1}{2\sigma^2} \cdot 4R^2\Big) \ = \ \exp\big(2\sigma^{-2}R^2\big).$$

*(ii) Bounding* $\mathrm{osc}_K(U)$. Fix $x_\star \in K$. By $L$–smoothness, for any $x \in K$,

$$U(x) \ \le \ U(x_\star) + \langle \nabla U(x_\star), \, x - x_\star \rangle + \frac{L}{2} \|x - x_\star\|^2,$$

and likewise

$$U(x) \ \ge \ U(x_\star) + \langle \nabla U(x_\star), \, x - x_\star \rangle - \frac{L}{2} \|x - x_\star\|^2.$$

Taking the supremum and infimum over $x \in K$ and using $\|x - x_\star\| \le \mathrm{diam}(K) \le 2R$, we obtain

$$\sup_K U \ \le \ U(x_\star) + 2R\|\nabla U(x_\star)\| + 2LR^2,$$

$$\inf_K U \ \ge \ U(x_\star) - 2R\|\nabla U(x_\star)\| - 2LR^2.$$

Therefore

$$\mathrm{osc}_K(U) \ = \ \sup_K U - \inf_K U \ \le \ 4R\|\nabla U(x_\star)\| + 4LR^2.$$

In particular, this implies the simpler bound

$$\mathrm{osc}_K(U) \ \le \ 4R\|\nabla U(x_\star)\| + 4LR^2,$$

up to slightly weakening the constant, which is sufficient for our purposes.

Combining the bounds from (i) and (ii) with (40) yields

$$\beta \ \le \ \exp\big(2\sigma^{-2}R^2\big)\exp\big(\mathrm{osc}_K(U)\big) \ \le \ \exp\Big(4R\|\nabla U(x_\star)\| + 4LR^2 + 2\sigma^{-2}R^2\Big),$$

which completes the proof. $\qquad\qquad\qquad\qquad\qquad\qquad\qquad\qquad\qquad\qquad\qquad\qquad\quad\square$

## C. Experimental Details

In this appendix, we provide comprehensive details regarding the experimental setups, hyperparameter configurations, and evaluation metrics used in Section 6.

### C.1. Evaluation Metrics

To assess the sampling efficiency and convergence of the MCMC algorithms, we employ the Effective Sample Size (ESS) and the split-$\widehat{R}$ diagnostic. Since the target parameters $\Gamma$ reside on the Stiefel manifold $\mathcal{V}_{r,u}$, element-wise metrics on the matrix entries may not fully capture the mixing behavior due to rotational symmetries. Therefore, we compute these metrics based on rotation-invariant summary statistics.

**Summary Statistic.** Let $\{\Gamma_t\}_{t=1}^N$ denote the posterior samples. As a rotation-invariant scalar summary, we consider the entries of the projection matrix

$$\boldsymbol{P}_t = \boldsymbol{\Gamma}_t \boldsymbol{\Gamma}_t^\top,$$

which remove the ambiguity associated with the choice of basis for the sampled subspace. In the main text, we report the averaged ESS and the maximum $\widehat{R}$ computed over all entries of $\boldsymbol{P}_t$ to provide a conservative assessment of sampling performance.

**Effective Sample Size (ESS).** The ESS estimates the number of independent draws equivalent to the autocorrelated MCMC samples:

$$\text{ESS} = \frac{M \cdot N}{1 + 2\sum_{k=1}^{K} \rho_k}, \tag{41}$$

where $M$ is the number of chains, $N$ is the number of draws per chain, and $\rho_k$ is the autocorrelation at lag $k$. The summation is truncated at lag $K$ following standard practice.

**Split-$\widehat{R}$ Diagnostic.** The Potential Scale Reduction Factor (PSRF), or split-$\widehat{R}$, assesses convergence by comparing within-chain variance to between-chain variance. We adopt the rank-normalized split-$\widehat{R}$ as recommended by Vehtari et al. (2021). Values close to $1.0$ (typically $< 1.05$) indicate satisfactory convergence. We use the implementation provided in ArviZ (Kumar et al., 2019).

## C.2. Hyperparameters and Implementation Details

For all experiments, we ran four parallel Markov chains, each consisting of $2{,}000$ warm-up iterations followed by $2{,}000$ sampling iterations.

**HMC Configuration.** We employ a standard HMC configuration with adaptive tuning during the warm-up phase, following the strategy used in the No-U-Turn Sampler (NUTS) (Hoffman et al., 2014). Specifically,

- **Step Size ($\eta$):** The step size is initialized at $\eta = 0.1$ and adapted during the warm-up phase using the dual-averaging algorithm, targeting an acceptance rate of $0.8$. The final value of $\eta$ is fixed after warm-up and used throughout the sampling phase.

- **Trajectory Length:** We fix the maximum number of leapfrog steps to $L_f = 50$. This choice is made empirically based on preliminary experiments, ensuring stable convergence as measured by $\widehat{R}$ and effective sample size (ESS). Given the tuned step size $\eta$, the trajectory length is determined as $T = L^\star \cdot \eta$ with $L^\star = \min\{\lceil T/\eta \rceil, L_f\}$.

- **Mass Matrix ($M$):** The mass matrix is initialized as the identity matrix and adapted during the warm-up phase using the empirical covariance of the momentum variables, in analogy with the mass-matrix adaptation employed in NUTS. The adapted mass matrix is fixed after warm-up.

**Proposed Method (PR-HMC) Configuration.** The proposed PR-HMC sampler employs a penalty-based extended potential $\widetilde{U}$ and reflective dynamics. Unless otherwise stated, the following hyperparameters are used throughout all experiments.

- **Geometric Constraint Parameters.** The eigenvalue lower bound is set to $c \in \{0.1, 0.01\}$ across experiments. The Frobenius-norm constraint is specified by $M = 20$, and the convex container radius is set to $R = 20$, enforcing $\|\boldsymbol{B}\|_F^2 \leq R^2$. The container constraint is chosen sufficiently loose so that it is inactive for the majority of trajectories. For the matrix von Mises–Fisher (mVMF) example, we use larger values $M = R = 40$ to accommodate the increased scale of the latent factors.

- **Constraint Aggregation** We define the individual constraint functions so that each constraint is satisfied when the corresponding function is nonpositive. Specifically, we use

$$g_{\text{eig}}(\boldsymbol{B}) = c - \lambda_{\min}\{\boldsymbol{B}^\top \boldsymbol{B}\}, \qquad g_{\text{F}}(\boldsymbol{B}) = \|\boldsymbol{B}\|_F^2 - M^2,$$

where $g_{\text{eig}}(\boldsymbol{B}) \leq 0$ enforces $\lambda_{\min}(\boldsymbol{B}^\top \boldsymbol{B}) \geq c$, and $g_{\text{F}}(\boldsymbol{B}) \leq 0$ enforces $\|\boldsymbol{B}\|_F^2 \leq M^2$. The exact joint violation would be

$$\max\{g_{\text{eig}}(\boldsymbol{B}), g_{\text{F}}(\boldsymbol{B})\}.$$

We replace this nonsmooth maximum by the smooth log-sum-exp approximation

$$\phi(\boldsymbol{B}) = \frac{1}{\beta_\phi} \log\left[\exp\{\beta_\phi g_{\text{eig}}(\boldsymbol{B})\} + \exp\{\beta_\phi g_{\text{F}}(\boldsymbol{B})\}\right].$$

Thus $\phi(\boldsymbol{B})$ serves as a smooth signed constraint-violation function: negative values indicate that both constraints are safely satisfied, whereas positive values indicate violation of at least one constraint. The sharpness parameter is set to $\beta_\phi = 10$.

- **Penalty-Based Extension Parameters.** We use a boundary-layer width $b = 0.2$ and penalty strength $\lambda = 2$. The positive-part operator is implemented via a scaled Softplus function with scale parameter $a = 10$, and the penalty growth function is chosen as a polynomial $\psi(z) = z^p$ with power $p = 4$.

## C.3. Detailed Results for Section 6.1 (mVMF Distribution)

we report the complete experimental results for the matrix von Mises–Fisher (mVMF) distribution on the Stiefel manifold $V_{r,u} = \{\mathbf{\Gamma} \in \mathbb{R}^{r \times u} : \mathbf{\Gamma}^\top \mathbf{\Gamma} = \mathbf{I}_u\}$ (Hoff, 2009; Pal et al., 2020). The target distribution is given by

$$\pi(\mathbf{\Gamma}) \propto \exp\big(\omega \operatorname{tr}(\mathbf{A}^\top \mathbf{\Gamma})\big), \qquad \mathbf{\Gamma} \in V_{r,u},$$

where $\mathbf{A} \in \mathbb{R}^{r \times u}$ is a fixed parameter matrix and $\omega > 0$ controls the concentration of the distribution around $\mathbf{A}$. Throughout the experiments, the reference matrix is fixed as

$$\mathbf{A} = [\mathbf{I}_u \,;\, \mathbf{0}_{(r-u) \times u}],$$

so that the target distribution concentrates around the canonical subspace spanned by the first $u$ coordinate axes. As $\omega$ increases, the distribution becomes increasingly concentrated, leading to challenging posterior geometries with sharp curvature near the mode.

**Log-probability used for Hamiltonian dynamics.** For the matrix von Mises–Fisher example, The log-probability evaluated at $q$ is given by

$$\log \pi(\mathbf{q}) = \kappa \operatorname{tr}(\mathbf{A}^\top \mathbf{\Gamma}(\mathbf{B})) - \frac{1}{2}\|\mathbf{B}\|_F^2 + \text{const}.$$

The Gaussian reference term $-\frac{1}{2}\|\mathbf{B}\|_F^2$ induces the Matrix Angular Central Gaussian structure after polar projection.

**Experimental design.** We consider ambient dimensions $r \in \{20, 40\}$ and vary the subspace dimension as

$$u \in \{2r/4,\, 3r/4,\, r - 1\},$$

covering moderate-, high-, and near-full-rank regimes. The concentration parameter is varied over

$$\omega \in \{10, 100, 200, 500\},$$

ranging from weakly to highly concentrated targets. For each configuration $(r, u, \omega)$, we run four independent Markov chains initialized from random orthonormal matrices. All samplers are run with identical computational budgets.

**Results.** Table 3 reports the full results of mVMF sampling for ambient dimensions $r = 20$ and $r = 40$. Across a broad range of subspace dimensions $u$ and concentration parameters $\omega$, the proposed PR-HMC method consistently achieves ESS values that are comparable to or higher than those of the baseline polar-expansion HMC, while maintaining PSRF values close to one. The advantage of PR-HMC becomes more pronounced as the subspace dimension $u$ approaches the ambient dimension $r$, where the target distribution exhibits increasing geometric degeneracy and strong curvature. In particular, in near-full-rank regimes (e.g., $u = r - 1$) with large concentration levels $\omega \in \{200, 500\}$, the baseline method often fails to converge or exhibits unstable behavior, whereas PR-HMC remains numerically stable and continues to mix effectively. These results highlight the robustness of PR-HMC in geometrically challenging settings where near-full-rank constraints and strong curvature dominate the target distribution.

## C.4. Detailed Results for Section 6.2 (Bayesian PCA)

This subsection reports the complete experimental results for the Bayesian PCA model introduced in Section 6.2, including configurations omitted from the main text.

**Model specification.** We consider the Bayesian PCA model

$$\mathbf{Y} = \mathbf{Z}\mathbf{R}\mathbf{\Gamma}^\top + \mathbf{E},$$

*Table 3.* Full mVMF sampling results for ambient dimensions $r = 20$ (left) and $r = 40$ (right). Effective Sample Size (ESS) and PSRF are reported across subspace dimensions $u$ and concentration parameters $\omega$. Entries marked as NA indicate configurations in which the baseline method did not yield valid estimates due to numerical instability. Larger ESS is better; for PSRF, values closer to 1.0 indicate convergence. Best per configuration in **bold**.

| | | $r = 20$ | | | | | | $r = 40$ | | | |
| | | BASELINE | | PR–HMC | | | | BASELINE | | PR–HMC | |
| $u$ | $\omega$ | ESS | PSRF | ESS | PSRF | $u$ | $\omega$ | ESS | PSRF | ESS | PSRF |
|---|---|---|---|---|---|---|---|---|---|---|---|
| | 10 | **1134.85** | 1.003 | 1109.44 | 1.003 | | 10 | 626.68 | 1.006 | **640.42** | 1.006 |
| 10 | 100 | **1098.47** | 1.003 | 1092.16 | 1.004 | 20 | 100 | **1432.54** | 1.004 | 1343.78 | 1.003 |
| | 200 | 1183.51 | 1.003 | **1212.40** | 1.003 | | 200 | 1351.09 | 1.005 | **1379.87** | 1.005 |
| | 500 | 1188.72 | 1.004 | **1216.52** | 1.002 | | 500 | 1046.10 | 1.005 | **1054.42** | 1.006 |
| | 10 | 1034.71 | 1.003 | **1092.72** | 1.003 | | 10 | 1214.27 | 1.002 | **1292.33** | 1.003 |
| 15 | 100 | 1174.36 | 1.003 | **1252.20** | 1.003 | 30 | 100 | 994.89 | 1.003 | **1056.54** | 1.003 |
| | 200 | **1193.32** | 1.004 | 1187.52 | 1.002 | | 200 | NA | NA | **1246.13** | 1.003 |
| | 500 | 906.93 | 1.003 | **1096.59** | 1.002 | | 500 | NA | NA | **1219.83** | 1.009 |
| | 10 | 1293.52 | 1.001 | **1449.52** | 1.001 | | 10 | 1352.64 | 1.002 | **1475.80** | 1.002 |
| 19 | 100 | 1179.94 | 1.003 | **1495.54** | 1.001 | 39 | 100 | 459.30 | 1.013 | **1162.82** | 1.002 |
| | 200 | 912.72 | 1.025 | **1492.98** | 1.001 | | 200 | 579.38 | 1.011 | **1331.58** | 1.002 |
| | 500 | 866.503 | 1.006 | **1463.00** | 1.002 | | 500 | 900.54 | 1.004 | **1522.68** | 1.002 |

where $Y \in \mathbb{R}^{n \times r}$ denotes the observed data matrix, $\Gamma \in V_{r,u}$ is an orthogonal loading matrix on the Stiefel manifold, $Z \in \mathbb{R}^{n \times u}$ represents latent scores drawn from a standard normal distribution, and $E \in \mathbb{R}^{n \times r}$ is isotropic Gaussian noise. The observed data are centered prior to inference.

The likelihood is invariant under right-orthogonal rotations of $\Gamma$. To resolve this rotational non-identifiability, we fix $R$ to be a diagonal scaling matrix with distinct ordered entries. A Matrix Angular Central Gaussian (MACG) prior (Kent et al., 2013) is placed on $\Gamma$, which is a natural prior distribution on the Stiefel manifold.

**Log-probability used for posterior sampling.** The log-probability defining the Hamiltonian dynamics is given by

$$\log \pi(q) = -\frac{1}{2} \operatorname{tr}\big(B^{\top} \Phi^{-1} B\big) + \frac{\beta}{2} \operatorname{tr}\big(\Gamma(B)^{\top}(YY^{\top})\Gamma(B)\big) + \text{const},$$

where $\Phi \in \mathbb{R}^{u \times u}$ is a positive-definite prior precision matrix, and $\beta > 0$ controls the strength of the likelihood term.

**Experimental design.** We fix the subspace dimension at $u = 2$ and vary the ambient dimension $r \in \{10, 20, 30, 40\}$. The number of observations is set to $n = 30$, and the scaling matrix is fixed as $R = \operatorname{diag}(5, 0.5)$ to induce distinct signal strengths. The prior precision matrix is chosen as $\Phi = \operatorname{diag}(1, \varepsilon)$, where $\varepsilon \in \{0.1, 0.05, 0.01\}$ controls the degree of anisotropy. The observation noise variance is fixed to $\sigma_Y^2 = 1.0$, and the likelihood scaling parameter is set to $\beta = 1/100$. These values are kept fixed across all experiments. To account for data-generation variability, each configuration is replicated over 30 independently generated datasets.

**Results.** Table 4 reports the full results for all configurations. Under moderate anisotropy ($\varepsilon \in \{0.1, 0.05\}$), both the baseline method and PR-HMC achieve comparable ESS values across all ambient dimensions $r$ and for both subspace dimensions $u = 2$ and $u = 4$. In contrast, under high anisotropy ($\varepsilon = 0.01$), the baseline method often fails to yield valid estimates due to numerical instability, whereas PR-HMC remains stable and consistently attains high ESS values across all reported settings. These results highlight the robustness of PR-HMC in highly anisotropic configurations, where standard HMC-based approaches encounter severe numerical difficulties.

For the largest ambient dimension ($r = 40$), we observe that the PSRF values of both methods exceed the commonly used convergence threshold of 1.05 across all anisotropy levels, suggesting that convergence diagnostics should be interpreted with caution in this setting. Nevertheless, PR-HMC consistently achieves higher ESS than the baseline method relatively. We note that these experiments are conducted in a setting where the sample size $n$ is smaller than the ambient dimension $r$, and we expect that increasing $n$ would further improve the PSRF behavior while preserving the relative advantage of PR-HMC.

*Table 4.* Full results for Bayesian PCA with orthogonal loadings. Effective Sample Size (ESS) and PSRF are reported for subspace dimensions $u = 2$ (left) and $u = 4$ (right) across ambient dimensions $r$ and anisotropy levels $\varepsilon$, where smaller $\varepsilon$ indicates stronger anisotropy. Entries marked as NA indicate configurations in which the baseline method did not yield valid estimates due to numerical instability. Larger ESS is better; for PSRF, values closer to 1.0 indicate convergence. Best per configuration in **bold**.

| | | u = 2 | | | | u = 4 | | | |
| | | Baseline | | PR-HMC | | Baseline | | PR-HMC | |
| $r$ | $\varepsilon$ | ESS | PSRF | ESS | PSRF | ESS | PSRF | ESS | PSRF |
|---|---|---|---|---|---|---|---|---|---|
| | 0.1 | 1066.73 | 1.0014 | **1075.97** | 1.014 | **1066.47** | 1.007 | 1065.51 | 1.006 |
| 10 | 0.05 | 1248.59 | 1.003 | **1258.29** | 1.003 | 1316.11 | 1.002 | **1326.73** | 1.002 |
| | 0.01 | NA | NA | **1061.04** | 1.007 | NA | NA | **942.44** | 1.007 |
| | 0.1 | 964.89 | 1.046 | **968.74** | 1.043 | 985.85 | 1.011 | **989.29** | 1.011 |
| 20 | 0.05 | 1109.68 | 1.011 | **1128.09** | 1.011 | **1069.84** | 1.009 | 1053.09 | 1.009 |
| | 0.01 | NA | NA | **989.37** | 1.012 | NA | NA | **776.15** | 1.018 |
| | 0.1 | 995.71 | 1.031 | **1011.71** | 1.044 | 880.57 | 1.038 | **886.78** | 1.036 |
| 30 | 0.05 | **1073.04** | 1.026 | 1067.64 | 1.040 | **1029.08** | 1.030 | 1008.60 | 1.027 |
| | 0.01 | NA | NA | **977.32** | 1.042 | NA | NA | **866.23** | 1.035 |
| | 0.1 | 1010.57 | 1.081 | **1027.61** | 1.073 | 829.36 | 1.057 | **852.95** | 1.058 |
| 40 | 0.05 | 993.40 | 1.092 | **1022.38** | 1.099 | 957.89 | 1.057 | 932.71 | 1.068 |
| | 0.01 | NA | NA | **921.12** | 1.089 | NA | NA | **830.42** | 1.056 |

# D. Application to Non-Smooth Domains

The penalty-based construction in Appendix A assumes a $C^2$ boundary for $\partial K$. In this section we demonstrate that PR-HMC can be applied to *non-smooth* domains—specifically, polytopes—by replacing the exact constraint indicator with a smooth surrogate.

**Smooth surrogate for polytope constraints**    A polytope $K = \{x \in \mathbb{R}^d : a_j^\top x \le b_j, \ j = 1, \dots, J\}$ has a piecewise-linear boundary that is not $C^2$ at edges and vertices. We use a two-step smooth approximation of the polytope violation. First, we replace the nonsmooth maximum of the half-space violations by a log-sum-exp approximation:

$$\phi(q) = \frac{1}{\beta_\phi} \log\left( \sum_{j=1}^{J} \exp\big(\beta_\phi(a_j^\top q - b_j)\big) \right).$$

Here $\beta_\phi > 0$ controls the sharpness of the approximation, and

$$\phi(q) \to \max_{1 \le j \le J}(a_j^\top q - b_j) \qquad \text{as } \beta_\phi \to \infty.$$

Thus $\phi_\beta$ serves as a smooth signed constraint surrogate for the polytope. Second, we apply a softplus positive-part approximation to obtain the smooth boundary-violation surrogate

$$\varphi_a(q) = \frac{1}{a} \log\{1 + \exp(a\phi(q))\}.$$

This approximates the positive part

$$\max\{\phi(q), 0\},$$

so that violations of the polytope constraint are penalized smoothly. Finally, we use the quartic penalty

$$\lambda \psi\left( \frac{\varphi_a(q)}{b} \right) = \lambda \left( \frac{\varphi_a(q)}{b} \right)^4,$$

as in Appendix A. The resulting extended potential $\widetilde{U}$ is $C^\infty$ on the convex container $\widetilde{K}$, so reflective HMC is well defined.

**Experimental setup.** We consider two standard polytopes in dimension $d = 100$:

- **Hypercube:** $K = \{x \in \mathbb{R}^d : -1 \le x_i \le 1\}$, with circumradius $\sqrt{d}$.
- **Simplex:** $K = \{x \in \mathbb{R}^d : x_i \ge 0, \ \sum_i x_i \le 1\}$, with circumradius 1.

In both cases, the target is a truncated standard Gaussian $\pi(x) \propto \exp(-\|x - x_c\|^2/2)\, \mathbf{1}\{x \in K\}$, centered at the Chebyshev center $x_c$ of $K$ (the origin for the cube; $(1/(d+1), \ldots, 1/(d+1))$ for the simplex). The convex container is the circumscribing Frobenius ball. We set the penalty parameters to $\lambda = 2$, $b = 0.2$, the log-sum-exp sharpness to $\beta_\phi = 10$, and the softplus scale to $a = 10$. The leapfrog integrator uses $L_f = 50$ steps with step size adaptively tuned during warm-up targeting an acceptance rate of $0.65$.

We compare PR-HMC with two random-walk baselines from the HOPS library (Jadebeck et al., 2021): Hit-and-Run (H&R) and Coordinate Hit-and-Run (CH&R). All methods use 4 parallel chains of 2 000 post-burn-in samples (2 000 burn-in iterations discarded). We report the effective sample size (ESS) averaged over coordinates and the maximum split-$\hat{R}$ across coordinates.

**Results.** Table 5 reports the results for $d = 100$. PR-HMC is the only method that achieves convergence (PSRF $< 1.05$) at this dimension, with ESS exceeding 2 500 on both polytopes. The random-walk baselines exhibit ESS values below 50 and PSRF well above the 1.05 threshold, indicating severe mixing failure. These results demonstrate that the smooth-surrogate strategy extends PR-HMC to nonsmooth constrained domains without sacrificing sampling efficiency. The large performance gap over random-walk methods suggests that gradient-based exploration via reflective HMC can be substantially more effective in high-dimensional constrained spaces, even when the original domain has non-smooth boundaries.

*Table 5.* Sampling performance on 100-dimensional polytopes. Larger ESS is better; PSRF closer to 1.0 indicates convergence. Best results in **bold**.

|  | 100-Cube | | 100-Simplex | |
| --- | --- | --- | --- | --- |
| Method | ESS | PSRF | ESS | PSRF |
| PR-HMC (ours) | **2568** | **1.007** | **2742** | **1.004** |
| CH&R-HOPS | 49 | 1.257 | 37 | 1.356 |
| H&R-HOPS | 14 | 2.190 | 12 | 2.756 |

# E. Empirical Dimension Scaling of Mixing Time

Theorem 4.3 predicts that the mixing time of ReHMC scales as $O(d^2)$ in the leading dimension dependence. In this section we verify this prediction empirically on a truncated Gaussian benchmark where the dimension can be varied over a wide range.

**Experimental setup.** We consider the target density

$$\pi(x) \ \propto \ \exp\!\left(-\frac{\|x - \mu\|^2}{2\sigma^2}\right) \mathbf{1}\{x \in [-1, 1]^d\},$$

with $\mu = (0.5, \ldots, 0.5) \in \mathbb{R}^d$ and $\sigma = 1.5$. The off-center mean ensures that the target is not symmetric with respect to the domain, providing a nontrivial test case. We vary the dimension over $d \in \{5, 10, 20, 40, 100, 200\}$.

For each $d$, we run four parallel chains of PR-HMC initialized near different corners of $[-1, 1]^d$ to ensure diverse starting points. The step size is set to $\eta = 0.3/d$, consistent with the step-size condition $\eta \lesssim d^{-1}$ in Theorem 4.3. The convex container is the circumscribing ball of radius $\sqrt{d}$, with penalty parameters $\lambda = 2$ and $b = 0.2$.

**Convergence criterion.** The theoretical mixing time is defined as the iteration count at which the total variation distance to the target falls below a threshold, but this quantity is not directly computable in practice. As a proxy, we define the empirical mixing time $k$ as the smallest iteration count at which the maximum coordinate-wise split-$\hat{R}$ (Vehtari et al., 2021) falls below 1.05. Taking the maximum over coordinates provides a conservative (worst-case) assessment, as it is governed by the slowest-mixing coordinate. To estimate the dimension exponent, we perform a least-squares fit of $\log k$ against $\log d$.

**Results.** Figure 2 displays the log-log plot of the empirical mixing time $k$ versus dimension $d$. The fitted slope is approximately 1.94, which is close to the theoretical exponent of 2. For reference, the $d^2$ and $d^1$ scaling lines are also shown; the observed data points track the $d^2$ reference closely across nearly two orders of magnitude in dimension. This confirms that the $O(d^2)$ dependence predicted by Theorem 4.3 is empirically tight in this setting.

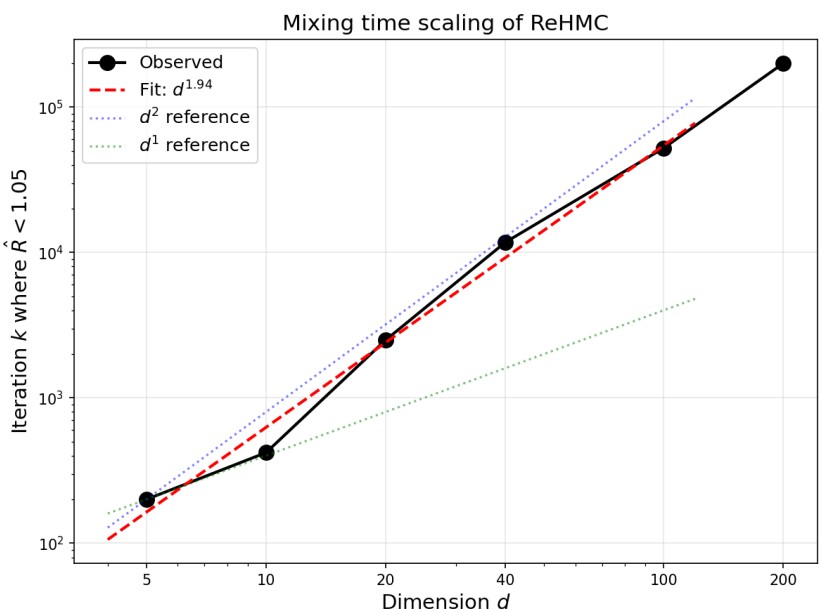

*Figure 2.* Log-log plot of the empirical mixing time $k$ (iteration at which max split-$\hat{R} < 1.05$) versus dimension $d$ for the truncated Gaussian target on $[-1,1]^d$. The fitted slope of 1.94 (red dashed) is consistent with the $O(d^2)$ theoretical bound (blue dotted). The green dotted line shows $d^1$ scaling for comparison.

## F. Effect of the Reflecting Boundary on Sampling Efficiency

In the convex-container-plus-thinning framework of Section 3, the reflecting boundary of $\widetilde{K}$ prevents trajectories from drifting far outside the original support $K$. An alternative is to dispense with the bounded container and to rely solely on the penalty term to concentrate mass near $K$, effectively taking $\widetilde{K} = \mathbb{R}^d$. This unbounded penalty-only construction is possible in principle, but without a reflecting boundary the penalty alone must keep the chain near $K$. In this appendix, we compare the bounded-container and unbounded penalty-only approaches on a synthetic nonconvex domain.

**Experimental setup.** We define a non-convex domain in $\mathbb{R}^{10}$ via hyperspherical coordinates $(r, \theta_1, \ldots, \theta_{d-1})$ as

$$K = \{x \in \mathbb{R}^d : \|x\| \le b(\theta)\}, \qquad b(\theta) = b_0\left(1 + \varepsilon\,\sin\left(\textstyle\sum_{j=1}^{d-1} k_j\,\theta_j\right)\right),$$

where $d = 10$, $b_0 = \sqrt{d}$, $\varepsilon = 0.3/\sqrt{d}$, and $k_j = 3$ for all $j$. The target density is $\pi(x) \propto \exp(-\|x\|^2/2\sigma^2)\,\mathbf{1}\{x \in K\}$ with $\sigma \in \{10, 30\}$; larger $\sigma$ yields a flatter target that places more mass near the boundary of $K$.

We compare the following approaches, all using the penalty-based extended potential of Appendix A with softplus scale $a = 10$, boundary width $b = 0.2$, and quartic penalty $\psi(s) = s^4$:

- **Bounded** ($\lambda = 2$): PR-HMC on the circumscribing ball $\widetilde{K} = \{x : \|x\| \le b_0(1 + \varepsilon)\}$ with reflection.
- **Unbounded** ($\lambda = 2$): HMC on $\mathbb{R}^d$ with the same mild penalty but no reflecting boundary.
- **Unbounded** ($\lambda \in \{10, 100\}$): HMC on $\mathbb{R}^d$ with stronger penalties to compensate for the absence of a reflecting boundary.

All methods use 4 parallel chains of 2 000 post-burn-in samples (2 000 burn-in), with leapfrog length $L_f = 50$ and step size adaptively tuned to a target acceptance rate of 0.65.

**Results.** Table 6 reports the ESS and retention rate for $\sigma \in \{10, 30\}$. At the same penalty strength ($\lambda = 2$), the bounded method achieves both higher ESS and higher retention rate than the unbounded alternative: for $\sigma = 10$, bounded attains ESS $= 392$ with retention 0.756, compared to ESS $= 344$ and retention 0.632 for unbounded. The reflecting boundary accounts for this improvement, as it prevents trajectories from drifting far outside $K$ without requiring a strong penalty.

Increasing $\lambda$ in the unbounded setting does raise the retention rate, but at a significant cost to ESS: at $\lambda = 100$, the retention rate approaches 0.98 but ESS drops to 244, well below the bounded method's 392. This is because a stronger penalty worsens the condition number of the extended potential, forcing the adaptive step size to shrink and reducing sampling efficiency. The results are consistent across $\sigma = 10$ and $\sigma = 30$, confirming that the advantage of the bounded approach is robust.

*Table 6.* Bounded vs. unbounded extensions on the wavy ball domain ($d = 10$). Larger ESS is better; retention rate closer to 1 is better. Best ESS per $\sigma$ in **bold**.

| $\sigma$ | Setting | $\lambda$ | ESS | Ret. Rate |
|---|---|---|---|---|
| 10 | Bounded | 2 | **392.19** | 0.756 |
| | Unbounded | 2 | 343.75 | 0.632 |
| | Unbounded | 10 | 299.95 | 0.806 |
| | Unbounded | 100 | 244.09 | 0.983 |
| 30 | Bounded | 2 | **388.65** | 0.755 |
| | Unbounded | 2 | 342.05 | 0.630 |
| | Unbounded | 10 | 295.75 | 0.804 |
| | Unbounded | 100 | 238.94 | 0.984 |

**Intuition in one dimension.** To provide further intuition, we consider a flat target $\pi(x) \propto \mathbf{1}\{x \in K\}$ on a disconnected domain $K = [-0.9, -0.3] \cup [0.3, 0.9]$ (Figure 3). We compare two approaches under the same $\lambda$: one that extends $\pi$ to $\widetilde{K} = [-0.9, 0.9]$ with reflection, and one that extends to all of $\mathbb{R}$ without reflection. The top panel shows the extended density $\tilde{p}(x) \propto e^{-\widetilde{U}(x)}$: without reflection, the mass spreads over $\mathbb{R}$ (red), whereas with reflection it is truncated at $|x| = 0.9$ (blue), concentrating more mass within $K$. The bottom panel confirms this via sample histograms: the bounded chain retains 67.8% of samples in $K$ compared to 47.1% for the unbounded chain. This illustrates that when the target density does not decay near the boundary of $K$, a mild penalty alone results in substantial mass leakage outside $K$, which the reflecting boundary prevents.

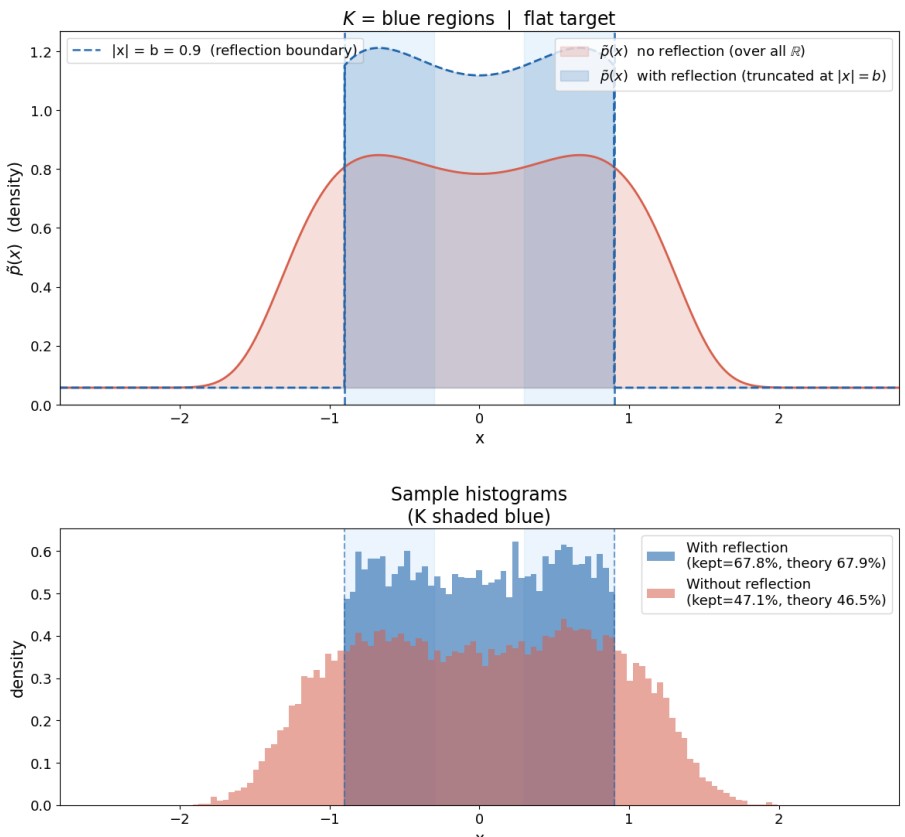

*Figure 3.* One-dimensional comparison of bounded (blue) and unbounded (red) extensions for a flat target on $K = [-0.9, -0.3] \cup [0.3, 0.9]$. Top: extended densities. Bottom: sample histograms with retention rates.

