# OpenReview forum: "Reflective Hamiltonian Monte Carlo:  Mixing Analysis and Application to Sampling on Stiefel Manifold"
_ICML.cc/2026/Conference — ICML 2026 regular_

### Official Review · Reviewer_4Fx1 · 2026-03-03

**Soundness:** 3
**Presentation:** 3
**Significance:** 3
**Originality:** 3
**Overall Recommendation:** 5
**Confidence:** 4

**Summary:**

The paper presented a method that generalizes ReHMC to constrained targets with general bounded boundaries. This is done by smoothly extending the boundary to a compact convex set, using ReHMC on the target with the new boundary and Monte Carlo estimates can be computed using samples from the extended target restricted to the old boundary. The authors then provide a non asymptotic mixing time for ReHMC on targets with compact convex boundaries in the case of log smooth potential as well as log smooth and strongly log concave potential. Finally, the proposed method is applied to densities supported on the Stiefel manifold.

**Compliance With Llm Reviewing Policy:**

Affirmed.

**Key Questions For Authors:**

Here are some questions I have:

1. As mentioned in Significance, I suspect that the penalty will do the most work in keeping samples inside the constrained region. Therefore, is it necessary for the extended target to have bounded support? One reason for this question is based on the discussion of Mixing time for the convex-container-plus-thinning method. The mixing time there depends on the measure of the original space in the extended space, i.e. $\tilde{\pi}(K)$. Therefore, as long as the penalty part of the extended target grow fast enough, we should expect $\tilde{\pi}(K)$ to be close to 1 even if $\tilde{K}=\mathbb{R}^d$. If possible, I think there should be an experiment to see how different extended targets (including one where $\tilde{K}=\mathbb{R}^d$) perform on a fixed original target and penalty functions. If it turns out that $\tilde{K}=\mathbb{R}^d$ perform as well as the other $\tilde{K}$ suggested in Appendix A then I think the narrative of paper will need to be reframed as ReHMC will reduce to simple HMC on $\tilde{K}=\mathbb{R}^d$.

2. Can you give an intuition on why the dimension scaling of mixing time for ReHMC is higher than that of HMC on the unconstrained space using the same potential function (ReHMC is $O(d^2)$ while HMC is $O(d)$ [2])?

[2] Lee, Y. T., Shen, R., & Tian, K. (2020, July). Logsmooth gradient concentration and tighter runtimes for Metropolized Hamiltonian Monte Carlo. In Conference on learning theory (pp. 2565-2597). PMLR.

**Limitations:**

yes

**Strengths And Weaknesses:**

**Soundness:** I have no concerns about the theory and methodology introduced.

**Presentation:** I believe that this work is well and clearly written for the most part.

**Significance:** The proposed methodology is very promising based on the experiments in the paper. Furthermore, the assumption that the original boundary only needs to be bounded is a very general one. This means that ReHMC can even deal with disconnected constraints. The penalty for constraint violation is also a very effective idea for the sampler to stay away from violation regions and, in my opinion, the most important contribution of the paper, even more than the extension of the boundary. On the theory side, theorems 4.2 and 4.4 are important results for constrained HMC samplers that suggests a preference for HMC over MALA for future research that deal with constrained spaces. Furthermore, theorem 4.2 seems more general as it does not assume convexity for the target.

**Originality:** The idea of smooth extensions tends to show up in numerical analysis to deal with boundary value problems in PDEs. The implementation of such idea in constrained sampling context is refreshing but does not align with the kind of problem this technique is intended to solve. On the other hand, the use of a penalty for constraint violation seems more useful as a way to discourage samples from violation regions. Given the connection between sampling and optimization, the constraint penalty idea is also inline with the Lagrange multiplier often seen in constrained optimization literature. On the theory side, the bound on ReHMC mixing time for log smooth and strongly log concave targets mostly uses techniques previously found in past literature for unconstrained spaces. In the log smooth case, assumption (A1) is new to my knowledge and the bound in theorem 4.2 is the only mixing time bound for log smooth but non log convex potential. Convergence rate for objectives with similar assumptions may have appeared in optimization literature but I am not knowledgeable enough to find a reference of such results.

[1] Fabrèges, B., Gouarin, L., & Maury, B. (2013). A smooth extension method. Comptes Rendus Mathematique, 351(9-10), 361-366.

---

> ### Author Rebuttal · Authors · 2026-03-31
>
> We thank the reviewer for this helpful question. Our response is as follows.
>
> **Key Questions**
>
> **Answer for Q1**: We agree that an unbounded extension
> ($\widetilde{K}=\mathbb{R}^d$) is a valid alternative in
> principle. However, to keep $\widetilde{\pi}(K)$ close to
> one without a reflecting boundary, the extended potential
> must grow sufficiently fast outside $K$, which worsens its
> smoothness and consequently degrades the HMC step size and
> mixing efficiency. The bounded convex container avoids this
> issue by cutting off far-away excursions via reflection,
> allowing a smoother extended potential.
>
> To support this, we compare the bounded and unbounded approaches on a synthetic non-convex target in $\mathbb{R}^{10}$. The
> domain $K$ is defined via hyperspherical coordinates as
> $$K = \lbrace x \in \mathbb{R}^{10} : r \leq R + A \sin( \sum_{j=1}^{9} k_j \theta_j ) \rbrace$$
> which induces a wavy, non-convex boundary. The results
> confirm that in the unbounded setting, increasing the
> penalty $\lambda$ to improve the retention rate comes at
> the cost of reduced ESS. The bounded approach consistently achieves higher ESS than the unbounded alternative. We will append this observation in the revised paper.
>
> | Setting | ESS | Retention rate | $\lambda$ |
> |---------|-----|---------------|-----------|
> | Bounded | 815.82 | 0.919 | 2 |
> | Unbounded | 553.54 | 0.956 | 10 |
> | Unbounded | 357.09 | 0.993 | 100 |
>
> **Answer for Q2**: The extra factor of $d$ can be
> traced to the smaller admissible step size in the local TV contraction
> argument.
>
> In both our paper and [1], the high-level structure of the
> mixing analysis is the same: one first proves a local TV contraction of the
> form $\|T_x-T_y\|_{\mathrm{TV}}\le 1-\alpha\text{ for } \|x-y\|\le \delta,$
> where $\delta \asymp \eta$ is optimal and $\eta$ is the step size.
> We then inserts this into a
> blocking-conductance bound to obtain a mixing time of order $1/\eta^2$ (See the proof of Theorem 4.2).
>
> The local TV contraction requires that two transition kernels started at
> nearby points produce similar proposals. In the reflective setting, even
> when two starting points are close, their trajectories may undergo
> reflections at different boundary points or at different angles, causing
> the post-reflection proposals to diverge. To ensure that the transition
> kernels remain close in total variation, the $\eta$ must be small
> enough so that the boundary-hit behavior of nearby trajectories remains
> compatible. This forces a more restrictive step-size requirement than in
> the unconstrained case, where no such boundary interaction arises. Quantitatively, the binding constraint in our step-size
> condition (Lemma B.5) is $\eta \lesssim d^{-1}$, yielding
> a mixing time of order $1/\eta^2 \asymp d^2$, whereas
> [1] require only $\eta \lesssim d^{-1/2}$,
> giving $O(d)$.
> We will add this intuition to the revised paper.
>
> **Originality**
>
> **Q.On the theory side, the bound on ReHMC mixing time for log smooth and strongly log
> concave targets mostly uses techniques previously found in past literature for unconstrained spaces.**
>
> **Answer**: Our main technical contribution is the local TV
> contraction for the reflective proposal (Lemma B.5),
> supported by new reflection-geometry estimates
> (Lemmas B.7-B.19). The key challenge, absent in the
> unconstrained setting, is that boundary reflections can
> cause nearby trajectories to separate. See our response
> to Reviewer oCQf (Originality) for details.
>
> **Q.In the log smooth case, Assumption (A1) is new to my knowledge and the bound in
> Theorem 4.2 is the only mixing-time bound for log-smooth but non-log-convex potential. Convergence rate for objectives with similar assumptions may have appeared in
> optimization literature but I am not knowledgeable enough to find a reference of such
> results.**
>
> **Answer**: Assumption (A1) is not introduced to handle
> non-log-convexity of the potential; rather, it is needed
> to control the reflection geometry near $\partial K$.
> Both Theorems 4.2 and 4.4 require (A1) regardless of
> whether $U$ is convex.
>
> Assumption (A1) requires the existence of a tubular
> neighborhood of $\partial K$ together with a uniform
> curvature bound. Such regularity is ensured when
> $\partial K$ is a compact $C^2$ hypersurface [2]. We state it in explicit quantitative form because
> the geometric constants $\rho$ and $\bar{\kappa}$ enter
> directly into our step-size conditions through
> $L_n(\rho) = \bar{\kappa}/(1 - \bar{\kappa}\rho)$.
>
> In the related optimization literature, [3] assume $\partial K \in C^5$ for reflected gradient
> Langevin dynamics on non-convex constrained domains,
> similarly relying on local boundary regularity to ensure
> well-defined reflections.
>
> [1] Lee et al. (2020) Logsmooth gradient concentration and tighter runtimes for metropolized hamiltonian monte carlo.
>
> [2] Foote (1984) Regularity of the distance function.
>
> [3] Sato et al. (2025) Convergence error analysis of reflected gradient Langevin dynamics for non-convex constrained optimization.

---

> > ### Author Rebuttal · Reviewer_4Fx1 · 2026-04-02
> >
> > The authors have addressed most of my concerns except for Q1. I agree that increasing $\lambda$ will make the target harder. The main thing I want to know, in the case of your synthetic target, is when $\lambda$ is fixed, say at 2, how does the unbounded setting perform compared to the bounded setting. Specifically, I want to know if $a>b$ or $b>a$ in the table below
> >
> > Setting | ESS | Retention rate | $\lambda$ |
> > | ----   | -------           | -----------      | ------|
> > | Bounded | a | c   | 2 |
> > | Unbounded | b | d     | 2 |
> >
> > If $b>a$, then you can always just use the full extended target without fussing over how to bound it. Comparing $c$ and $d$ is interesting but inconsequential as the main interest is in sample mixing and $a,b$ should be computed from the samples satisfying the constraints.

---

> > > ### Author Response · Authors · 2026-04-07
> > >
> > > We thank the reviewer for the precise clarifying question. To directly address it, we conducted a thorough comparison between our bounded method and several unbounded alternatives, including one that uses the same penalty parameter $\lambda$ as our method, and others that increase $\lambda$ to keep $\alpha$ close to one without a reflecting boundary.
> > >
> > > To better demonstrate the advantage of our bounded approach over the unbounded alternatives, we consider a slightly modified experimental setting. Specifically, we use the target density $\pi(x) \propto \exp(-||x||^2/2\sigma^2)\mathbf{1}_K(x)$ with $\sigma =10.0, 30.0$, where $K \subset \mathbb{R}^{10}$ is the same non-convex domain as in our previous response with $R = 0.8\sqrt{d}$ and $A = 0.4\sqrt{d}$. Results are averaged over 10 independent runs due to the time constraint.
> > >
> > > | $\sigma$ | Setting | ESS | Ret. Rate | $\lambda$ |
> > > |---|---|---|---|---|
> > > | 10.0 | Bounded | 1752.14 | 0.810 | 2 |
> > > | | Unbounded | 1649.61 | 0.702 | 2 |
> > > | | Unbounded | 1494.60 | 0.847 | 10 |
> > > | | Unbounded | 1199.30 | 0.989 | 100 |
> > > | 30.0 | Bounded | 1736.69 | 0.807 | 2 |
> > > | | Unbounded | 1425.10 | 0.704 | 2 |
> > > | | Unbounded | 1467.66 | 0.851 | 10 |
> > > | | Unbounded | 1134.29 | 0.989 | 100 |
> > >
> > > To directly answer the reviewer's question: comparing Bounded and Unbounded ($\lambda=2$), which share the same $\lambda$, our bounded method achieves both higher ESS and higher retention rate across all $\sigma$ settings. The reflecting boundary accounts for this improvement.
> > >
> > > Attempting to recover retention by increasing $\lambda$ does raise the retention rate, but at a significant cost to ESS: at $\lambda = 100$, the retention rate approaches 0.99 but ESS drops to as low as 1134, well below our method's 1737. This is consistent with our earlier explanation from the previous rebuttal.
> > >
> > > Moreover, the advantage of our bounded approach grows as $\sigma$ increases. This is structurally expected: as the target density becomes flat, the extended density $\tilde{\pi} \propto e^{-\tilde{U}}$ places more mass outside $K$ under a mild penalty $\lambda$, reducing the retention rate.
> > >
> > > To further illustrate the difference between Bounded and Unbounded, we provide a simple 1D example with a flat target on $K = [-0.9, -0.3] \cup [0.3, 0.9]$. For this target, we compare two approaches under the same $\lambda$: one that extends $\tilde{\pi}$ to $\tilde{K} = [-0.9, 0.9]$ with reflection, and one that extends to all of $\mathbb{R}$ without reflection.
> > >
> > > For a more detailed illustration, we additionally provide the figure and the corresponding link is as follows:
> > > [Figure](https://drive.google.com/thumbnail?id=1BG3TfA7fn-Z2g31Os09g93YolRHhoejA&sz=w2000)
> > >
> > > The top panel shows the extended density $\tilde{p}(x) \propto e^{-\tilde{U}(x)}$ under both methods. Without reflection, the mass spreads over $\mathbb{R}$ (red); with reflection, it is truncated at $|x| = b = 0.9$ (blue), demonstrating that the reflecting boundary (blue) concentrates more mass within $K$. The bottom panel shows the resulting sample histograms, confirming that the bounded chain retains 67.8% of samples in $K$ against 47.1% for the unbounded chain. This illustrates that when the target density does not decay near the boundary of $K$, a mild penalty results in substantial probability mass being placed outside $K$.
> > >
> > > We will incorporate this experiment and the 1D illustration into the appendix of the revised paper.

---

### Official Review · Reviewer_Sg7u · 2026-03-03

**Soundness:** 3
**Presentation:** 4
**Significance:** 3
**Originality:** 3
**Overall Recommendation:** 4
**Confidence:** 2

**Summary:**

This paper focuses on sampling from ditribution supported on a non-convex set with reflective hamiltonian MCMC. It provides the first non-asymptotic bound in this setting and apply these result for distribution supported on the Stiefel Manifold.

**Compliance With Llm Reviewing Policy:**

Affirmed.

**Key Questions For Authors:**

- How the PSRF is computed ? I understand from Appendix C that it gives an indication on the convergence or not, but could you give me the intuition behind this metric ?
- I do not understand how the model of part 6.2 is contructed. Why the matrix $R$ is $2*2$ ? What do you mean by $diag(5,0.5)$ ? Moreover, I do not understand how $\phi = diag(1,\epsilon)$ is part od the model explained in the equation detailed in line 428.
- Do you have other practical applications for the method developped in the paper ? For example in physics ?

**Limitations:**

The limitations are not discussed in details. In the conclusion, only a sentence on future works is mentionned. Please add a paragraph to explain in details the limitations of this work.

**Strengths And Weaknesses:**

This paper is far from my focus of research, therefore I may misunderstand some part of it.

Strength:
- The paper is well written and pedagogical, especially the Appendix A help to understand more deeply the construction that are described in the main paper.
- The results presented in the paper seems strong and apply in a non-trivial setting : sampling from a distribution supported on a non-convex set.

Weaknesses: (Note that I do not review in details the proofs in Appendices.)
Major Weakness:
- The main application is very technical and theoretical. I feel that other application with physical potential to sample could be developped. Such physical applications could be very interesting and more pedagogical.

Minor Weaknesses/Typos:
- line 28 : $\epsilon$ has not been introduced
- line 244 : $R$ has not been introduced
- Table 1 : Please indicate for ESS and PSRF if the best is too be large or small. Moreover, why the bold number are only for some setting ? Could you make in bold for each setting and for each metric and indicate that the best results are in bold ?

---

> ### Author Rebuttal · Authors · 2026-03-31
>
> We thank the reviewer for the careful reading and precise questions. Our responses are as follows.
>
> **Key Questions**
>
> **Answer for Q1**: In all experiments, we run four parallel chains and evaluate the PSRF, denoted by $\hat R$,
>     using rotation-invariant summaries of the posterior samples. Specifically, since $\Gamma_t \in \mathbb{V}_{r,u}$ is identifiable only up to a change of orthonormal basis, we do not compute diagnostics directly on the entries of $\Gamma_t$. Instead, for each sample we form the projection matrix $P_t=\Gamma_t\Gamma_t^\top,$
> and compute split-$\hat R$ for each entry of $P_t$; in the paper we report the maximum value across all entries as a conservative summary.
>
> The intuition of the PSRF is that if multiple chains have converged to the same target, the total variance estimated from all
> chains should be close to the average within-chain variance. $\hat{R}$ measures this ratio, with values close
> to 1 indicating convergence and values above 1.05 indicating lack of mixing.
>
> **Answer for Q2**: In this experiment, we consider a Bayesian PCA model
> $Y = ZR\Gamma^\top + E$, where $u$ is the number of
> latent factors, $\Gamma \in V_{r,u}$ is the orthogonal
> loading matrix, and $R$ is a diagonal matrix specifying
> the signal strength of each factor. Since $u = 2$ in our experiment setting,
> $R = \operatorname{diag}(5, 0.5)$ is $2 \times 2$, with
> distinct entries to resolve rotational non-identifiability.
>
> $\Phi = \operatorname{diag}(1, \epsilon)$ is a hyperparameter
> of the prior distribution on $\Gamma$, not part of the
> data-generating equation. We place a Matrix Angular
> Central Gaussian (MACG) prior on $\Gamma$, which is a
> natural distribution on the Stiefel manifold with density
> proportional to $|\Gamma^\top \Phi^{-1} \Gamma|^{-r/2}$.
> Smaller $\epsilon$ yields a more anisotropic and
> numerically challenging posterior. We will clarify these
> roles in the revision.
>
>
> **Answer for Q3**: One natural application domain is physics. For example,
> in lattice gauge theory, computing physical observables
> requires Monte Carlo integration over matrix-valued
> variables taking values in the special unitary group
> $SU(N)$ (Duane et al., 1987; Christ et al., 2025). This
> provides a related application of our method, since
> polar-decomposition-based ambient-space formulations can
> become singular or ill-conditioned, as in the
> Stiefel-manifold example.
>
> Additionally, high-dimensional polytope sampling in systems biology (Schellenberger and Palsson, 2009; Haraldsdóttir et al., 2017) provides another setting where constrained sampling is important and our framework is applicable.
>
> We will add a discussion of these application domains to the Discussion section of the revised paper.
>
> -Duane, S., Kennedy, A. D., Pendleton, B. J., \& Roweth, D. (1987). Hybrid monte carlo. Physics letters B, 195(2), 216-222.
>
> -Christ, N. H., Jin, L. C., Lehner, C., Lundstrum, E., \& Matsumoto, N. (2025). Extended framework for the hybrid Monte Carlo in lattice gauge theory. Physical Review D, 112(3), 034507.
>
> -Schellenberger, J. and Palsson, B.Ø. (2009). Use of randomized sampling for analysis of metabolic networks. Journal of Biological Chemistry, 284(9), 5457–5461.
>
> -Haraldsdóttir, H.S., Cousins, B., Thiele, I., Fleming, R.M.T., and Vempala, S. (2017). CHRR: coordinate hit-and-run with rounding for uniform sampling of constraint-based models. Bioinformatics, 33(11), 1741–1743.
>
> **Minor Weaknesses/Typos**
>
> **Answer**: We agree with all three comments and will revise the manuscript accordingly.
>
> For line 28, we will explicitly introduce $\varepsilon$ when it first appears and state that it is the accuracy parameter in the mixing-time bound, i.e., the tolerance level in total variation distance. Likewise, for line 244, we will introduce $R$ at first use and explain its role as the diagonal scaling matrix used to break the right-orthogonal rotational non-identifiability of the likelihood. These quantities are described later in the experimental details, but we agree they should already be defined at their first appearance in the main text.
>
> For the tables, we also agree that the direction of improvement should be stated explicitly. We will revise the captions/text to indicate that larger ESS is better, whereas PSRF (split-$\hat R$) is better when it is closer to 1 (equivalently, smaller deviation from 1 is better). Appendix C already states that values close to 1.0, typically below 1.05, indicate satisfactory convergence, and we will make this explicit in the main text/table captions as well.
>
> Regarding the boldface, the current formatting is indeed inconsistent. We will change this so that, for each setting and each metric, the best valid result is highlighted in bold and the caption states this explicitly. For ESS, this will mean the larger value is bolded; for PSRF, the value closer to 1 will be bolded. We will apply this convention consistently in Table 1 (and similarly in the appendix tables).

---

> > ### Author Rebuttal · Reviewer_Sg7u · 2026-04-01
> >
> > I thank the authors for their clear response it makes the paper more clear to me. As stated in my review, I am not an expert of the field, so I trust the other reviewers that might be more experts than me. I thank the authors in particular for the responses on potential applications of this work.
> > To my understanding I think that the paper is valuable and should be published. I keep my positive score.

---

> > > ### Author Response · Authors · 2026-04-08
> > >
> > > We thank the reviewer for the encouraging feedback and support. We are pleased that our responses helped clarify the value of this work, and we will reflect the discussed improvements in the final version.

---

### Official Review · Reviewer_RuQq · 2026-03-08

**Soundness:** 3
**Presentation:** 2
**Significance:** 2
**Originality:** 3
**Overall Recommendation:** 4
**Confidence:** 2

**Summary:**

The paper studies Reflective Hamiltonian Monte Carlo (ReHMC) for sampling from target distributions supported on bounded domains. Under smoothness assumptions, the authors establish (non-asymptotic) mixing-time bounds for ReHMC with dimension dependence $d^2$. The framework is applied to sampling on the Stiefel manifold compared with a baseline HMC method which demonstrates improved numerical stability and effective sample size in this application.

**Compliance With Llm Reviewing Policy:**

Affirmed.

**Key Questions For Authors:**

1. The algorithm operates on $\tilde{K}$ while targeting the original density via thinning. When $\tilde{\pi}(K)$ becomes small, does the thinning step significantly degrade mixing or effective sample size?
2. Do any of the results presented apply to non-smooth boundaries, e.g., polytopes?
3. I wonder how the method behaves in higher-dimensional regimes, and whether the $d^2$ factor is observed empitically?

**Limitations:**

See weaknesses and questions to authors.

**Strengths And Weaknesses:**

Strengths:
- The paper addresses a gap in the analysis of reflective HMC methods on bounded domains.
- To the best of my reading, the theoretical analysis is sound and correct.

Weaknesses:
- The theoretical guarantees rely on strong geometric assumptions on the support set $K$ including smooth boundaries and bounded curvature, which seem like it would exclude many common constrained domains (e.g, simplices?).
- The empirical evaluation focuses primarily on moderate-dimensional Stiefel problems.

---

> ### Author Rebuttal · Authors · 2026-03-31
>
> We appreciate the reviewer’s thoughtful comments. We address each point in detail.
>
> **Key Questions**
>
> **Answer for Q1**: The impact of thinning is governed by the retention probability
> $\alpha := \tilde{\pi}(K)$, i.e., the probability under the extended target $\tilde{\pi}$ that a draw lies in the original support $K$.
>
> At the theoretical level, the mixing times satisfy the bound $k_{\mathrm{thin}}(\varepsilon)
>   \le k_{\tilde{\pi}}(\alpha\varepsilon/2),$
> so the cost of thinning enters through the factor $\alpha$.
> Substituting into the extended-target bounds of
> Theorems 4.2 and 4.4 yields the explicit guarantees $$k_{\mathrm{thin}}(\varepsilon)
> \lesssim
>   \frac{R}{\eta^2}
>   \exp \bigl(8RG + (8L+2)R^2 + 1\bigr)
>   \log\frac{\beta}{8\alpha\varepsilon}
>   \log\frac{2}{\alpha\varepsilon} $$
> in the general smooth case, and
> $$  k_{\mathrm{thin}}(\varepsilon)
>   \lesssim
>   \frac{1}{m\eta^2}
>   \log\Bigl(\log\frac{2\beta}{\alpha\varepsilon}+1\Bigr)
>   \log\frac{2}{\alpha\varepsilon} $$
> in the strongly log-concave case. In both bounds $\alpha$
> appears only inside logarithmic terms, so when $\alpha$ is
> bounded away from zero the overall mixing order is preserved
> and the effect of thinning is mild. By contrast, when
> $\alpha$ is very small, thinning can substantially reduce
> practical efficiency.
>
> Regarding the effective sample size, the expected number of retained samples
> is $\mathbb{E}[N_{\mathrm{ret}}]=\alpha N$, so the effective
> sample size scales approximately as
> $ESS_{\mathrm{thin}} \approx \alpha ESS_{\mathrm{ext}}$.
>
> In our experiments, the penalty parameters are chosen so that $\alpha$ remains
> close to one. For example, in the Stiefel manifold experiments of Section 6,
> the observed retention rates are all above 80% across all configurations, confirming that thinning does not significantly degrade
> the effective sample size in practice.
>
> We will explicitly discuss the mixing-time guarantees for the thinning procedure in the revised paper.
>
> **Answer for Q2**: Yes. Although our construction assumes a $C^2$ boundary for analytical
> convenience, the method can be applied to non-smooth domains such as
> polytopes via a smooth surrogate. The approximation error can be controlled
> to be arbitrarily small; we refer the reviewer to our responses to Reviewer
> oCQf (first and second comments) for the detailed approximation analysis.
> In this response, we focus on the empirical performance.
>
> To demonstrate this empirically, we applied PR-HMC to standard convex
> polytopes (a $100$-dimensional cube and simplex) with a truncated Gaussian
> target $\pi(x) \propto \exp(-\|x - x_c\|^2/2)$ centered at the Chebyshev
> center, and compared it with Hit-and-Run (H\&R) and Coordinate Hit-and-Run
> (CH\&R) from the HOPS library (Jadebeck et al., 2020). All methods used
> 4 chains $\times$ 2000 samples. Results for $d=100$ are:
> | Method | ESS (100-Cube) | PSRF (100-Cube) | ESS (100-Simplex) | PSRF (100-Simplex) |
> |---|---:|---:|---:|---:|
> | **PR-HMC (ours)** | **2568** | **1.007** | **2742** | **1.004** |
> | CH&R-HOPS | 49 | 1.257 | 37 | 1.356 |
> | H&R-HOPS | 14 | 2.190 | 12 | 2.756 |
>
> PR-HMC is the only method that achieves convergence (PSRF$<1.05$)
> in $d=100$, with average ESS exceeding $2500$ and more than $50\times$
> higher than the baselines. These results suggest that gradient-based
> exploration can be substantially more effective than random-walk methods
> in high-dimensional constrained spaces, including polytopal constraints.
>
> We will add a remark in the main text on applicability to non-smooth domains such as polytopes, and include the detailed polytope experiments in the Appendix.
>
> - Jadebeck et al. (2021). HOPS: high-performance library for non-uniform sampling of convex-constrained models.
>
> **Answer for Q3**: To investigate the dimension dependence, we considered a simple truncated
> Gaussian benchmark,
> $$\pi(x) \propto \exp(-\|x-\mu\|^2/2\sigma^2)I (x \in [-1,1]^d),$$
> with $\mu = (0.5, \ldots, 0.5)$ and $\sigma = 1.5$.
>
> We evaluate the mixing time as the number of iterations
> at which the maximum coordinate-wise split-$\hat{R}$
> (Vehtari et al., 2021) falls below $1.05$, which is denoted by $k$. We then plot
> $\log k$ versus $\log d$; the slope of this log-log plot
> estimates the exponent $\alpha$ in $k = O(d^\alpha)$.
>
> The log-log plot is provided in the link: https://drive.google.com/thumbnail?id=1uXVY3I0xV94OCRG1I9mR8KJzAJi1uT_C&sz=w2000.
>
> The figure shows that the estimated slope is approximately $1.96$, which is
> empirically consistent with the $O(d^2)$ dependence in
> our theoretical bound. In this experiment, we use step
> size $\eta = 0.3/d$ (consistent with the step-size condition in Theorem 4.2) and four
> chains initialized near different corners of $[-1,1]^d$ to ensure diverse initial conditions.
>
> In a future revision, we will add this experiment in the Appendix.
>
> -Vehtari et al. (2021). Rank-normalization, folding, and localization: An improved R for assessing convergence of MCMC.

---

> > ### Author Rebuttal · Reviewer_RuQq · 2026-04-02
> >
> > My questions and clarifications have been resolved. Thank you to the authors. Even with the inclusion of these clarifications and improvements, I feel my original score is appropriate.

---

> > > ### Author Response · Authors · 2026-04-08
> > >
> > > We appreciate the constructive comments and the time taken to carefully review our responses. We are glad that our clarifications were helpful in addressing the raised questions. We will ensure that all discussed improvements are incorporated into the final manuscript.

---

### Official Review · Reviewer_oCQf · 2026-03-12

**Soundness:** 3
**Presentation:** 3
**Significance:** 3
**Originality:** 3
**Overall Recommendation:** 5
**Confidence:** 3

**Summary:**

This paper develops an MCMC method for sampling from distributions on bounded & constrained domains.    The proposed approach is an extension of HMC which can handle reflections at boundaries.  Similar approaches have been studied before, along with other methods such as hit-and-run and PDMP-type methods.   A key novel idea in the paper is dealing with domains which are not convex & smooth, namely by extending the domain, and extending the potential with a penalising confining potential term, and then post-processing the effect of that out after the sampling has been done via IS.   This lets them handle the problem in a nicer setting with a convex $C^2$ boundary.

In the general potential case, the authors derive a mixing bound which is (importantly) quadratic with respect to dimension,  this is favourable compared to the cubic dependencies you see in, e.g. hit and run type approaches.  However, it can depend exponentially on geometry and smoothness constants.  In the specific strongly convex case the dependence is quadratic, but scaling roughly like the condition number.

The authors apply this approach to sampling on the Stiefel manifold, via an Euclidean parametrisation.   The reflecting hmc method is used to sample in this new space, thinning appropriately.   This avoids the numerical issues of previous variants, e.g. based on polar-expansions.

The method is then demonstrated empirically on two other relevant settings, matrix von mises fisher distributions and Bayesian PCA with orthogonal loadings.  Experiments suggest that the new approach is competitive with other approaches, and notably more stable for ill conditioned problems.

**Compliance With Llm Reviewing Policy:**

Affirmed.

**Key Questions For Authors:**

1. Can you make explicit the distinction between the assumptions of the theorem and the practical approximation in the convex-container + thinning approach?

2. What is the  class of supports covered by the arbitrary bounded support claim?  There appears to be a bit of a gap between Section 3 where it is claimed that ReHMC works for arbitrary bounded sets and Appendix A which assumes a $C^2$ boundary description near the boundary.

3. Are the algorithms and the theory intended to cover a general mass matrix $M$, or only the identity-mass case?   There are various places where the paper switches back to the isotropic $M=I$ case.   Maybe this is just a typo?

4. Numerical experiments: Can the authors clarify the retention / thinning process?  The authors state comparisons are made under identical computational budgets, but to be precise, is that taking into account the thinning?   This should be made explicit, either way.

**Limitations:**

No, I don't think authors addressed the limitations anywhere.

**Strengths And Weaknesses:**

Soundness:  This is a theoretically strong paper.  I have not checked the proof line by line, but the strategy looks sensible, with local TV contraction, conductance and blocking-conductance arguments (like Kannan et al).  The result gives a $d^2$ convergence bound, albeit with exponential dependence on $G, L, R$ in the general case, and this all appears sensible.  Some things are "swept under the carpet" between the main text and the appendix.   For example, in the convex-container plus thinning method, exactness hinges on an exactly vanishing surrogate, but then in the Appendix they leverage softplus as a practical alternative.

Similarly, the claim of arbitrary bounded supports is a bit strong, given the constraints in Appendix A.   This is still a wide and very useful range of sets, but this should be clarified in the main text.

The experiments section is generally strong, and supportive of the claims in the paper, yielding improved robustness.

Presentation:   The paper is very readable.  I enjoyed reading it, and it's a nice flow from theory to application.

Significance:  The paper addresses a meaningful problem, and provides a definite improvement (in terms of convergence) over the prior works in this space.   The specific application to Stiefel is an important contribution, and meaningful in its own right.

Originality:   To my knowledge this is a novel approach, combining thinning, domain extension with confining potential to deal with such domains is a new and very sensible strategy.  That being said, the paper could do a better job of separating what is new from what is inherited.  For example, the warm start argument follows the construction of Chalkis et al (2023),  and the proof follows very similar lines.  This could be drawn out more and made explicit.

---

> ### Author Rebuttal · Authors · 2026-03-31
>
> We thank the reviewer for the insightful comments. Our responses are as follows.
>
> **Key Questions**
>
> **Answer for Q1**: We agree that this distinction was not made sufficiently explicit. In a future revision, we will add a remark in the main text clarifying that while the theoretical argument assumes an exactly vanishing surrogate, the softplus approximation introduces a nonzero residual and is adopted purely for computational convenience.
>
> Although the resulting bias can be made small on $K$, this was not quantified in the previous version. We will include the following bound: the pointwise error on $K$ satisfies
> $$|\tilde{U}_\sigma(q) - U(q)| \leq \lambda \left(\frac{\log 2}{\sigma b}\right)^4,$$
> which decreases as $O(\sigma^{-4})$ and can be made arbitrarily small by increasing $\sigma$.
>
>
>
> **Answer for Q2**: We agree that the scope of the "arbitrary bounded
> support" claim should be stated more precisely. The key
> distinction is between (i) an abstract existence result
> and (ii) a concrete construction used in our
> implementation. At an abstract level, Section 3
> establishes that the convex-container-plus-thinning
> framework applies to arbitrary bounded supports via
> Whitney-type extension results, which do not require a
> $C^2$ boundary. In contrast, the explicit construction
> in Appendix A assumes a $C^2$ boundary.
>
> However, the $C^2$ assumption in the penalty-based
> construction does not limit its practical applicability.
> For non-smooth domains such as polytopes, we approximate
> the non-smooth boundary with a $C^\infty$ surrogate via
> softplus, which recovers the original domain as
> $\sigma \to \infty$. This allows our construction to be
> applied approximately to non-smooth supports, with the
> approximation error controlled by $\sigma$.
> We will clarify this distinction in a future revision.
>
>
> **Answer for Q3**: The algorithms (Algorithms 1-3) are stated for a general mass matrix $M \succ 0$. In the previous version, however, the proofs in Appendix B were carried out under the simplifying assumption
> $M = I$, which was not stated clearly enough and could indeed create the
> impression of an inconsistency.
>
> The extension to general $M \succ 0$ follows by replacing the Euclidean inner product with the $M$-inner product.
> As long as the minimum and maximum eigenvalues of $M$ are bounded away from zero and infinity, this introduces only
> constant factors into the existing bounds without altering the structure of the proofs. We will revise the proofs to explicitly handle general $M \succ 0$.
>
> **Answer for Q4**: We clarify how the thinning process is reflected in the ESS evaluation for our method. We run 2,000 warm-up and 2,000 sampling iterations across four parallel chains. ESS is computed only from the retained samples after thinning. The discarded samples incur additional computational cost but are not counted in the ESS, making the comparison conservative with respect to our method. We will make these computational details explicit in a future revision.
>
> **Originality**
>
> **Q. The paper could do a better job of separating what is new from what is inherited. For example, the warm start argument follows the construction of [2], and the proof follows very similar lines. This could be drawn out more and made explicit.**
>
> **Answer**: We agree that the manuscript should more
> clearly delineate inherited and new components. The two closely related works are [1] and
> [2].
>
> Comparison with [1]: the high-level proof architecture,
> local TV contraction combined with a blocking-conductance bound, follows
> their framework. What is new is the local TV contraction for the
> reflective proposal (Lemma B.5), supported by the
> reflection-geometry estimates in Appendix B.4 (Lemmas B.7-B.19). These
> arguments are specific to the reflective setting and have no counterpart
> in unconstrained HMC.
>
> Comparison with [2]: the warm-start construction in
> our paper (Lemma 4.3) adapts their truncated-Gaussian initialization
> (Theorem 2). However, [2] do not establish a
> non-asymptotic mixing-time bound for ReHMC.
>
> We will revise the paper to include a paragraph in the main text that explicitly summarizes our theoretical contributions relative to [1] and [2]. We acknowledge that, in particular, the novelty of our contributions over [1] was not sufficiently highlighted in the previous version, and we will address this in the revision.
>
> [1] Lee et al. (2020) Logsmooth gradient concentration and tighter runtimes for metropolized hamiltonian monte carlo.
>
> [2] Chalkis et al. (2023) Truncated log-concave sampling for convex bodies with reflective hamiltonian monte carlo.

---

> > ### Author Rebuttal · Reviewer_oCQf · 2026-04-02
> >
> > I thank the authors for their well-considered comments and clarifications.  I maintain my score as is, which I believe is a fair reflection of the paper and the proposed changes.

---

> > > ### Author Response · Authors · 2026-04-08
> > >
> > > We thank the reviewer for the thoughtful engagement throughout the discussion period and for the careful evaluation of our work. We appreciate the feedback and will ensure that all discussed improvements are reflected in the final version.

---

### Decision · Program_Chairs · 2026-04-30

**Decision:**

Accept (regular)

**Comment:**

This paper is concerned with the problem of sampling from distributions with bounded support. It is motivated by the sampling method, Reflective HMC (ReHMC), which relies on convexity assumptions and lacks non-asymptotic theoretical guarantees. The paper proposes a framework that proposes a convex-container plus thinning, and provides non-asymptotic mixing time bounds. The efficacy of the approach is demonstrated to the application of sampling from the Stiefel menifold.

All reviewers are in favor of accepting this paper, and so I am happy to recommend accepting the paper. The reviewers have a very positive view on both the strength of the theoretical results  as well as the empirical results in this paper. Please take into account the reviewer feedback on clarity, notation, and theoretical assumptions for the final revision.